

# Synoptic perspective on the conversion and maintenance of local available potential energy in extratropical cyclones

Marc Federer[1], Lukas Papritz[1,2], Michael Sprenger[1], and Christian M. Grams[3,4]

[1]Institute for Atmospheric and Climate Science, ETH Zurich, Zurich, Switzerland
[2]European Centre for Medium-Range Weather Forecasts (ECMWF), Bonn, Germany
[3]Institute of Meteorology and Climate Research Troposphere Research (IMKTRO), Karlsruhe Institute of Technology (KIT), Karlsruhe, Germany
[4]now at: Federal Office of Meteorology and Climatology, MeteoSwiss, Zurich Airport, Switzerland

**Correspondence:** Marc Federer (marc.federer@env.ethz.ch)

**Abstract.**

Extratropical cyclones are the predominant weather system in the midlatitudes. They intensify through baroclinic instability, a process in which available potential energy (APE) is converted into kinetic energy (KE). While the planetary-scale conversion of APE to KE is well understood as a mechanism for maintaining the general atmospheric circulation against dissipation, the synoptic-scale perspective on this conversion is less explored. In this study, we analyze the three-dimensional distribution of APE and the physical processes that consume APE for an illustrative case study and a climatology of 285 intense North Atlantic extratropical cyclones in the winters 1979–2021 using the ERA5 reanalysis. We utilize a recently introduced local APE framework that allows APE to be quantified at the level of individual air parcels. The geographical APE distribution is shown to be controlled by the large-scale upper-level circulation. Cyclones draw energy from the upper-tropospheric polar APE reservoir along with the development of the associated upper-level trough. This upper-level APE is converted into KE by air descending along the trough's western flank and acts as the incipient cyclone's primary source of KE. Conversely, KE is converted back into APE during the ascent ahead of the trough, reflecting the deceleration of air parcels as they exit the cyclone region. The diabatic dissipation of APE due to surface sensible heat fluxes along the Gulf Stream front is small compared to the adiabatic conversion of APE to KE, since most of the APE is concentrated and consumed in the middle to upper troposphere and cannot be exposed to surface diabatic forcing. In conclusion, by employing a local APE framework, this study provides a detailed investigation of the synoptic dynamics linking extratropical cyclones and planetary-scale energetics.

## 1 Introduction

The global atmospheric circulation in midlatitudes is maintained against friction by the conversion of available potential energy (APE) to kinetic energy (KE). Margules (1903) recognized that only a minor fraction of the total potential energy of the



atmosphere can be converted to KE by an adiabatic redistribution of mass. This fraction of the total potential energy later became know as the APE. In his seminal work, Lorenz (1955) defined APE as the difference between the total potential energy of the atmosphere and the minimum potential energy attainable by an adiabatic rearrangement, a state he referred to as the reference state. This hypothetical reference state represents an atmosphere in mechanical equilibrium, where the highest
density air is at the bottom and the lowest density air is at the top of the atmosphere, such that no meridional density gradients exist. A deviation in the mass distribution from the reference state indicates the availability of potential energy, that can be converted into KE through the sinking of high-density air and the rising of low-density air.

The dominant physical mechanism for the conversion of APE to KE in midlatitudes is given by baroclinic instability (Charney, 1947; Eady, 1949), which leads to an equatorward and downward movement of cold air, as well as a poleward and
upward movement of warm air within intensifying midlatitude baroclinic waves, and thus in particular in extratropical cyclones (Browning and Roberts, 1994). Within the global APE budget, this baroclinic conversion results in the consumption of APE, while the differential heating of the atmosphere by the solar irradiance restores APE (Oort, 1964; Peixóto and Oort, 1974).

The Lorenz APE is defined as a volume integral of a closed system, which is why previous studies have largely focused
on the global integral of APE. Bosart et al. (1996) pointed to the significance of individual synoptic-scale weather systems for the planetary-scale APE reservoir. They demonstrated that the development of a single explosive extratropical cyclone over North America was concurrent with a 10% drop in northern hemispheric APE. Further studies have underscored the relevance of synoptic-scale dynamics for the variability of APE (Wintels and Gyakum, 2000; Bowley et al., 2019). However, the volume-integrated formulation of Lorenz APE limits its applicability on synoptic scales. A decomposition into eddy and
mean components allows for the attribution of volume-integrated APE conversion to synoptic-scale dynamics (Lorenz, 1955; Peixóto and Oort, 1974), but it remains challenging to attribute such time-filtered quantities to individual weather systems. The Lorenz APE can be localized by selecting a local domain, such as a box around a cyclone, within which the reference state is calculated (Smith, 1969, 1980; Johnson, 1970; Gertler et al., 2023). However, a local reference state makes it difficult to quantify the advection of APE into and out of the local domain. Moreover, a cyclone does not represent a closed system,
complicating the interpretation of energy exchange between the local and the global domain. In fact, as we will show here, the advection of high-APE air into the environment of the extratropical cyclones is a key component in their energy budgets.

In order to describe APE conversion within weather systems, it is necessary to employ a truly local APE framework. Holliday and McIntyre (1981) and Andrews (1981) introduced local frameworks that describe APE as a local energy density for a compressible nonhydrostatic fluid and an incompressible fluid, respectively. Subsequently, Novak and Tailleux (2018) extended
this framework to the dry hydrostatic primitive equations and demonstrated that the global integral of this local APE density is equivalent to the Lorenz APE.

The climatological analysis of Novak and Tailleux (2018) reveals spatial characteristics of APE and its conversion, which were previously discussed only as global integrals (Oort, 1964; Peixóto and Oort, 1974). In particular, they find that APE is mainly located in the polar middle and upper troposphere from where it is advected into the storm tracks. There eddy APE is
converted to eddy KE by adiabatic processes, while diabatic processes create eddy APE. Since their analysis was climatological,



many questions remain regarding the synoptic environment leading to the advection of APE into the storm tracks, the physical processes giving rise to adiabatic and diabatic tendencies of APE and their link to baroclinic wave activity in the storm tracks.

In their analysis of local APE within an idealized baroclinic channel simulation, Federer et al. (2024) employed Lagrangian trajectories to demonstrate how APE is advected into the baroclinic zone along the upper-level trough in the cold sector of a surface cyclone. Their findings revealed that locally within a baroclinic wave, substantial amounts of APE are not only consumed, but also produced. This is consistent with a local notion of APE, where air parcels accelerate by converting APE to KE, but also decelerate again by converting KE back to APE. The volume-integrated net conversion then informs about the intensification or the decay of the baroclinic wave in terms of its KE. However, important differences exist between the highly idealized baroclinic channel and the real atmosphere.

First of all, the reference state based on an adiabatic rearrangement of the whole atmosphere differs from the reference state computed from a symmetric baroclinic channel due to the spherical geometry of Earth. Consequently, we anticipate that APE conversion in the real atmosphere will differ from that observed in a baroclinic channel (Federer et al., 2024). Second, the baroclinic channel does not include topography, land-sea contrasts or surface fluxes, which we expect to significantly influence the distribution and conversion of APE (Brayshaw et al., 2009, 2011; Molteni et al., 2011; Portal et al., 2022).

In particular, the northern hemispheric storm tracks are significantly influenced by strong sea-surface temperature (SST) fronts along the western boundary currents. The intense heat transfer from the ocean to the atmosphere in these regions (Kwon et al., 2010; Czaja et al., 2019) is known to affect the large-scale circulation (Omrani et al., 2019; Mathews and Czaja, 2024; Wenta et al., 2024). On one hand, latent heat release following moisture uptake along the Gulf Stream SST front (Pfahl et al., 2014) is known to invigorate cyclone development (Davis and Emanuel, 1991; Dacre and Gray, 2013; Binder et al., 2016) and to contribute to downstream ridge building (Yamamoto et al., 2021; Wenta et al., 2024). On the other hand, it is generally believed that SST fronts anchor midlatitude storm tracks through a process called oceanic baroclinic adjustment (Nakamura et al., 2008; Hotta and Nakamura, 2011). Essentially, the SST front dampens local atmospheric temperature anomalies induced by the meridional advection within developing cyclones, thereby maintaining low-level baroclinicity. A number of studies have indicated that this process results in the dissipation of APE, dampening the development of baroclinic waves (Swanson and Pierrehumbert, 1997; Marcheggiani and Ambaum, 2020). In contrast, strong surface sensible heat fluxes that occur in marine cold air outbreaks (CAO) following cyclones have been linked to the maintenance of low-level baroclinicity by reducing static stability (Papritz and Spengler, 2015), which amplifies baroclinic growth rates. This points towards an ambiguous role of sensible and latent heat fluxes along the North Atlantic storm track in the local APE budget.

The objective of this study is to gain a comprehensive three-dimensional understanding of the distribution and conversion of local APE along the North Atlantic storm track through both adiabatic and diabatic processes, and to elucidate the relationship between local APE and the development of extratropical cyclones. In particular, we will focus on Gulf Stream cyclones in order to investigate the relevance of air-sea interaction on the local APE budget, which potentially has implications for the downstream storm track evolution (Wenta et al., 2024). To this end, we address the following specific questions:

– What determines the distribution and conversion of local APE on synoptic scales over the North Atlantic?



– How does the development of Gulf Stream cyclones influence the distribution and conversion of local APE?

– What is the role of air-sea interaction in the Gulf Stream region in the local APE budget?

The study is structured as follows. In section 2, we will introduce the dataset, the identification and tracking of cyclones and the local APE framework. Then in section 3, we present a case study of a well-studied rapidly intensifying cyclone in the Gulf Stream region (e.g. Wenta et al., 2024). In section 4, we extend our understanding of the case study to a set of 285 deep Gulf
Stream cyclones. Finally, the study concludes with a discussion in section 5.

## 2    Data and Methodology

### 2.1    Dataset

This study relies on the ERA5 reanalysis dataset provided by the European Centre for Medium-Range Weather Forecasts (ECMWF; Hersbach et al. 2020). We utilize data on model levels interpolated to a $0.5° \times 0.5°$ longitude-latitude grid at hourly
resolution. Our analysis spans 41 winter seasons (December to February) between 1979 and 2021, encompassing data from 1 Dec 1979 to 28 Feb 2021.

The pressure at the boundary layer top is diagnosed from the boundary layer height provided by ERA5. In addition, we compute quasi-geostrophic $\omega$ (QG$\omega$) by inverting the QG$\omega$-equation following Besson et al. (2021). For this study we only consider QG$\omega$-forcing at 500 hPa from upper levels.

For studying local APE conversions in cyclones, we objectively identify and track cyclones based on the sea level pressure (SLP) field using the contour search algorithm developed by Wernli and Schwierz (2006) and refined by Sprenger et al. (2017). Each cyclone is characterized by a pressure minimum, which is tracked over time to generate cyclone tracks, along with an outer-most closed contour in the SLP field. The difference between this outer-most closed contour and the pressure minimum defines the cyclone's depth, serving as a measure of its intensity. Furthermore, we compute the intensification rate of a cyclone
at a given time step by evaluating its pressure change over a 24 h period centered on that time step. As an additional weather feature, we also identify CAOs following the definition by Papritz et al. (2015), i.e. CAO grid points are identified using the criterion $\theta_{\mathrm{SST}} - \theta_{850} > 8\,\mathrm{K}$, where $\theta_{\mathrm{SST}}$ is the potential sea surface temperature and $\theta_{850}$ is the potential temperature at 850 hPa.

Additionally, we use the Lagrangian Analysis Tool (LAGRANTO; Wernli and Davies 1997; Sprenger and Wernli 2015) to compute kinematic backward and forward trajectories. The integration of APE changes along trajectories has proven to be a
valuable technique for relating the movement of air parcels to changes in the APE distribution (Federer et al., 2024).

### 2.2    Local APE framework

In this study we use the local APE framework for a dry hydrostatic atmosphere developed by Novak and Tailleux (2018). Here, we present a brief overview of the most relevant features of this framework. For a detailed introduction into the use of local APE on synoptic scales we refer to Federer et al. (2024).





The local APE density is defined as

$$E_a(\theta,p,t) = \int\limits_{p_r}^{p} \{\alpha(\theta,p') - \alpha[\theta_r(p',t),p']\}\mathrm{d}p', \tag{1}$$

where the APE density ($\mathrm{J\,kg^{-1}}$) of an air parcel is denoted by $E_a$ and $\alpha$ is the specific volume. The integrand of Eq. 1 is the buoyancy force experienced by an air parcel at pressure $p'$ within the reference atmosphere, given by $\theta_r(p',t)$. The reference pressure $p_r$ is determined by the level of neutral buoyancy (LNB) equation $\theta_r(p_r,t) = \theta(p,t)$. Therefore, the buoyancy force

experienced by the air parcel is zero when the parcel's density matches the surrounding air at the reference pressure $p_r$. The buoyancy force is maximal at the pressure $p$ where the density difference between the air parcel and the reference environment is greatest. Then, Eq. 1 expresses the work needed to move the air parcel, within the reference atmosphere, from its reference pressure $p_r$ to its actual pressure $p$.

APE of an air parcel can change by an adiabatic vertical displacement of the air parcel, by diabatic heating or cooling of the

air parcel, or by a change in the reference state. Formally, this is expressed by the total derivative of Eq. 1 representing the total change of APE along the parcel's trajectory. This change is given by

$$\frac{\mathrm{D}E_a}{\mathrm{D}t} = \underbrace{(\alpha - \alpha_r)\omega}_{(I)} + \underbrace{\left(\frac{T - T_r}{T}\right)Q}_{(II)} - \underbrace{\int\limits_{p_r}^{p} \frac{\partial \alpha_r}{\partial t}\mathrm{d}p'}_{(III)}, \tag{2}$$

where $\omega$ is the vertical pressure velocity, $Q$ is the diabatic heating rate and $T$ is the temperature. According to Eq. 2, the adiabatic tendency of APE (I) is given by $\omega$ and the buoyancy of the air parcel with respect to the reference state ($\alpha -$

$\alpha_r$), which we refer to as the adiabatic efficiency. The diabatic tendency (II) is given by the product of $Q$ and the thermal efficiency $(T - T_r)/T$. Finally, the third term (III) expresses the APE tendency due to a change in the reference state with time. Consequently, the APE tendency due to adiabatic and diabatic processes does not only depend on the magnitude of $\omega$ and $Q$, respectively, but also on the respective efficiency, which quantifies the departure of the air parcel from its reference state. Because the adiabatic (I) and diabatic (II) tendencies of APE are much larger than the tendency due to changes in the reference

state (III), in this study, we will focus on the adiabatic and diabatic tendency.

### 2.3    Computation of the reference state

For the computation of the reference state we follow the parcel-sorting method described by Novak and Tailleux (2018). The atmosphere is partitioned into air parcels, which are subsequently sorted by ascending potential temperature. Next, the reference state is reconstructed by calculating the pressure contribution of each air parcel when its mass is distributed uniformly across

the surface of the Earth, and air parcels are piled up, beginning with the air parcel with lowest potential temperature. This yields a monotonically increasing vertical profile of potential temperature $\theta_r(p,t)$, which is the same for every grid point, but specific to a given hourly time step of ERA5. $\theta_r(p,t)$ describes the state of minimum potential energy attainable by an adiabatic rearrangement of the global atmosphere (Lorenz, 1955; Novak and Tailleux, 2018).



In order to facilitate the comparison of the distribution of APE among cyclones at different times, we remove the diurnal
cycle of the reference state by a daily average. Additionally, we compute a 9-year running mean of the reference state to account
for interannual variations.

## 3 Case study

We first illustrate local APE and its tendencies in a case study to gain a detailed understanding of the synoptic processes that
govern the evolution of local APE. For this purpose, an episode of European blocking was selected, which is defined according
to the definition of the year-round weather regimes of Grams et al. (2017). The event lasted from 20 to 27 Feb 2019 and was
accompanied by record-high temperatures in France, the Netherlands, and the United Kingdom (Young and Galvin, 2020;
Leach et al., 2021). We chose this case study because the build-up and maintenance of the block were linked to several strongly
intensifying extratropical cyclones that developed off the coast of North America and along the Gulf Stream SST front (Wenta
et al., 2024). Such cyclones are expected to export APE from high latitudes to the North Atlantic storm track (Novak and
Tailleux, 2018). In this section, we will illustrate how the large-scale circulation is related to the distribution of local APE,
where APE is consumed and generated in the vicinity of cyclones, and what physical processes drive local APE tendencies
during the case study period. The insights gained in this case study will subsequently be generalized and extended with a
climatological analysis of deep Gulf Stream cyclones in section 4.

### 3.1 Synoptic APE development

### 3.1.1 Column-integrated APE distribution

Figure 1 gives an overview of the large-scale circulation around the case study period. The upper-level circulation is indicated
by the 2-PVU (1 PVU is equivalent to $10^{-6}\,\mathrm{K\,m^2\,kg^{-1}\,s^{-1}}$) contour of isentropic Ertel potential vorticity (PV) at 315 K. On
Feb 18 (Fig. 1a) the first primary cyclone (1a) of the case study begins to develop in the Gulf Stream region in a baroclinic
zone which lies at the edge of large APE values toward the north. As the cyclone continues to intensify and propagates into
the central North Atlantic (Fig. 1b,c), an upper-level trough forms and extends into lower latitudes, while a ridge is established
downstream. Together with the trough, high APE values extend into the North Atlantic. This points to a strong link between
the large-scale circulation and the distribution of APE. As cyclone 1a propagates toward the southern tip of Greenland, a
secondary cyclone (1b) develops on the first cyclone's trailing cold front (Fig. 1d) and propagates poleward (Fig. 1e). The
development of cyclone 1b also occurs adjacent to high APE values. Concurrently, a second primary cyclone (2a) develops
in the Gulf Stream region and follows a very similar evolution to cyclone 1a (Fig. 1d-g). Large APE values persist in the
central North Atlantic and are again clearly collocated with the upper-level trough. Subsequently, another secondary cyclone
(2b) intensifies along the trailing cold front of cyclone 2a. In contrast to cyclone 1b, the high APE values in the central North
Atlantic decrease concurrently with the intensification of cyclone 2b. The decline of midlatitude APE is ushered in by the
development of cyclones 3a and 3b over continental North America. The advection of low-APE air from the south by this





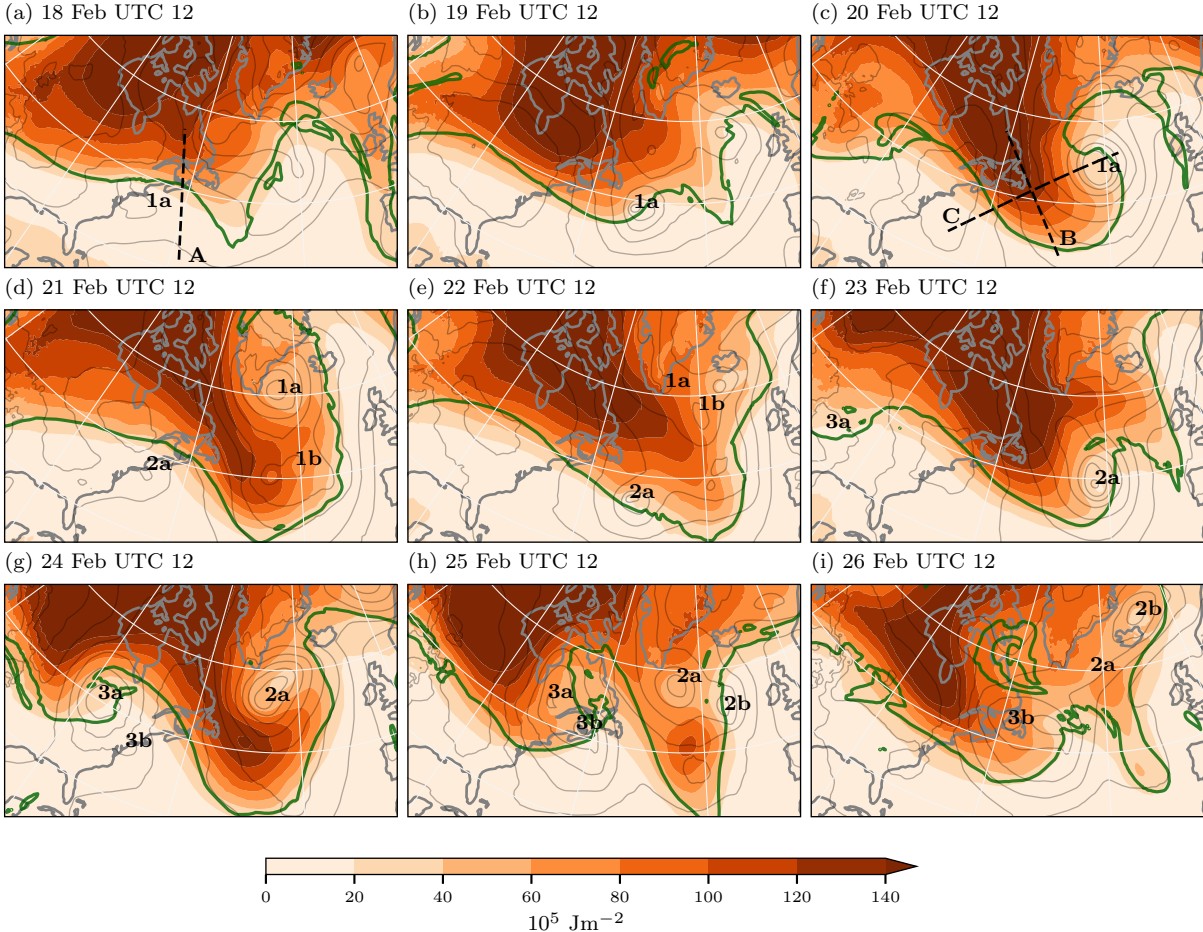

**Figure 1.** Synoptic overview between 18 Feb UTC 12 and 26 Feb UTC 12 at 24 hour intervals. Shown are vertically integrated APE between 1050 and 250 hPa (color shading), mean sea-level pressure (grey contours; every 10 hPa) and the 2-PVU contour on the 315 K isentrope (green). Cyclones discussed in the text are labelled. The dashed lines in panels (a) and (c), labeled by capital letters, match the cross sections in Fig. 2.

low-pressure system cuts off the APE supply from higher latitudes into the North Atlantic storm track, and the quasi-stationary trough/ridge couplet over the North Atlantic decays.

In summary, the case study is characterized by the development of two primary and two secondary North Atlantic cyclones that follow remarkably similar tracks adjacent to regions of high APE values. Those cyclones tap into the reservoir of APE in the Canadian Arctic and advect APE into midlatitudes. The extension of high-APE air into midlatitudes is collocated with an

upper-level trough, which indicates a strong link of the APE distribution to upper level flow features. The decline of midlatitude APE and the end of intense North Atlantic cyclogenesis are accompanied by the development of an upstream cyclone, which inhibits further APE supply to the North Atlantic.



### 3.1.2 Vertical structure

The close match between the large-scale circulation and the distribution of vertically integrated APE warrants an investigation
of the vertical structure of the APE distribution. Figure 2 shows vertical cross sections of the APE distribution along the dashed
lines in Fig. 1. The first cross section (Fig. 2a) shows the APE distribution approximately normal to the baroclinic zone shortly
before the development of cyclone 1a. The baroclinic zone is characterized by a strong meridional temperature gradient near
the surface and a jet stream aloft, which coincides with a steep tropopause. APE is confined poleward of the baroclinic zone,
whereas very little APE is located within the baroclinic zone and equatorward of it. APE also increases with height up to the
tropopause and rapidly decreases with height in the stratosphere.

Figure 2b shows a meridionally oriented cross section through the trough, which stretches into the central North Atlantic after
cyclone 1a intensified. The vertical distribution across the trough is strongly related to the height of the dynamical tropopause.
At the center of the cross section, the tropopause reaches 500 hPa and is accompanied by a local cold anomaly underneath.
This local cold anomaly translates to high APE because the air parcels are colder than their reference state. APE located at the
southern tip of the trough follows the tropopause, which reaches higher altitudes in the atmosphere compared to the center of
the trough.

The zonally oriented cross section through the trough (Fig. 2c) illustrates the isentropic lifting associated with the low
tropopause at the center of the trough. The APE located in this region is bounded by a jet stream toward the west and cyclone
1a toward the east.

The vertical distribution of APE shows that APE is concentrated in the middle and upper troposphere north of the midlatitude
baroclinic zone. The upward bending of isentropes by positive anomalies of PV implies cold air advancing into the midlatitudes
below the trough, which in turn is associated with large APE since it is colder than its reference state. In summary, the close link
between upper-level PV and high-APE air beneath points to the key role of the PV distribution in determining the distribution
of APE.



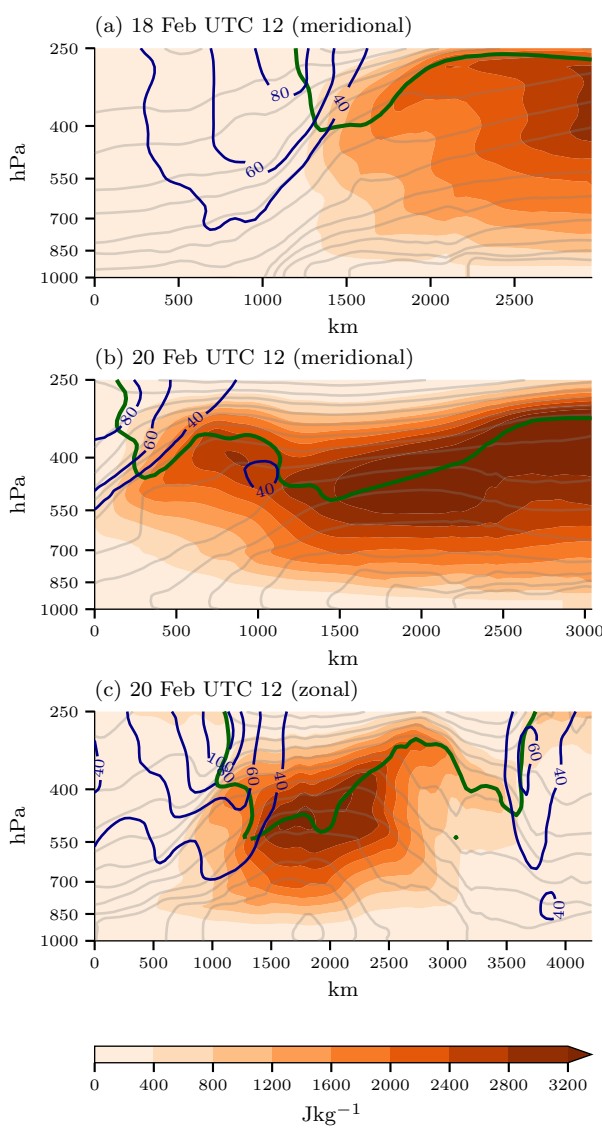

**Figure 2.** Cross sections as indicated in Fig. 1. Shown are APE (color shading), the 2-PVU contour representing the dynamical tropopause (green), the horizontal wind speed (blue contours; every $20\,\mathrm{ms^{-1}}$) and potential temperature (grey contours; every $6\,\mathrm{K}$). For each panel, the position of the cross section is shown in Fig. 1, labeled by capital letters.





## 3.2 Adiabatic and diabatic APE modification within the free troposphere

### 3.2.1 Lagrangian APE tendency

Given that APE is concentrated in the upper troposphere, adjacent to the developing cyclones, the question arises how the cyclones tap into the APE reservoir and how the midlatitude APE reservoir is maintained throughout the case study period. To answer this questions, we make use of Lagrangian trajectories, which allow us to describe the fate of the APE contained in a given air parcel. To this end, we start 48 h forward trajectories at every grid point between 1000 and 250 hPa (every 10 hPa). Next we integrate the APE tendency forward along those trajectories, project this APE change onto the starting position of the respective trajectory, and integrate the resulting field vertically. This yields the Lagrangian APE change of a given column of air within the next 48 h.

Figure 3 shows this vertically-integrated Lagrangian 48 h change of APE from 19 Feb 12 UTC to 21 Feb 12 UTC. The character of the results does not change fundamentally if shorter integration times down to 24 h are chosen. Comparing the APE change (in color) with the APE distribution (black contours), in the case of a negative tendency, informs about the fraction of APE located at any given point which will be consumed in the following 48 h. Conversely, a positive tendency informs about the origin of air parcels, which will experience an increase of APE. Note that this Lagrangian APE tendency does not indicate the change in the APE distribution at a given grid point, but the APE change along the trajectories of air parcels constituting a given column of air.

Generally, we observe that air parcels with large APE experience a loss of APE, and air parcels with low APE experience a gain of APE. The net depletion of APE (Fig. 3a) is largest over the Canadian Arctic. These air parcels are located within the APE reservoir adjacent to the Gulf Stream region, suggesting that this APE reservoir is most relevant for the supply of APE. Positive net APE changes are found at the boundary of the APE reservoir, in particular in the vicinity of cyclone 1a and a low-pressure system downstream.

The separation of the net tendency into adiabatic (vertical motion) and diabatic (diabatic heating and cooling) contributions reveals that the net tendency is dominated by the adiabatic tendency (Fig. 3b). Hence, vertical motion is responsible for the majority of the net APE tendency. The diabatic tendency (Fig. 3c) is weaker than the adiabatic tendency and exhibits a more complex pattern. First, we recognize a wide-spread pattern of positive diabatic APE tendency toward the north and negative diabatic APE tendency toward the south. This pattern is due to radiative cooling of the free troposphere, which is reflected in an APE increase for air parcels colder than the reference state, and an APE decrease for air parcels warmer than the reference state. This planetary-scale forcing of APE is interrupted by synoptic-scale regions of negative diabatic APE tendency within the APE reservoir and slightly downstream of cyclone 1a, which will be addressed in the next section.

This analysis demonstrates that the Lagrangian APE tendency in the vicinity of a cyclone shows a complex pattern of APE consumption and generation. In particular the intensification of the cyclone in this case study is not only accompanied by a loss of APE, but also a gain of APE.



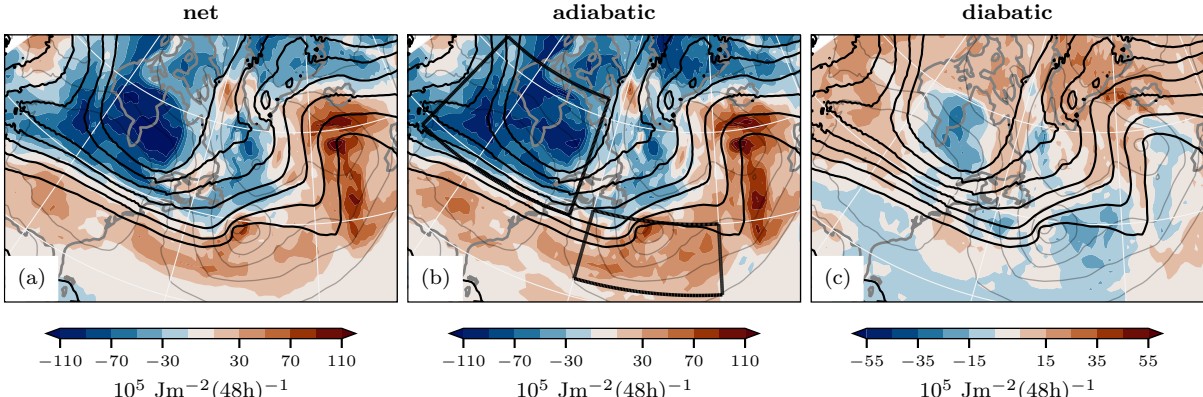

**Figure 3.** Vertically integrated 48 h forward APE change on 19 Feb UTC 12 (color shading). Shown are the (a) net change, (b) adiabatic contribution and (c) diabatic contribution. Also shown are the vertically integrated APE distribution (black contours; same intervals as Fig. 1) and the mean sea-level pressure (grey contours; every 10 hPa). The regions indicated by the two black boxes in (b) show the start locations for trajectories evaluated in Fig. 4.

### 3.2.2 The motion of air parcels giving rise to APE tendencies

In order to understand the physical processes governing APE modification it is instructive to study the motion of the air parcels contributing to the APE budget. For this purpose, we select two regions of particularly large APE tendencies (Fig. 3b). The

first region encompasses the APE reservoir over the Canadian Arctic, for which we will consider air parcels which lose APE. The second region includes the warm sector of cyclone 1a, where we consider air parcels which gain APE. For both regions, we select the air parcels with the largest 48 h AOE changes and which together account for 80 % of the vertically-integrated APE loss and gain, respectively.

Trajectories started in the APE reservoir in the lower troposphere (p>600 hPa; Fig. 4a) lose APE by descending in the North

Atlantic behind the cold front of cyclone 1a. Trajectories started in the APE reservoir in the upper troposphere (p<600 hPa; Fig. 4b) show a different behaviour. Some trajectories descend strongly and arrive in the lower troposphere behind the cold front of cyclone 1a. But other trajectories descend weakly and are advected around the upper-level trough. Still all trajectories contribute significantly to the decrease of vertically-integrated APE.

Trajectories started in the second region within the lower troposphere (p>600 hPa; Fig. 4c) ascend strongly from within the

warm sector of cyclone 1a until they reach the upper troposphere and outflow into the upper-level ridge downstream of cyclone




1a. Since the ascending air parcels gain APE, they must be colder than their reference state, which suggests that the ascent occurs in anomalously cold air with respect to the reference state. The rapid ascent of these trajectories is reminiscent of a warm conveyor belt (WCB), which is associated with strong latent heat release. Indeed, the region of negative diabatic APE tendency observed slightly downstream of cyclone 1a coincides with the ascending air parcels. Thus, latent heat release in the

ascending air stream leads to a destruction of APE. Lastly, trajectories started in the second region within the upper troposphere (p<600 hPa, Fig. 4d) tend to start upstream of cyclone 1a and ascend into the upper troposphere. Since they already start at high altitudes they cannot ascend as much as trajectories from the lower troposphere, but still account for a significant APE production.

The trajectories highlight the role of the upper-level dynamics for the APE budget. While air parcels in the upper troposphere

require little vertical motion to yield a significant change of APE, air parcels from within the lower troposphere require large vertical displacements to contribute substantially to the local APE budget. Latent heat release within the ascending air stream from the warm sector of cyclone 1a is identified as a sink for local APE, which also explains the second maximum of APE loss identified over the Canadian Arctic (Fig. 3c). Some air parcels which descend into the lower troposphere later ascend again. Since the descent incurs a large loss of APE, ascent in the lower troposphere later on does not substantially influence

the integrated adiabatic APE tendency over 48 hours. However, the latent heat release and the associated APE destruction outweighs the APE gain by radiative cooling in the upper troposphere, which yields a negative integrated diabatic APE tendency (Fig. 3c).



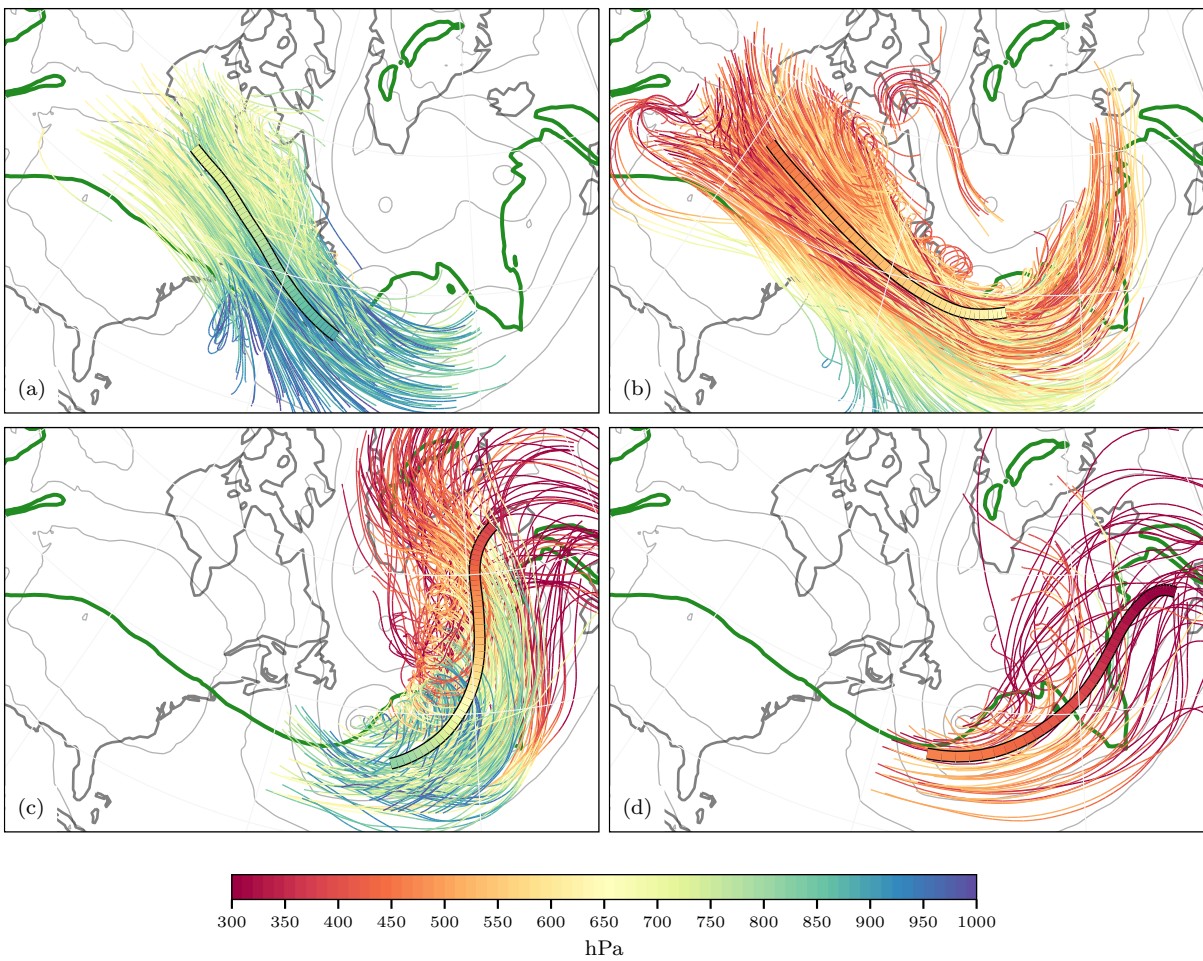

**Figure 4.** 48 h forward trajectories (colored by pressure; every 10th trajectory) initialized within the APE reservoir (top) and within the warm sector of cyclone 1a (bottom). Trajectories are split between starting locations in the upper troposphere (p<600 hPa; b,d) and lower troposphere (p>600 hPa; a,c). For each set of trajectories, a mean trajectory is shown. Also shown are the mean sea-level pressure (grey contours; every 10 hPa) and the 2-PVU contour on the 315 K isentrope (green).





### 3.2.3 Dry-dynamic forcing of the APE tendency

Since the APE density is largest in the upper troposphere, we next study the quasi-geostrophic forcing due to the large-
scale upper-tropospheric circulation. Figure 5 shows the positions of ascending and descending trajectories between 400 and
600 hPa and the upper-level QG$\omega$-forcing at 500 hPa at two time steps. On 20 Feb 00 UTC (Fig. 5a) the pattern of QG$\omega$ shows
a dipole structure of ascent on the eastern flank and descent on the western flank of the trough. The upward QG$\omega$-forcing
is concentrated above the center of cyclone 1a, whereas the downward QG$\omega$-forcing extends westward together with the jet
stream. As cyclone 1a moves further poleward on 20 Feb 12 UTC (Fig. 5b) the dipole structure in QG$\omega$ collocated with cyclone
1a detaches from the band of downward QG$\omega$-forcing on the eastern flank of the trough. Thus, large parts of the downward
ageostrophic circulation due to the trough do not remain collocated with the surface cyclone during its life cycle. The positions
of the descending trajectories show that they are affected by downward QG$\omega$-forcing as they descend. On 20 Feb 00 UTC
(Fig. 5a) the majority of trajectories descend in the vicinity of the jet entrance, while few trajectories descend further south
in the vicinity of cyclone 1a. In the meantime the ascending trajectories are collocated with the upward QG$\omega$-forcing around
cyclone 1a. On 20 Feb 12 UTC (Fig. 5b) the majority of descending trajectories remains within the influence of the downward
QG$\omega$-forcing beneath the upper-level jet, while none of the selected trajectories descend directly within cyclone 1a. In contrast,
the ascending trajectories continue their ascent around cyclone 1a due to the upward QG$\omega$-forcing.

The close association of the movement of air parcels and the QG$\omega$-forcing from upper levels indicates that the redistribution
of air parcels in the vicinity of cyclone 1a, which leads to the observed APE tendencies, is a result of the circulation at upper-
levels. In particular, the loss of APE along the western flank of the trough is closely connected to the ageostrophic circulation
induced by this trough. Therefore, the depletion of the APE reservoir shown in Fig. 3 does not take place directly within
cyclone 1a, but at the western flank of the trough.

### 3.3 The role of air-sea interaction

Many air parcels which start in the APE reservoir and lose APE as they descend eventually arrive in the lower troposphere,
where they experience diabatic warming due to the interaction with the warm waters of the Gulf Stream SST front (Fig. 4a,b).
The air parcels loose APE through their descent, since they are colder than their reference state. Therefore, diabatic warming
of those air parcels should result in a destruction of APE, which decreases the amount of APE available for conversion to KE.
In order to test this hypothesis, we start 48 h backward trajectories within the boundary layer of the strong marine CAO on 20
Feb 12 UTC. Figure 6 shows that indeed the selected trajectories descend into the CAO region and are of continental origin.
The air parcels arriving within the CAO typically start their descent between 900 and 600 hPa within the poleward reservoir
of high-APE air. Thus, the air parcels feature a significant APE density 48 h before they arrive within the CAO, which is almost
entirely depleted upon their arrival in the CAO (Fig. 7a). On average, the air parcels descend by more than 100 hPa (Fig.
7b), which results in a negative adiabatic APE tendency (Fig. 7c). In contrast to the adiabatic tendency, the average diabatic
tendency of APE becomes negative only 12 h before arrival in the CAO when the air parcels start to be exposed to diabatic
heating via surface sensible heat fluxes and latent heating. Moreover, the magnitude of the diabatic tendency is a factor four



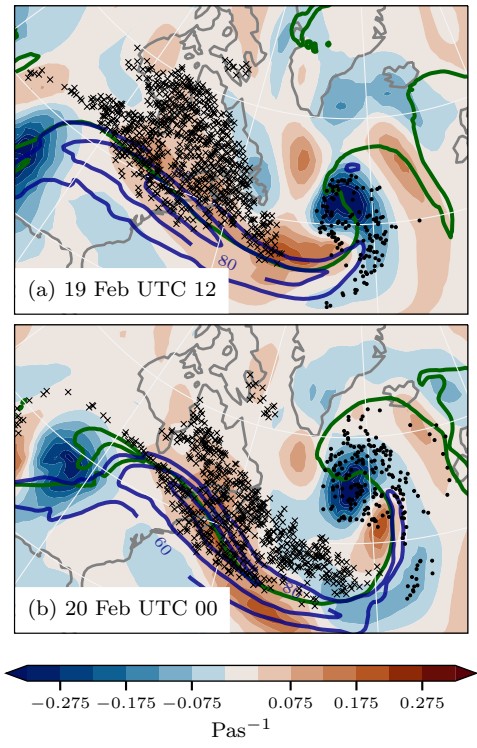

**Figure 5.** Upper-level quasi-geostrophic $\omega$ on 500 hPa (color shading) for two time steps. Descending (x) and ascending (·) trajectory positions for every 10th trajectory are marked in black as the reach the layer between 400 and 600 hPa. Also shown are the 2-PVU contour on the 315 K isentrope (green) and wind speed on 300 hPa (blue contours; 60, 80 and 100 ms$^{-1}$).

smaller than the adiabatic tendency (Fig. 7d). Even though the warming of air parcels within the CAO might be strong, the air parcels are already close to their reference state because of the adiabatic descent prior to their arrival in the CAO. Therefore, the air parcels have a low APE density and a low thermal efficiency. Since the diabatic tendency is given by the product of the diabatic heating rate and the thermal efficiency, even large heating rates do not result in a large diabatic APE tendency. Thus,
locally, the high APE of the upper tropospheric polar cold air is primarily converted to KE rather than depleted by diabatic air mass transformations.



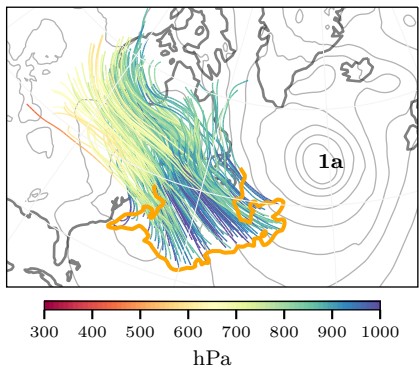

**Figure 6.** 48 h backward trajectories (colored by pressure; every 10th trajectory) initialized in the CAO area south of 50°N within the boundary layer on 20 Feb 12 UTC. The 8 K contour of the CAO index south of 50°N is shown in orange. Also shown is the mean sea-level pressure (grey contours; every 10 hPa).

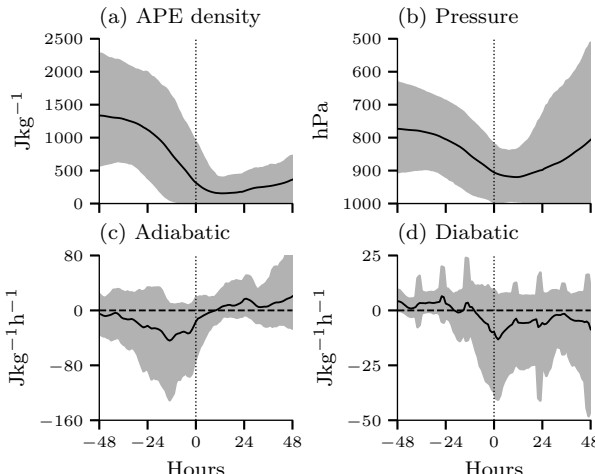

**Figure 7.** Evolution of (a) APE density, (b) pressure, (c) adiabatic APE tendency and (d) diabatic APE tendency of CAO trajectories started on 20 Feb 12 UTC, centered on their arrival in the CAO region (time zero). Note the different scales of the tendency in (c) and (d). The solid black line indicates the mean, and the grey shading denotes the range between the 5th and 95th percentile of the trajectories.



## 3.4 Key characteristics of APE on synoptic scales

Before turning our attention to the climatological analyses, we aim here to briefly summarize our findings from the case study. In light of questions 1–3 posed in the introduction, we can draw the following conclusions regarding local APE during a period of rapid North Atlantic cyclogenesis:

– Cyclones within the case study period develop at the edge of the polar APE reservoir. The intensification of the cyclones is associated with an extension of this APE reservoir into the North Atlantic in the wake of the cyclones. This APE is located in the mid to upper troposphere, below the upper-level trough to the west of the surface cyclone.

– The vertical circulation induced by the intensifying cyclone and upper-level trough lead to both APE consumption and production. APE is primarily consumed by the descent along the western flank of the trough, while APE is produced by the ascent ahead of the trough. This pattern of ascent and descent is consistent with QG$\omega$ forcing by upper levels. Consequently, the principal drivers of significant APE tendencies across the troposphere are not directly attributable to surface cyclonic activity, but rather to the dynamics of the upper-level atmospheric circulation.

– The diabatic dissipation of APE induced by surface fluxes is predominantly confined to the lower troposphere. By the time air parcels have reached the lower troposphere, the bulk of APE has already been consumed adiabatically. Therefore, the impact of surface fluxes on the local APE budget is small compared to the previous conversion into KE via subsidence due to low thermal efficiency.

In the next section, we will explore the validity of those findings for deep North Atlantic cyclones in general.



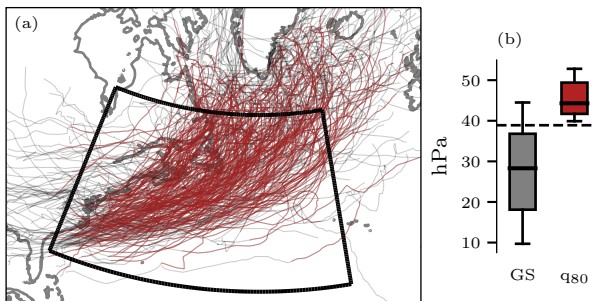

**Figure 8.** (a) Selected cyclone tracks within the Gulf Stream region, indicated by the black box. The section of the tracks between 12 h before maximum intensification and 12 h after maximum depth is highlighted in red. (b) Distribution of maximum depth for cyclones attaining maximum intensification within the Gulf Stream region for all cyclones (grey) and the 20% deepest cyclones (red). Whiskers show the 10th-90th percentile range. The black dashed line indicates the 80-percentile of the climatological maximum depth of Gulf Stream cyclones, which was used as a selection criterion.

## 4 Climatology

For the climatological analysis, our focus is on strongly intensifying Gulf Stream cyclones. First, we identify all cyclones in the study period (DJF) with a lifetime of at least 24 h which reach their maximum intensification rate within 24 h in the defined Gulf Stream region, bounded by 30°N to 55°N latitude and 80°W to 30°W longitude. Subsequently, from this set of cyclones, we select the 20% deepest ones, where the depth is defined as the difference in mean sea-level pressure between the outermost closed contour and the pressure minimum (Fig. 8b). This results in a total of 285 cyclone tracks, as illustrated in Fig. 8a.

### 4.1 Cyclone-centered composites

#### 4.1.1 Cyclone life cycle

Figure 9 depicts cyclone-centered composites of Gulf Stream cyclones for various stages of their life cycle, anchored on the time of maximum intensification ($\Delta p_{max}$) and maximum depth ($d_{max}$). We define the time 12 h before $\Delta p_{max}$ is reached as the onset stage, and the time 12 h after $d_{max}$ is reached as the decay stage. Analysis of average fields of isentropic PV at 315 K

and horizontal wind speed at 300 hPa reveals distinctive characteristics at these different stages of the cyclone life cycle.

At onset, the cyclone center is located ahead of a trough, near the right entrance of the jet stream (Fig. 9a). By the time the cyclone reaches $\Delta p_{max}$, the trough has intensified, and the cyclone has propagated towards the left exit region of the jet (Fig. 9b). Upon reaching $d_{max}$, the cyclone is located at the left exit of the jet, accompanied by upper-level Rossby wave breaking, yielding a ridge downstream of the cyclone (Fig. 9c). 12 h later, the cyclone enters the decay phase, characterized by

its displacement from the eastern flank of the trough and a weakening of the jet aloft (Fig. 9d).





### 4.1.2 Horizontal APE evolution

The cyclone typically is found at the southern boundary of the APE reservoir at the onset stage (Fig. 9e). The vertically integrated net APE tendency shows a weak dipole pattern with APE production near the cyclone center and consumption upstream. As the cyclone intensifies, the APE reservoir extends farther south relative to the cyclone center, accompanied by an

amplification of the dipole in the APE tendency (Fig. 9f). At $d_{max}$ the positive APE tendency is clearly more intense than the negative APE tendency (Fig. 9g). Furthermore, the negative APE tendency extends downstream of the cyclone. A comparison with the PV field at $d_{max}$ reveals that the cyclone is part of a dispersive baroclinic wave, manifesting in the formation of another trough downstream of the cyclone. Consequently, the negative APE tendency downstream is indirectly linked to the cyclone via downstream development. During the decay of the cyclone, the APE tendencies weaken significantly (Fig. 9h).

Throughout the cyclone's life cycle, the evolution of vertically integrated APE closely follows the development of the trough, reinforcing findings from the case study that underscore the close connection between the upper-level large-scale circulation and the APE distribution. Additionally, our analysis indicates a consistent pattern of APE consumption at the western flank of the trough and APE production at the eastern flank of the trough.

Next, we decompose the net APE tendency into its adiabatic and diabatic components and investigate the atmospheric
conditions driving these tendencies. For all stages of the cyclone life cycle, the adiabatic APE tendency shows a pattern and magnitude closely resembling that of the net APE tendency. This reaffirms findings from the case study: the dominant role of the adiabatic contribution in the APE budget, suggesting that the net change of APE predominantly stems from vertical motion. This is also in line with upper-level QG$\omega$ forcing. In fact, during the onset phase, the upper-level QG$\omega$ forcing at 500 hPa indicates forcing for ascent ahead of the trough and a forcing for descent behind it, aligning closely with the adiabatic
APE tendency (Fig. 9i). As the cyclone reaches $\Delta p_{max}$, there is an increase in the magnitude of upper-level QG$\omega$ forcing, reflected in substantial adiabatic APE tendencies (Fig. 9j). At $d_{max}$ the weaker negative adiabatic APE tendency compared to the positive APE tendency is consistent with the upper-level QG$\omega$ forcing (Fig. 9k). 12 h later, the decay of the surface cyclone coincides with a decrease in the upper-level QG$\omega$ forcing (Fig. 9l). Notably, the upward forcing is displaced from the cyclone center, indicating a weakened coupling between upper and lower levels, which also translates to smaller adiabatic
APE tendencies. The close association between the upper-level QG$\omega$ forcing and the adiabatic APE tendency corroborates the findings of the case study, suggesting that APE changes are primarily driven by upper-level dynamics.

Finally, the diabatic APE tendency exhibits a north-south dipole background associated with radiative cooling of the free troposphere, wherein APE is generated toward the poles and dissipated toward the equator (Fig. 9m). Superposed on this background pattern is a pronounced negative diabatic APE tendency in the cyclone's warm sector (Fig. 9n-p). Notably, this
negative diabatic APE tendency coincides with intense precipitation, wherein latent heat release warms the air colder than its reference state, consequently leading to APE destruction. Also, this finding is consistent with observations from the case study.



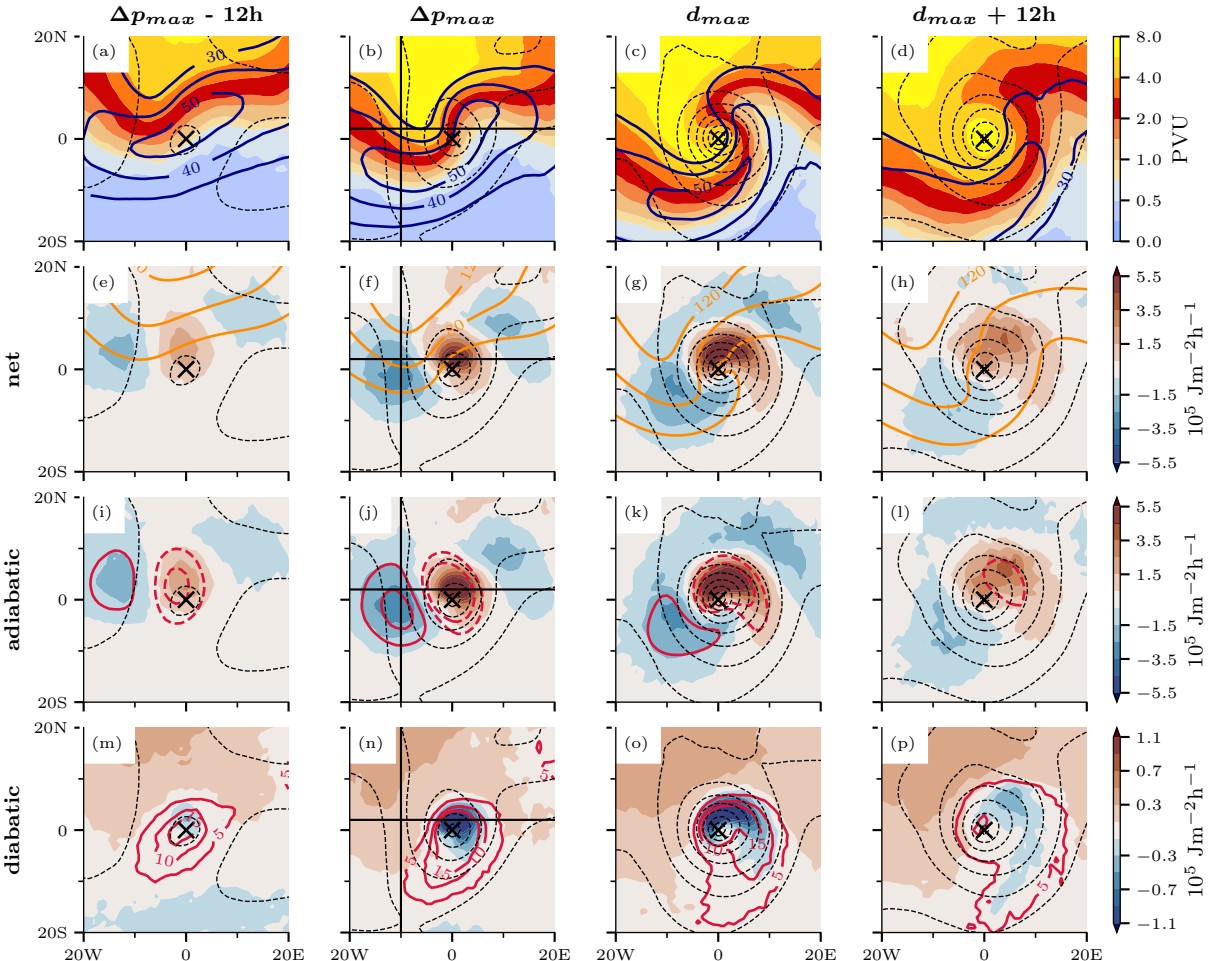

**Figure 9.** Cyclone-relative composites 12 h before time of maximum intensification ($\Delta p_{max}$ - 12 h), at time of maximum intensification ($\Delta p_{max}$), at time of maximum depth ($d_{max}$) and 12 h after time of maximum depth ($d_{max}$ + 12 h). Black dashed contours indicate mean sea-level pressure (every 8 hPa). (a)-(d) isentropic PV at 315 K (shading) and wind speed on 300 hPa (every 10 ms$^{-1}$). (e)-(h) net APE tendency (shading) and the distribution of vertically integrated APE density (orange contours; 40, 80, 120 · 10$^5$ Jm$^{-2}$). (i)-(l) adiabatic APE tendency (shading) and upper-level QG$\omega$ on 500 hPa (red contours; -0.1, -0.05, 0.05, 0.1 Pas$^{-1}$). (m)-(p) diabatic APE tendency (shading) and one-hour accumulated precipitation (red contours; every 5 mm). Note the different color scales for the APE tendencies. Black lines indicate the location of cross sections in Figure 11.

### 4.1.3 Evolution of the net APE tendency

As the cyclone intensifies the relative magnitude of the positive and negative local contributions to the net APE tendency changes (Fig. 9e-h). To quantify this evolution, we compute the volume-integral of the net tendency for each cyclone within a 2000 km radius around the cyclone center (which corresponds to a radius of approximately 20°in Fig. 9) and compare it with





the intensification of the cyclone. Figure 10a shows the distribution of the mean sea-level pressure minimum ($p_{min}$) of the selected cyclones during the four stages of their life cycle. By design, the strongest deepening takes place around the time of $\Delta p_{max}$ until the maximum depth $d_{max}$ is reached. Between the time of $d_{max}$ and $d_{max} + 12$ h the minimum pressure increases, which indicates the decay of the surface cyclone and a weakening of the circulation.

The integrated net APE tendency, averaged over a 12 h window centered on the selected life cycle stages, is depicted in Fig. 10b. Between the onset stage and the time of $\Delta p_{max}$, the net tendency becomes more negative, indicating that the conversion of APE to KE intensifies. At the time of $\Delta p_{max}$, the net tendency reaches a minimum, which is consistent with the observation that this is the stage in which the strongest intensification of the cyclone occurs. Subsequently, the net tendency becomes more positive as the intensification weakens prior to the time of $d_{max}$, which marks the beginning of the decay phase of the

cyclone. This is reflected in a sharp increase of the net tendency, which becomes positive 12 h after $d_{max}$. At this stage, the net conversion is from KE back to APE. Hence, the intensification of the surface cyclone is reflected in the life cycle of the net APE tendency in the vicinity of the cyclone.

    Consistent with the horizontal evolution of the APE tendency, the integrated net APE tendency is largely adiabatic (Fig. 10c). This indicates that the net APE tendency is determined by the balance between the negative adiabatic APE tendency due

to descent along the western flank of the trough and the positive tendency due to ascent along the eastern flank of the trough. As the cyclone intensifies, more APE is converted to KE by descent than KE is converted back to APE by ascent, resulting in a net increase of KE. However, once the cyclone has reached its maximum intensity ($d_{max}$) and begins to decay, this balance shifts in favor of the positive tendency due to ascent, leading to a net decrease in KE.

    The life cycle is somewhat less apparent in the adiabatic tendency compared to the net tendency. This discrepancy can

be attributed to the diabatic contribution (Fig. 10d), which reaches a minimum at the time of $\Delta p_{max}$, yet stays net positive throughout the cyclone's life cycle. This accentuates the evolution observed for the adiabatic tendency. The diabatic tendency is positive due to the creation of APE by radiative cooling of the free troposphere on the cold side of the baroclinic zone, which is largely independent of cyclogenesis. Consequently, the reduction in the diabatic tendency as the cyclone intensifies is attributable to the latent heat release in the ascending air stream of the cyclone, which dissipates APE as it warms air that is

colder than its reference state.

    Despite the clear life cycle being recovered from the integrated net APE tendency, the spread is wide. This is due to the fact that APE conversion is closely linked to the large-scale circulation, as discussed in the previous section. Therefore, the extraction of KE does not have to occur in the immediate vicinity of the surface cyclone. For instance, it can occur at the entrance of the jet stream, which can be located at a considerable distance from the cyclone if the cyclone intensifies at the jet

exit (cf. Fig. 9b). In contrast, APE production through the ascending air stream occurs in close proximity to the center of the cyclone.

### 4.1.4   Vertical APE structure

The vertical structure of the cyclone-relative composites is illustrated in Fig. 11 through both meridional and zonal cross sections at the time of maximum intensification. The horizontal location of the cross sections is indicated in Fig. 9. The



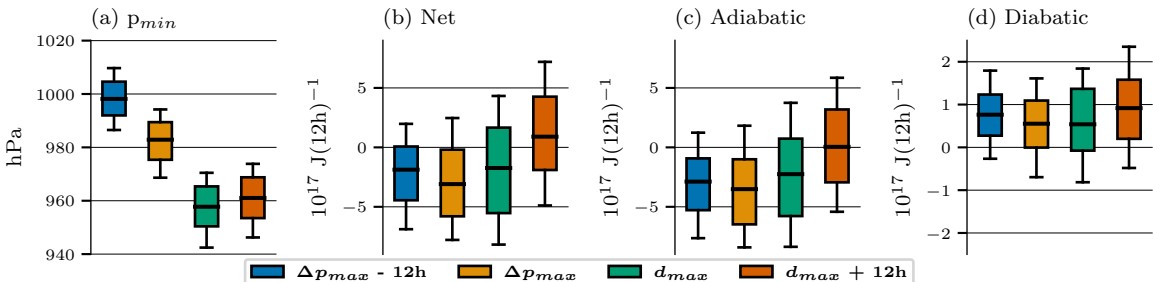

**Figure 10.** (a) Distribution of minimum mean sea-level pressure of the 285 selected cyclones, averaged +/- 6 h around the four life cycle stages. The (b) net, (c) adiabatic and (d) diabatic APE tendency is integrated in a 2000 km radius around the cyclone center within a +/- 6 h window around the four life cycle stages. Whiskers show the 10th-90th percentile range.

meridional cross section shows the surface cyclone slightly poleward of the baroclinic zone and the upper-level jet stream, because the cross section cuts through the cold sector of the cyclone (Fig. 11a). Also the height of the tropopause, marked by the 2-PVU line, is constant poleward of the baroclinic zone but increase steeply at the baroclinic zone. The APE density is concentrated on the poleward side of the cyclone below the tropopause, where the air is much colder than its reference state (Fig. 11b). The APE density decreases toward the surface, reaching its minimum in the lower troposphere. The negative

adiabatic APE tendency spans from 850 to 250 hPa, peaking around 500 hPa. Conversely, the diabatic APE tendency is positive throughout much of the free troposphere north of the baroclinic zone, contributing to the maintenance of the poleward APE reservoir (Fig. 11c).

   Examining the zonal cross section of the PV field reveals the presence of the upper-level trough upstream of the cyclone center, alongside jet cores flanking it (Fig. 11d). Below the trough, isolines of potential temperature are bent upward, indicative

of a cold anomaly beneath the trough, where APE density is maximal (Fig. 11e). Analogous to the positive adiabatic APE tendency, the negative diabatic APE tendency extends through most of the column, peaking around 500 hPa. Given the negative adiabatic APE tendency within regions of high APE density and positive APE tendency downstream in the area of low APE density, the dipole of APE tendency leads to an eastward propagation of the APE density maximum. Furthermore, the diabatic APE tendency is most pronounced where the largest positive adiabatic APE tendency is observed (Fig. 11f) such that the

diabatic APE tendency partially offsets the adiabatic APE tendency. In fact, the ascent leading to the negative adiabatic APE tendency also leads to considerable latent heat release, which translates to the negative diabatic APE tendency.




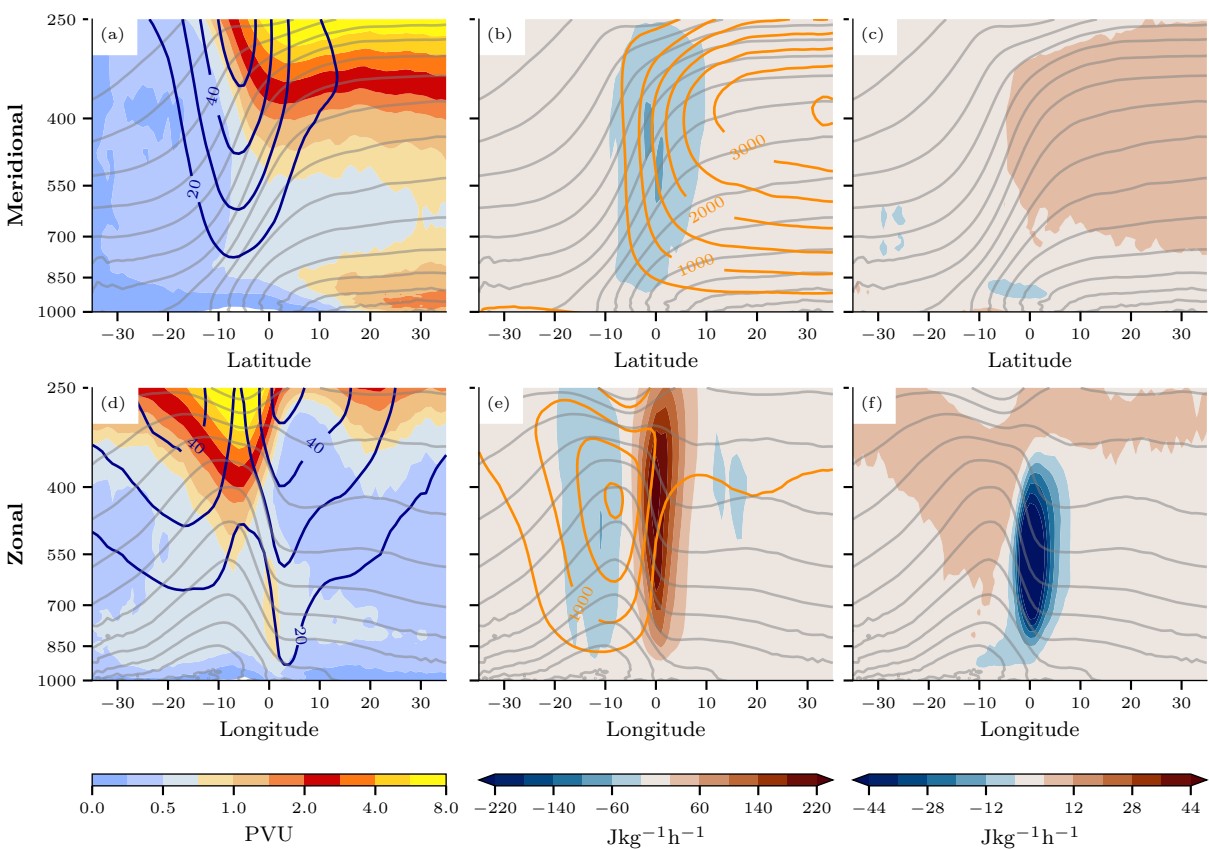

**Figure 11.** Cross sections of cyclone-centered composites at the time of maximum intensification. (a)-(c) are meridional cross sections at $10°$W (ca. 1000 km west) and (d)-(f) are zonal cross sections at $2°$N (ca. 200 km north) relative to the cyclone center. (a),(d) PV (shading) and horizontal wind speed (blue; every $10\,\mathrm{ms}^{-1}$). (b),(e) adiabatic APE tendency (shading) and APE density (orange contours; every $500\,\mathrm{Jkg}^{-1}$). (c),(f) diabatic APE tendency. Potential temperature is indicated by grey contours (every 6 K). The locations of the cross sections are indicated in Fig. 9.



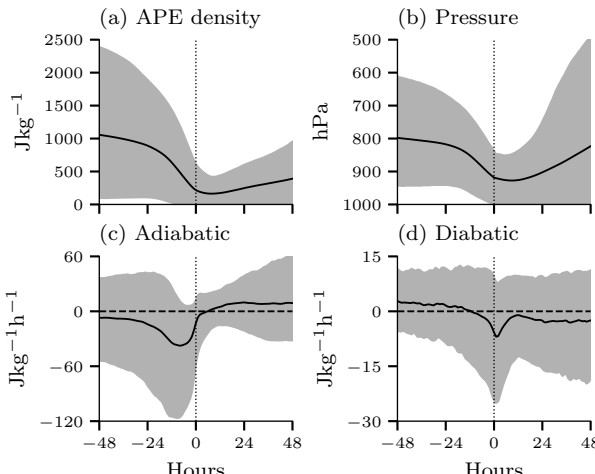

**Figure 12.** Evolution of (a) APE density, (b) pressure, (c) adiabatic APE tendency and (d) diabatic APE tendency of CAO trajectories, centered on their arrival in the CAO region (time zero). The solid black line indicates the mean, and the grey shading denotes the range between the 5th and 95th percentile of the trajectories. For details about the trajectory calculation, see text.

## 4.2 Cold air outbreak composite

To clarify the climatological role of surface fluxes along the Gulf Stream front in the APE budget, we focus on marine CAOs associated with Gulf Stream cyclones, which are typically accompanied by substantial latent and sensible heat fluxes. We
identify CAOs as areas where the CAO index exceeds 8 K and that lie within 30°N - 50°N latitude and 130°W - 30°W longitude at the time of maximum intensification of the cyclones. From within these identified regions, we initiate 2-day forward and 5-day backward trajectories at each grid point between 1000 hPa and the boundary layer height in 25 hPa intervals. The mean trajectory evolution, centered on the first time step when trajectories reach a CAO index surpassing 8 K, is depicted in Figure 12.

The mean APE density of the air parcels exhibits a continuous decrease over the 48 h period preceding their arrival in the CAO, reaching a minimum approximately 6 h later (Fig. 12a). Concurrently, the mean pressure along trajectories indicates that this APE decrease coincides with a descent of approximately 100 hPa (Fig. 12b), reflected in the negative adiabatic APE tendency prior to arrival in the CAO (Fig. 12c). In contrast, the diabatic APE tendency remains slightly positive and only transitions to negative values 12 h before the air parcels arrive in the CAO (Fig. 12d). By the time the negative diabatic
APE tendency peaks, the majority of APE has already been depleted by the descent of air parcels into the CAO. Thus, the climatological analysis confirms the observation from the case study that surface fluxes act as a sink of APE, albeit their magnitude is small compared to the adiabatic conversion of APE to KE.





## 5  Conclusions

### 5.1  Synthesis

In this study, we have conducted an analysis of the spatial distribution and temporal tendencies of local APE density within a dataset comprising 285 intense North Atlantic cyclones that undergo intensification in the Gulf Stream region. Our investigation has focused on elucidating the interplay between local APE and the large-scale atmospheric circulation, while also evaluating the impact of the intense air-sea interaction in the Gulf Stream region on the local APE budget. The principal findings of our analysis can be summarized as follows:

– Intense Gulf Stream cyclones typically originate at the edge of the polar APE reservoir. As these cyclones develop, the concomitant formation of an upper-level trough induces a cold anomaly in the mid to upper troposphere beneath it. This cold anomaly serves as a reservoir of high-APE air, which is transported from polar latitudes to midlatitudes, and provides the energy for baroclinic conversion.

       – While APE is predominantly consumed adiabatically through descent along the western flank of the upper-level trough,
it is concurrently generated by ascent ahead of the trough. This dipole of adiabatic APE tendencies aligns closely with the ageostrophic vertical circulation induced by the trough, thereby highlighting the intimate connection between the local APE budget and the upper-tropospheric large-scale atmospheric circulation. Furthermore, the release of latent heat in the ascending air stream in the cyclone's warm sector leads to a negative diabatic tendency, resulting in the destruction of APE. Consequently, latent heat release acts as a sink for APE.

– Our Lagrangian trajectory analysis of the influence of air-sea interaction on the local APE budget reveals that the bulk of APE is converted to KE during descent prior to air parcels experiencing surface fluxes within CAOs. Therefore, comparatively little APE is dissipated by surface sensible heat fluxes in the lower troposphere.

       In conclusion, our study underscores the significant influence of synoptic-scale circulation patterns on the distribution of local APE. We observe complex patterns of APE tendency, characterized by both positive and negative contributions associated
with the ageostrophic circulation induced by large-scale flow features such as upper-level troughs. Thus, while surface cyclones are a manifestation of baroclinic waves, the conversion of APE occurs predominantly aloft between troughs and ridges.

### 5.2  Discussion

A novelty of our work is the coupling of global energetics with the large-scale circulation and PV thinking (Hoskins et al., 1985). We show that, locally, midlatitude APE is found in the cold anomaly below an upper-level trough. Therefore, isentropic
lifting below upper-level troughs, termed the "vacuum cleaner" effect in Hoskins et al. (1985), links the distribution of APE and PV. The distribution of PV is also linked to the adiabatic APE tendency via the $\omega$-equation (Hoskins et al., 2003). As a midlatitude PV anomaly propagates eastward in a baroclinic zone, it leads to descent at the western flank and ascent at the eastern flank of the anomaly, which corresponds to a negative APE tendency west of the anomaly and a positive APE tendency





east of the anomaly. Therefore, APE is converted to KE upstream of the anomaly, where the flow is accelerated. In turn, the
deceleration of the flow downstream of the anomaly means that KE is converted back to APE, effectively propagating the
maximum of APE below the upper-level trough. From that perspective, the upper-level flow controls the distribution of APE.

Our study differs substantially from previous works on the atmospheric energy cycle in that we consider the total APE
rather than a decomposition into mean and eddy APE. While a decomposition into mean and eddy contributions facilitates the
interpretation of climatological APE conversions, these conversions cannot readily be attributed to individual weather systems
on synoptic charts. Nevertheless, we recover some fundamental patterns in the climatology of APE tendencies shown in Novak
and Tailleux (2018) within our cyclone-relative perspective. They identify a hot spot of adiabatic conversion of eddy APE to
eddy KE at the upstream edge of the North Atlantic storm track, where the large-scale flow is generally accelerated and the
entrance of the climatological jet stream is found (Koch et al., 2006). This is consistent with our finding that the adiabatic APE
tendency in cyclones is negative along the western flank of the upper-level trough, where the large-scale flow is accelerated.
Novak and Tailleux (2018) also find that the conversion from mean KE to mean APE is largest in the downstream part of
the North Atlantic storm track, where the exit of the climatological jet stream is located (Koch et al., 2006). We show that the
positive adiabatic tendency is greatest in the decay phase of cyclones ahead of the upper-level trough, where the large-scale flow
is decelerated. Thus, our results show how individual cyclones can contribute to the climatological patterns of APE conversion.

A consequence of abandoning the eddy-mean decomposition is that the ascending air stream near the cyclone center converts
KE back into APE, which is counterintuitive given that the rising air stream originates in a local warm anomaly. However,
this result is consistent with the local APE framework, because the global reference state is similar to a spatial average of
potential temperature on pressure surfaces. Thus, the ascent occurs within negative adiabatic efficiency with respect to the
global reference state. In contrast, the eddy-mean decomposition introduces a temporal average of the reference state, which
allows a physically more localized interpretation of the conversion of eddy APE to eddy KE, because it accounts for high-
frequency warm and cold anomalies introduced by the meridional circulation around a low pressure system. However, it is
difficult to attribute eddy and mean conversions to individual weather systems.

Novak and Tailleux (2018) have also shown that in the North Atlantic, the diabatic mean and eddy APE tendencies are largest
along the storm track. However, the mean component destroys APE while the eddy component generates APE, obscuring the
net effect on the total APE. We show that the total diabatic APE tendency due to latent heat release in the warm sector of
cyclones is negative, because the latent heat release occurs in air which is colder than its reference state. Since the local APE
density is derived from the dry hydrostatic primitive equations, the feedback of diabatic heating on the vertical velocity (Nie
and Sobel, 2016; Li and O'Gorman, 2020) is not explicitly captured. Therefore, we hypothesize that the negative diabatic APE
tendency due to latent heat release is smaller than the enhancement of the adiabatic APE tendency due to stronger updrafts,
which would reconcile our result with the fact that latent heat release generally accelerates cyclone development (Schemm
et al., 2013; Binder et al., 2016). This result also follows from moist APE (MAPE) frameworks, which include the potential
of latent heat release in APE (Lorenz, 1979; Gertler and O'Gorman, 2019; Gertler et al., 2023). However, a moist atmosphere
may have several local minima of potential energy, which makes the definition of a unique APE density difficult (Tailleux,
2013).





Even for the dry APE density, the choice of a reference state is not straightforward (Shepherd, 1993; Tailleux, 2018). Federer
et al. (2024) computed a reference state for a baroclinic channel, where the baroclinic zone is approximately in the center of
the channel. Therefore, the APE reservoirs on the poleward and equatorward sides of the baroclinic zone have similar APE
densities and are equally important as APE source regions for the development of the baroclinic wave. In contrast, the Earth's
spherical geometry means that there is more mass on the warm side than on the cold side of the midlatitude baroclinic zone.
Consequently, the Lorenz reference state constructed from the Earth's atmosphere yields a high APE density on the cold side
and a low APE density on the warm side of the baroclinic zone. This explains why we identify the equatorward descent of cold
air as the main extraction mechanism of KE, whereas in the baroclinic channel both ascent and descent convert APE into KE
with similar magnitudes (Federer et al., 2024).

Finally, our analysis reveals the influence of surface sensible heat fluxes within marine CAOs in the wake of intense Gulf
Stream cyclones on the local APE budget. Our work is in agreement with previous studies that identified surface sensible heat
fluxes as a sink for APE (Swanson and Pierrehumbert, 1997; Marcheggiani and Ambaum, 2020). However, we conclude that
this effect is small compared to the conversion of APE into KE in cold air streams descending from the polar APE reservoir.
Due to the adiabatic conversion during descent, the cold air in the lower troposphere is already close to its reference state
when it arrives in the lower troposphere and there is very little APE. Therefore, surface sensible heat fluxes cannot exert a
large influence on the APE reservoir. This result reconciles the fact that air-sea interaction has been identified as both a source
of energy for the midlatitude circulation (Papritz and Spengler, 2015; Wenta et al., 2024) and a sink for APE, because the
maintenance of low-level baroclinicity and moisture is vertically separated from the conversion of APE into KE.

### 5.3   Final remarks

Our study revealed that local APE is mainly concentrated in the middle and upper troposphere and that the large-scale circu-
lation governs its distribution. However, the physical link between upper-tropospheric APE and surface baroclinicity remains
unclear. APE formally quantifies the potential for baroclinic growth. Surface baroclinicity, as measured by the Eady growth rate
(Eady, 1949), quantifies the maximum growth rate of a baroclinic disturbance. Nevertheless, surface baroclinicity is sometimes
equated with the potential for cyclone development. Our findings show that the APE reservoir is not directly associated with
surface baroclinicity, because APE is concentrated in the middle and upper troposphere. Therefore, our study encourages fur-
ther investigation into the connection between surface baroclinicity and local APE. Specifically, it is unclear whether instances
of high surface baroclinicity typically coincide with the advection of significant amounts of APE into storm tracks, or if the
growth of cyclones within a baroclinic zone is limited by the synoptic-scale availability of APE.

The relationship between surface baroclinicity and cyclone development can be described using a predator-prey model,
where surface baroclinicity gradually increases before a sudden rise in heat flux activity, leading to a collapse in surface baro-
clinicity (Ambaum and Novak, 2014; Novak et al., 2017). Future research could explore the predator-prey dynamics of storm
tracks also from the perspective of local APE. We expect such a complimentary approach, based on APE and baroclinicity, to
provide deeper insights into the mechanisms maintaining favorable conditions for cyclone development along the storm tracks.



*Code and data availability.* The datasets are referenced in Sect. 2, with ERA5 data available on the Copernicus Climate Change Service (C3S) Climate Data Store: https://doi.org/10.24381/cds.adbb2d47 (Hersbach et al., 2023). Other code and data from this study can be obtained from the authors upon request.

*Author contributions.* MF, LP, MS and CMG planned and designed the study. MF analyzed the data and wrote the manuscript. LP, MS and CMG gave important guidance during the project and provided feedback on the manuscript.

*Competing interests.* At least one of the (co-)authors is a member of the editorial board of Weather and Climate Dynamics.

*Acknowledgements.* We gratefully acknowledge the European Centre for Medium-Range Weather Forecasts (ECMWF) for providing the
ERA5 reanalysis dataset. We also thank Marta Wenta (KIT) and the members of the Atmospheric Dynamics group at ETH Zürich, especially Heini Wernli, for fruitful discussions. The contribution of MF is funded by the Swiss National Science Foundation (SNSF; Grant 200021E_196978) as part of the Swiss–German collaborative project "The role of coherent air streams in shaping the Gulf Stream's impact on the large scale extratropical circulation (GULFimpact)." The contribution of CMG is funded by the Helmholtz Association as part of the Young Investigator Group "Sub-seasonal Predictability: Understanding the Role of Diabatic Outflow" (SPREADOUT; Grant VH-NG-1243).



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
