# Peer review of "Synoptic perspective on the conversion and maintenance of local available potential energy in extratropical cyclones"

_EGUsphere, 2024_

## Referee Comment (RC1)

**Review of WCD-2024-2112: Synoptic perspective on the conversion and maintenance of local available potential energy in extratropical cyclones**

**Overview:** This manuscript explored a relatively new technique of quantifying local available potential energy (APE) and applying it to cyclone dynamics for strong extratropical winter cyclones in the North Atlantic basin. Their results show a close connection between the extratropical tropospheric baroclinic zone ("polar front") and a strong gradient in local APE. They also assessed the movement of air parcels using trajectory analysis to explore the tendencies in APE, demonstrating strong APE to KE conversion upstream of a trough axis (advecting into the trough base) and strong KE to APE conversion downstream of a trough axis (ascending generally through the warm conveyor belt). I commend the authors on a well written manuscript that was generally easy to follow. I do have some suggestions for the clarity of the presentation in several places, some changes/new figures that I believe would help aid a reader in interpreting the results, and questions about the sensitivity of some of the analysis to choices in methodology. Thus, I believe this is on the border between a major and minor revision.

**General Comments:**

1. *Description of local APE and reference state:* I appreciate the challenges of word limits for manuscripts and the necessity of referring readers to prior literature for specifics on a methodology. That said, I would suggest that the authors add a bit more detail/clarity in sections 2.2 and 2.3 to help the readers understand the method a bit more clearly given the relatively new methods being applied here. For example, I found the description of how the reference state was computed (lines 143-145) challenging to understand/visualize, thus making interpretation of the APE figures later in the paper challenging in turn. It was unclear to me at times whether we were comparing a specific parcel to its specific reference state, or a column, or otherwise. Lastly – it may be beneficial up-front in this manuscript (rather than at the end) to discuss local APE in light of other metrics used for assessing energy for storms (eg. Eady growth rate/baroclinicity) and resistance to vertical velocity (eg. static stability).

2. *Discussion of APE changes linked to trajectories :* I appreciated the approach taken here by tracking parcels through the troposphere. However, I found the discussion surrounding figure 3 challenging. I could understand (eventually) how the method discussed in line 216-217 was applied to the analysis, but it made the interpretation really challenging. I can see the idea behind wanting to see how much APE is depleted from the starting location, but it made a) the interpretation of what the actual change in column APE was at a location really difficult, and b) made comparison to figure 9 really challenging. A more clear motivation here may be helpful, as well as clarification here (or at figure 9) about the differences in what we're looking at. One other approach could be to actually show either APE or dAPE/dt for your trajectories (akin to how you have pressure shown in figure 4). Further through this section, it would be helpful to really quantify the contribution due to omega/diabatic heating relative to the respective efficiency terms. Perhaps a box-and-whisker plot of distribution of the efficiency terms for the trajectories at different levels would help? There is good qualitative discussion here (see lines 264-266), but a bit more time spent on the quantitative side would really help. Lastly, your line 266-268 doesn't quite make sense to me r.e. linking the latent heat release to the second max of APE loss over the Canadian Arctic.

3. *Dynamic interpretation/explanation of results:* I appreciated that there were flavors of QG forcing, jet ageostrophic circulations, and PV thinking infused throughout the paper. That said, I think there were a few points where the discussion could've been taken into more depth and/or the QG/ageostrophic/PV discussions could've been more unified (eg. it's three perspectives to explain rising/sinking motion). For example, there was a lot of discussion on jet entrance/exit regions (which were at times not clear to me as entrance/exits rather than poleward flanks of the jet), but much less discussion on the role of curvature (where both the trough and upstream ridge can instigate ageostrophic circulations resulting in +/- omega), transverse thermally (in)direct circulations along jet streaks, and whether there was temperature advection occurring across the jet (which shifts where we see +/- omega in response to ageostrophic responses). Much of this could also be discussed from the perspective of PV theory (which emerges late in the paper but could've been interwoven throughout).

4. *Inclusion of CAOs:* Though I found the inclusion of the CAO case and composite interesting, I wasn't really sure how much it contributed to the paper. This may benefit from either removing it, or better motivating why you're including it.

**Specific comments:**
- Lines 53-54: Please clarify/elaborate on the 'why' of APE being in the polar middle and upper troposphere and how it is advected (eg. is it a material quantity?).
- Lines 61-62: For local APE, is it only accelerations/decelerations? For example, do all air parcels in the warm conveyor belt that contribute to a positive APE tendency explicitly have a deceleration occurring?
- Lines 110-112: Make two sentences – it's a bit hard to follow as written.
- Section 2.2: Please include units throughout here. It may be helpful to also provide equations for specific density and your diabatic heating rate.
- Line 170: From your figure 1, it appears less that an upper level trough is developing rather than propagating in from the west. Please be sure that it is developing (rather than propagating) if you use the term 'upper-level trough forms'.
- Lines 171-172: I think you need to spell out more how the collocation of high APE with the DT is demonstrating the connection to the large scale circulation (I don't disagree with you, but it needs to be more clearly demonstrated).
- Lines 179-180: Do we know that this is low APE advection vs. a time tendency due to conversion? In theory, one could compute APE advection to show this explicitly. The same goes for lines 183-184.
- Lines 202-204: This isn't really a stand-alone paragraph – please aim to elaborate or merge with another paragraph.
- Lines 206-207: The cross-section does show the cold dome and therefore a troposphere-deep cold anomaly. It does not, however, imply cold air advancing into the midlatitudes. Consider providing different evidence or re-writing.
- Figure 1 cross-section lines/Figure 2: Consider given start/end point markers for your cross-sections (eg. A-A', B-B', C-C') to avoid any ambiguity about the direction of your cross-sections.
- Lines 222-225: This may benefit from a schematic (this infuses with general comment 2 above).

- Line 227, 228 (and elsewhere): Careful on geolocation descriptions. For example – the Canadian Arctic is a pretty expansive area. But it appears your focus in figure 3 may be more on Hudson Bay/Quebec/Labrador? Likewise, I wouldn't consider the Gulf Stream to be adjacent to the Canadian Arctic. A reference geographic map (or outline of the Gulf Stream region) may be helpful.
- Lines 243-248: There's a large region of positive APE tendency south of Iceland that likely is playing a factor the in the cyclone evolution. It would be helpful to either look at trajectories/APE tendencies in that region as well, or to provide justification for not examining it.
- Lines 258-260: I would find it helpful to see some discussion of this also in reference to the efficiency terms.
- Line 280: I'm not sure the line 'detaches from the band of downward …' is the best choice of phrasing. Please reword.
- Line 283: I'm not sure I would classify this as the entrance region of the jet. It may be better discussed as downstream of the upper-tropospheric ridge.
- Lines 290-291: Though the trough may play a role here, so too does the upstream ridge (contributing to QG forcing for descent).
- Figure 5: I found it challenging to differentiate the dark green (2-PVU) contour from the black (300 hPa wind speed) contours. Please consider a slightly more contrasting color choice. I would also add a 'L' to represent your surface low position in both panels.
- Lines 322-323: I would be careful about making this statement. The surface cyclone can influence the upper-troposphere and vice versa (eg. cyclone growth positive feedback cycle), and your cross-section shows a troposphere-deep cyclonic cold anomaly and circulation.
- Figure 8: Aim to make figure 8b seems more its own independent panel (maybe a box around it?). I didn't notice it even when referenced in the text.
- Line 338: How many times steps overlap as being both in the delta-Pmax category and the dmax category?
- Line 341-342: Again, not sure this is the jet entrance region. I suspect curvature dynamics are playing more of a factor.
- Line 345: I think you can lean into PV thinking for the occlusion stage here (eg. evidence of a stacked cyclonic PV anomaly circulation from surface to dynamic tropopause, along with the surface low receding into the cold PV hook.
- Line 351-352 (and elsewhere in this section): Consider adding some values to the description here.
- Lines 352-353: Is this PV field shown anywhere? It may be beneficial to include.
- Lines 372-374; 399-405: Consider bringing in some discussion of the poleward movement of your composite when discussion the radiative cooling of the free atmosphere.
- Lines 379-380: How sensitive is your analysis to the region chosen here? I can understand 2000 km from a synoptic scale perspective, but does this strongly impact the interpretation of storm-scale contributions?
- Lines 415-416: Why not take your cross-section through the center of the surface cyclone? Here your baroclinic zone is south of the surface cyclone simply because the trough is further south to the west of the cyclone (as you noted).

- Lines 425-426: I found this sentence unclear. Which positive adiabatic APE tendency?
- Section 4.2: Please provide information on how many CAO cases were involved.
- Lines 455-456: I think you need to reverse the order here (eg. trough leads to surface cyclone, not vice versa).
- Lines 469-470: Your zonal cross section implies a troposphere-deep circulation, so this feels a bit misleading as written to me.
- Lines 485-492: You mentioned the Bowley et al. 2019 paper in the introduction – though their results are a bit hard to interpret relative to yours (given the global zonal APE vs. local total APE perspective), you may want to include some of their interpretation here. For example, they found a dominant mechanism for synoptic scale APE increase to be ascent on the poleward flank of the wave guide in the exit region of the North Pacific jet, which fits well to the results of Koch et al. 2006 and your adiabatic generation interpretation here.
- Line 517-520: I think your points here would be beneficial to appear earlier in the manuscript when introducing the local APE framework to help unify the global vs. local perspectives.

**Technical corrections:**
- Line 75: Change 'and to contribute' to 'and can contribute'.
- Line 98: I'm not sure 'relies' is the right word choice for your data here.
- Line 167: Please write out 'potential vorticity unit' the first time you define PVU.
- Line 343: Please add 'cyclonic' between 'upper-level' and 'Rossby wave breaking'
- Line 430: Should this read 'ascent leading to the positive adiabatic APE tendency'?

---

## Author Comment (AC1)

Final author comments for

**Synoptic perspective on the conversion and maintenance of local available potential energy in extratropical cyclones**

by Marc Federer, Lukas Papritz, Michael Sprenger, and Christian M. Grams

October 2, 2024

In the following, the comments of the reviewers are shown in **blue** and our replies in **black**. Mentioned references are listed at the end of the document.

**Review by Kevin A. Bowley**

Review of WCD-2024-2112: Synoptic perspective on the conversion and maintenance of local available potential energy in extratropical cyclones

Overview: This manuscript explored a relatively new technique of quantifying local available potential energy (APE) and applying it to cyclone dynamics for strong extratropical winter cyclones in the North Atlantic basin. Their results show a close connection between the extratropical tropospheric baroclinic zone ("polar front") and a strong gradient in local APE. They also assessed the movement of air parcels using trajectory analysis to explore the tendencies in APE, demonstrating strong APE to KE conversion upstream of a trough axis (advecting into the trough base) and strong KE to APE conversion downstream of a trough axis (ascending generally through the warm conveyor belt). I commend the authors on a well written manuscript that was generally easy to follow. I do have some suggestions for the clarity of the presentation in several places, some changes/new figures that I believe would help aid a reader in interpreting the results, and questions about the sensitivity of some of the analysis to choices in methodology. Thus, I believe this is on the border between a major and minor revision.

Thank you for your positive evaluation of our manuscript. We are grateful for your constructive and insightful feedback, and we highly appreciate the time you put into this review. We believe the suggested changes will significantly improve our manuscript and we hope the revised version will match your expectations.

General Comments:
1. Description of local APE and reference state: I appreciate the challenges of word limits for

manuscripts and the necessity of referring readers to prior literature for specifics on a methodology. That said, I would suggest that the authors add a bit more detail/clarity in sections 2.2 and 2.3 to help the readers understand the method a bit more clearly given the relatively new methods being applied here. For example, I found the description of how the reference state was computed (lines 143-145) challenging to understand/visualize, thus making interpretation of the APE figures later in the paper challenging in turn. It was unclear to me at times whether we were comparing a specific parcel to its specific reference state, or a column, or otherwise. Lastly – it may be beneficial up-front in this manuscript (rather than at the end) to discuss local APE in light of other metrics used for assessing energy for storms (eg. Eady growth rate/baroclinicity) and resistance to vertical velocity (eg. static stability).

Thank you for pointing this out. Given that local APE is a concept that is not widely used, we agree that discussing the construction of the reference state and what local APE "means" in more detail already in the introduction and methods sections will ease the understanding of the paper. In addition, we will add a schematic of how the reference state is constructed to the supplement (Fig. R1.1). We also agree that it would be beneficial to discuss local APE in light of other metrics for assessing the energetics of storms in the introduction, and will relocate this discussion from the end of the manuscript to the introduction accordingly.

[Figure]

*Fig. R1.1: Schematic illustration of the computation of the reference state. (a) Meridional cross section of an idealized baroclinic zone composed of air parcels, represented by boxes. Color indicates potential temperature of the air parcels, increasing from blue to red. (b) Sorting of air parcels by increasing potential temperature. (c) Construction of the reference state from sorted air parcels illustrated by a section through the solid Earth (grey) and the atmosphere (color). The dashed line represents Earth's rotation axis. The atmosphere is assumed to be shallow such that each circle has the same circumference.*

2. Discussion of APE changes linked to trajectories : I appreciated the approach taken here by tracking parcels through the troposphere. However, I found the discussion surrounding figure 3 challenging. I could understand (eventually) how the method discussed in line 216-217 was applied to the analysis, but it made the interpretation really challenging. I can see the idea behind wanting to see how much APE is depleted from the starting location, but it made a) the interpretation of what the actual change in column APE was at a location really difficult, and b) made comparison to figure 9 really challenging. A more clear motivation here may be helpful, as well as clarification here (or at figure 9) about the

differences in what we're looking at. One other approach could be to actually show either APE or dAPE/dt for your trajectories (akin to how you have pressure shown in figure 4). Further through this section, it would be helpful to really quantify the contribution due to omega/diabatic heating relative to the respective efficiency terms. Perhaps a box-and-whisker plot of distribution of the efficiency terms for the trajectories at different levels would help? There is good qualitative discussion here (see lines 264-266), but a bit more time spent on the quantitative side would really help. Lastly, your line 266-268 doesn't quite make sense to me r.e. linking the latent heat release to the second max of APE loss over the Canadian Arctic.

Thank you for bringing up the difficulty of interpreting Fig. 3 and your suggestions for improvement. We agree that the methodology behind Fig. 3 is not straightforward. Therefore, we will include a new figure in the supplement, which only shows selected trajectories and how APE evolves along them. This figure will also aid a more in-depth discussion of the role of efficiency vs forcing in creating APE tendencies.

We agree that the difference in methodology between Fig. 3 and Fig. 9 needs to be highlighted more. To better motivate Fig. 3 and to also facilitate the comparison of the case study to the climatology we will include Fig. R1.2 in section 3.2 before Fig. 3. Figure R1.2 shows the vertically-integrated instantaneous APE tendencies on February 19, 12 UTC. From this figure we can clearly see that the APE change mostly occurs on the edge of the polar APE reservoir. However, in comparison to Fig. 3, instantaneous APE tendencies are noisier than the Lagrangian APE tendencies and exhibit various dipole structures, which, when averaged along trajectories, are largely smoothed out. In fact, the smoothing of Lagrangian APE tendencies in our view facilitates linking the APE tendencies to the large-scale flow structures. Therefore, in our view showing the Lagrangian tendencies in Fig. 3 is preferable and better illustrates how the baroclinic development depletes the APE reservoir.

Lastly, the second maximum of APE loss over the Canadian Arctic is due to WCB ascent over the North Atlantic. The 48-h trajectories start over the Canadian Arctic and descend over the North Atlantic, consuming APE (Fig. 3b). However, some trajectories arrive in the boundary layer and already start to ascend again in a WCB airstream within those 48 hours. The 48 hours won't include all of the WCB ascent, which is why the negative APE tendency of the previous descent still largely outweighs this small positive adiabatic tendency due to the ascent phase at the end of the trajectories. In contrast, diabatic cooling (pos. diab. APE tendency) in the descent phase is weak, whereas diabatic heating (neg. diab. APE tendency) due to condensation in the ascent phase is very intense. Therefore, we observe a negative diabatic tendency in the 48-h forward integral of the trajectories over the Canadian Arctic. This effect can also be seen in the pressure evolution in Fig. 7b. We will add additional clarification to the text.

[Figure]

*Fig. R1.2: Vertically integrated APE tendency on 19 Feb UTC 12 (color shading). Shown are the (a) net change, (b) adiabatic contribution and (c) diabatic contribution. Also shown are the vertically integrated APE distribution (black contours; same intervals as Fig. 1) and the mean sea-level pressure (grey contours; every 10 hPa). The location of cyclone 1a is indicated.*

3. Dynamic interpretation/explanation of results: I appreciated that there were flavors of QG forcing, jet ageostrophic circulations, and PV thinking infused throughout the paper. That said, I think there were a few points where the discussion could've been taken into more depth and/or the QG/ageostrophic/PV discussions could've been more unified (eg. it's three perspectives to explain rising/sinking motion). For example, there was a lot of discussion on jet entrance/exit regions (which were at times not clear to me as entrance/exits rather than poleward flanks of the jet), but much less discussion on the role of curvature (where both the trough and upstream ridge can instigate ageostrophic circulations resulting in +/- omega), transverse thermally (in)direct circulations along jet streaks, and whether there was temperature advection occurring across the jet (which shifts where we see +/- omega in response to ageostrophic responses). Much of this could also be discussed from the perspective of PV theory (which emerges late in the paper but could've been interwoven throughout).

Thank you for this observation. We agree that we neglected the role of curvature in instigating ageostrophic vertical circulations in the discussion of our results, which we plan to improve. We will also include PV thinking earlier on along the omega equation perspective.You bring up temperature advection across the jet, and how it shifts the ageostrophic circulation relative to the jet axis. While we believe this effect is very interesting and highly relevant in synoptic meteorology, we do not intend to further distinguish between the effects of confluence/diffluence, curvature and temperature advection on QGomega. Our overarching goal is to extend the planetary perspective of APE to synoptic scales, and then to qualitatively relate it to classical theories of synoptic meteorology. Providing a quantitative account of the different QG omega forcings would lead too far away from this qualitative goal. Thus, we plan to include a discussion of the different possible effects on QGomega, including temperature advection, without analyzing in detail which effect dominates for our case study.

4. Inclusion of CAOs: Though I found the inclusion of the CAO case and composite interesting, I wasn't really sure how much it contributed to the paper. This may benefit from either removing it, or better motivating why you're including it.

Thank you for this comment. We agree that the discussion of CAOs stands a bit isolated in the manuscript. This part is motivated by the observation of Novak & Tailleux (2018) that local APE is diabatically consumed by upward surface sensible heat fluxes and the release of latent heat in convection over the western boundary currents. Strong upward fluxes are generally related to cold air sweeping across the Gulf Stream front towards the warm waters (e.g., Grossmann and Betts, 1990; Fletcher et al. 2016). We will refocus the discussion on this question, the role of Gulf Stream air-sea interaction and the relevant synoptic conditions for creating this diabatic tendency. This line of reasoning will better motivate our analysis of CAOs.

Specific comments:
• Lines 53-54: Please clarify/elaborate on the 'why' of APE being in the polar middle and upper troposphere and how it is advected (eg. is it a material quantity?).

We will refer to the reference state here to explain why APE is located in the polar middle and upper troposphere and rephrase as follows:
"In particular, they find that APE is mainly located in the polar middle and upper troposphere, where the air is colder than its reference state. They also showed that APE is advected from this polar reservoir into the storm tracks."

We also agree that it would be helpful to clarify what is meant by advection of APE. However, we think this is more suitable for Sec. 2 (Data and Methodology), where we formally introduce the equation for the APE tendency. We will add the following to Sec. 2.2:
"Terms (I)-(III) present sources and sinks to local APE. Therefore, APE is conserved along adiabatic flow if omega is zero. This means that APE is conserved only if isobaric and isentropic surfaces overlap. Otherwise, adiabatic motion will incur an adiabatic APE tendency."

• Lines 61-62: For local APE, is it only accelerations/decelerations? For example, do all air parcels in the warm conveyor belt that contribute to a positive APE tendency explicitly have a deceleration occurring?

The acceleration due to a conversion of APE to KE by an air parcel is not necessarily experienced by this air parcel itself. This can be seen from the kinetic energy equation (Novak and Tailleux 2018; Federer et al. 2024):

$$\frac{D}{Dt}\frac{\mathbf{V}^2}{2} + \nabla_h \cdot (\Phi'\mathbf{V}) + \frac{\partial(\omega\Phi')}{\partial p} = -\{\alpha(\theta, p) - \alpha[\theta_r(p, t), p]\}\omega + \mathbf{F} \cdot \mathbf{V}, \quad (5)$$

Here, the first term on the right hand side in curly brackets represents the conversion between APE and KE and the second term is dissipation of KE due to friction. While the

conversion of APE to KE is the only source term in this equation, there is also a material change of KE due to the second and third terms on the left hand side, which represent geopotential fluxes. When volume-integrated globally, these fluxes become zero. Locally, however, these terms are generally not zero and will redistribute the KE gained from APE conversion (Papritz and Schemm, 2013). Thus, an air parcel that loses APE, may not gain the equivalent amount of KE in case it is located in a region of divergent geopotential fluxes. We think that our wording might be misleading, which is why we will rephrase as follows: "This is consistent with a local notion of APE, where air parcels accelerate the flow by converting APE to KE, but also decelerate it by converting KE back to APE."

• Lines 110-112: Make two sentences – it's a bit hard to follow as written.

Thank you for the suggestion. We will rephrase accordingly.

• Section 2.2: Please include units throughout here. It may be helpful to also provide equations for specific density and your diabatic heating rate.

Thank you for this suggestion. We will include the equations for specific volume, the pressure velocity and the diabatic heating rate. We think that the units will be more clear from those equations.

• Line 170: From your figure 1, it appears less that an upper level trough is developing rather than propagating in from the west. Please be sure that it is developing (rather than propagating) if you use the term 'upper-level trough forms'.

Thank you for this observation. We plotted the PV field on 315 K for an extended time window centered on  the North Atlantic to clarify (Fig. R1.3). We agree that there exists an upstream feature over the western United States (Fig. R1.3a). As cyclone 1a intensifies, this PV trough remains stationary and does not propagate into the North Atlantic (Fig. R1.3b). Given the strong elongation of the trough in the North Atlantic (Fig. R1.3c), we think it's fair to say that this trough formed in the North Atlantic concurrent with the intensification of cyclone 1a. However, we agree that the APE located over the Great Lakes in Fig. 1a appears to propagate into the North Atlantic. This is in line with our argument that APE is supplied to the storm track through advection.

[Figure]

*Fig. R1.3: Isentropic PV on 315 K (color shading) for (a) 18 Feb UTC 12, (b) 19 Feb UTC 12 and (c) 20 Feb UTC 12. Also shown is the mean sea-level pressure (grey contours; every 10 hPa). The location of cyclone 1a is indicated.*

• Lines 171-172: I think you need to spell out more how the collocation of high APE with the DT is demonstrating the connection to the large scale circulation (I don't disagree with you, but it needs to be more clearly demonstrated).

We agree and will rephrase as follows:
"High APE values are clearly co-located with the trough extending into the North Atlantic. This points to a strong link between the upper-level PV pattern and the distribution of APE. Because the distribution of PV governs the large-scale circulation, APE is likely linked to the large-scale circulation as well, as will be explored in the following sections."

• Lines 179-180: Do we know that this is low APE advection vs. a time tendency due to conversion? In theory, one could compute APE advection to show this explicitly. The same goes for lines 183-184.

On one hand, we agree that at this point in the manuscript the negative APE tendencies in the warm sector of cyclone 3a could also be interpreted as a local consumption of APE. However, local consumption alone would appear to be somewhat counterintuitive, because we typically observe warm air advection ahead of a developing cyclone, and we show that APE values to the south of cyclone 3a are not particularly high. Therefore, we think that our claim that low-APE air is advected to this location is well supported. On the other hand, the results following this section also show that the ascending air ahead of cyclone 1a generates APE due to the positive adiabatic efficiency, which is consistent with the southerly origin of the air. It is reasonable to assume that this is also the case for cyclone 3a.
Indeed it would be possible to compute the advection of APE explicitly. However, we didn't compute the advection on purpose. In our opinion, the trajectory analysis yields more insight into how APE is transported along with the flow and how the evolution of the APE is linked to the dynamical processes.

• Lines 202-204: This isn't really a stand-alone paragraph – please aim to elaborate or merge with another paragraph.

We agree and will merge this paragraph with the previous one.

• Lines 206-207: The cross-section does show the cold dome and therefore a troposphere-deep cold anomaly. It does not, however, imply cold air advancing into the midlatitudes. Consider providing different evidence or re-writing.

Considering the co-evolution of the trough and the cold-dome suggests that the development of the upper-level trough is typically associated with cold air advancing equatorward underneath the trough as opposed to local diabatic cooling. However, we agree that this cannot be concluded based on the cross section alone. Therefore, we will rephrase as follows:
"The vertical distribution of APE shows that APE is concentrated in the middle and upper troposphere north of the midlatitude baroclinic zone. The upward bending of isentropes by the positive PV anomaly implies a dome of cold air beneath the trough, which in turn is associated with large APE since the air is colder than its reference state."

• Figure 1 cross-section lines/Figure 2: Consider given start/end point markers for your cross-sections (eg. A-A', B-B', C-C') to avoid any ambiguity about the direction of your cross-sections.

Thank you for this suggestion. We have added start and end point markers to Figs. 1 and 2.

• Lines 222-225: This may benefit from a schematic (this infuses with general comment 2 above).

Thank you for this suggestion. We think the figure we describe in response to your comment 2 will address this comment here too.

• Line 227, 228 (and elsewhere): Careful on geolocation descriptions. For example – the Canadian Arctic is a pretty expansive area. But it appears your focus in figure 3 may be more on Hudson Bay/Quebec/Labrador? Likewise, I wouldn't consider the Gulf Stream to be adjacent to the Canadian Arctic. A reference geographic map (or outline of the Gulf Stream region) may be helpful.

Thank you for pointing this out. We will add more precise location references.

• Lines 243-248: There's a large region of positive APE tendency south of Iceland that likely is playing a factor the in the cyclone evolution. It would be helpful to either look at trajectories/APE tendencies in that region as well, or to provide justification for not examining it.

Thank you for this suggestion. We indeed did not discuss this region of large positive APE tendencies. The reason for this is that this APE tendency is due to ascent in the warm sector of the preceding cyclone, which is located south of Iceland in Fig. 3. Therefore, this tendency is most relevant for the downstream cyclones, whereas here we are concerned with the Gulf Stream region. We will rephrase slightly to put more emphasis on the fact that we are interested in the APE conversions associated with cyclone 1a:
"For this purpose, we select two regions of particularly large APE tendencies in the vicinity of cyclone 1a (Fig. 3b)."

• Lines 258-260: I would find it helpful to see some discussion of this also in reference to the effiiency terms.

Thank you for this suggestion. We agree that this part would benefit from a discussion of the role of efficiency, in particular in determining the sign of the APE tendency. We will add a discussion of the efficiency as follows:
"The rapid ascent of these trajectories is reminiscent of a warm conveyor belt (WCB), which is associated with strong latent heat release. Indeed, the region of negative diabatic APE tendency observed slightly downstream of cyclone 1a coincides with the ascending air parcels. This also implies that the thermal efficiency of the air parcels must be negative, which means that the air parcels are colder than their reference state. Therefore, latent heat release in the ascending air stream leads to a destruction of APE."

• Line 280: I'm not sure the line 'detaches from the band of downward …' is the best choice of phrasing. Please reword.

We agree and will reword.

• Line 283: I'm not sure I would classify this as the entrance region of the jet. It may be better discussed as downstream of the upper-tropospheric ridge.

Thank you for raising this point. We agree that the location of the trajectory descent is better described as downstream of the upper-tropospheric ridge. However, the descent does not occur uniformly distributed along the jet. Therefore, we will reformulate as follows:
"On 20 Feb 00 UTC (Fig. 5a) the majority of trajectories descend along the downstream flank of the upper-tropospheric ridge, predominantly on the poleward side of the jet axis, while few trajectories descend further south in the vicinity of cyclone 1a."

• Lines 290-291: Though the trough may play a role here, so too does the upstream ridge (contributing to QG forcing for descent).

We agree that the phrasing is misleading since the ageostrophic circulation is a consequence of the baroclinic wave as a whole. Since we want to simply emphasize the connection between APE conversion and the ageostrophic circulation, we will not mention the trough here.

• Figure 5: I found it challenging to differentiate the dark green (2-PVU) contour from the black (300 hPa wind speed) contours. Please consider a slightly more contrasting color choice. I would also add a 'L' to represent your surface low position in both panels.

Thank you for your suggestions. We will choose a more contrasting color for the 2-PVU line and will mark the location of the surface cyclone.

• Lines 322-323: I would be careful about making this statement. The surface cyclone can influence the upper-troposphere and vice versa (eg. cyclone growth positive feedback cycle), and your cross-section shows a troposphere-deep cyclonic cold anomaly and circulation.

Thank you for pointing this out. We agree that the baroclinic interaction between upper levels and the surface cyclone is inherent to baroclinic instability. Therefore, we think a different phrasing is needed to express our conclusion that APE conversion does not happen at lower levels within the cyclone, as for example meridional heat fluxes do, but is governed by the upper levels. We will rephrase this bullet point as follows:

"The vertical motion induced by the interaction of the intensifying surface cyclone and upper-level trough lead to both APE consumption and production. APE is primarily consumed by the descent along the western flank of the trough, while APE is produced by the ascent ahead of the trough. This pattern of ascent and descent is consistent with QGomega forcing by upper levels. Therefore, the major part of APE conversion takes place through the circulation in the mid to upper levels, and not within the surface cyclone itself."

• Figure 8: Aim to make figure 8b seems more its own independent panel (maybe a box around it?). I didn't notice it even when referenced in the text.

We will increase the size of the boxplot to make it stand out more.

• Line 338: How many times steps overlap as being both in the delta-Pmax category and the dmax category?

Each life cycle stage is defined as only a single time step along a cyclone track. As the maximum depth implies that no more deepening must take place, delta-Pmax will always occur before dmax. Hence, there is no overlap. Also for Fig. 10 and the corresponding analysis, there is no overlap between the 12-hour windows used.

• Line 341-342: Again, not sure this is the jet entrance region. I suspect curvature dynamics are playing more of a factor.

Thank you for pointing out the role of curvature here. We agree that the dipole pattern in the QGomega forcing aligns well with the forcing expected from an upper-level trough. We will rephrase accordingly.

• Line 345: I think you can lean into PV thinking for the occlusion stage here (eg. evidence of a stacked cyclonic PV anomaly circulation from surface to dynamic tropopause, along with the surface low receding into the cold PV hook.

Thank you for this suggestion. We will better incorporate the dynamics of the occlusion stage from the PV perspective here.

• Line 351-352 (and elsewhere in this section): Consider adding some values to the description here.

Thank you for this suggestion. We agree that this is helpful and will include some of the amplitudes for the APE tendencies observed in Fig. 9.

• Lines 352-353: Is this PV field shown anywhere? It may be beneficial to include.

We refer to the PV field at the time of maximum depth shown in Fig. 9c. Downstream of the ridge, the curvature of the PV field suggests a new trough.

• Lines 372-374; 399-405: Consider bringing in some discussion of the poleward movement of your composite when discussion the radiative cooling of the free atmosphere.

Thank you for this suggestion. We will add the mean locations of the composites in Fig. 8 and also include the following discussion:
"The diabatic APE generation due to radiative cooling increases between the times dp_max and d_max when the poleward displacement of the mean composite location is largest (Fig. 8). This means that the environment of the cyclone becomes colder with a more negative thermal efficiency. Thus, the radiative cooling leads to larger diabatic APE generation."

• Lines 379-380: How sensitive is your analysis to the region chosen here? I can understand 2000 km from a synoptic scale perspective, but does this strongly impact the interpretation of storm-scale contributions?

Thank you for bringing up the sensitivity of our analysis. As we discuss in the text, the main factor influencing this sensitivity is the fact that the positive APE tendency is mostly found close to the center of the cyclone, whereas the negative APE tendency can be located further away at later stages of the cyclone life cycle. We think that the chosen radius adequately addresses this issue while still reflecting the dynamics of the synoptic system. To illustrate the sensitivity, we will recompute the spatial integral for different radii and include the resulting figures in the supplement.

• Lines 415-416: Why not take your cross-section through the center of the surface cyclone? Here your baroclinic zone is south of the surface cyclone simply because the trough is further south to the west of the cyclone (as you noted).

We take the cross sections to best capture the vertical structure of the strongest APE tendencies. Therefore, they cut through maxima in the negative and positive APE tendencies. Fig. R1.4 shows the same as Fig. 11 but through the center of the cyclone. Here, the meridional cross section of the adiabatic APE tendency (Fig. R1.4b) only shows the positive tendency at the cyclone center, while we want to focus on the structure of the negative tendency in the cold sector of the cyclone.

[Figure]

*Fig. R1.4: Cross sections of cyclone-centered composites at the time of maximum intensification. (a)-(c) are meridional cross sections and (d)-(f) are zonal cross sections through the cyclone center. (a),(d) PV (shading) and horizontal wind speed (blue; every 10 m/s). (b),(e) adiabatic APE tendency (shading) and APE density (orange contours; every 500 J/kg). (c),(f) diabatic APE tendency. Potential temperature is indicated by grey contours (every 6 K).*

• Lines 425-426: I found this sentence unclear. Which positive adiabatic APE tendency?

Thank you for pointing this out. We agree that the sentence is unclear because we don't refer to this positive adiabatic APE tendency before. We will reformulate as follows: "Both the positive and negative adiabatic APE tendencies extend through most of the column, peaking around 500 hPa."

• Section 4.2: Please provide information on how many CAO cases were involved.

Roughly 10% of the 285 cyclone tracks (27) were not associated with a CAO. We will include this information in the text.

• Lines 455-456: I think you need to reverse the order here (eg. trough leads to surface cyclone, not vice versa).

Our intention here was not to imply causality, but to describe the interaction between the cyclone and the trough as the mechanism that strengthens both the trough and the APE beneath it. We agree that we can improve the phrasing to better reflect this as follows:

"Intense Gulf Stream cyclones typically originate at the edge of the polar APE reservoir. The baroclinic interaction of those cyclones with upper-level troughs induces a cold anomaly in the mid to upper troposphere beneath the trough."

• Lines 469-470: Your zonal cross section implies a troposphere-deep circulation, so this feels a bit misleading as written to me.

Thank you for pointing this out. Indeed the word 'aloft' is misleading here. We will rephrase as follows:
"We observe complex patterns of APE tendency, largely following the trough and ridge patterns aloft. These ensue from the  ageostrophic circulation induced by large-scale flow features such as upper-level troughs and ridges and their baroclinic interactions with the lower troposphere."

• Lines 485-492: You mentioned the Bowley et al. 2019 paper in the introduction – though their results are a bit hard to interpret relative to yours (given the global zonal APE vs. local total APE perspective), you may want to include some of their interpretation here. For example, they found a dominant mechanism for synoptic scale APE increase to be ascent on the poleward flank of the wave guide in the exit region of the North Pacific jet, which fits well to the results of Koch et al. 2006 and your adiabatic generation interpretation here.

Thank you for your suggestion. We agree that the adiabatic generation of APE through ascent fits well to the findings reported in Bowley et al. 2019. However, Bowley et al. identified diabatic heating and cooling as the dominant mechanism of increasing the meridional temperature gradient across the baroclinic zone in the Pacific and with that increasing zonal APE. Our study suggests the possibility of a different pathway to APE build-ups: Instead of diabatic generation, a dominance of positive adiabatic APE tendency could also lead to an APE build-up. This would correspond to a dry-dynamic global deceleration of the flow, converting KE to APE through ascent of cold air. We will add this comparison to our discussion section.

• Line 517-520: I think your points here would be beneficial to appear earlier in the manuscript when introducing the local APE framework to help unify the global vs. local perspectives.

Thank you for your suggestion. We agree that the structure of the reference state and its implications for the distribution of local APE should be discussed in more depth in the introduction and will make additions as necessary.

Technical corrections:
• Line 75: Change 'and to contribute' to 'and can contribute'.
• Line 98: I'm not sure 'relies' is the right word choice for your data here.
• Line 167: Please write out 'potential vorticity unit' the first time you define PVU.
• Line 343: Please add 'cyclonic' between 'upper-level' and 'Rossby wave breaking'
• Line 430: Should this read 'ascent leading to the positive adiabatic APE tendency'?

Thank you for your suggestions. We fully agree and will correct the text accordingly.

**Review by Lance F. Bosart**

A) Overview: Federer et al. (2024): Synoptic Perspective on the Conversion and Maintenance of Local Available Potential Energy in Extratropical Cyclones

The authors are to be commended for preparing a well-evidenced story and well-written story on the often challenging concept of atmospheric energetics in general and available potential energy in particular. I learned something from reading this paper.

Thank you for your positive evaluation of our manuscript. We are grateful for your constructive and insightful feedback, and we highly appreciate the time you put into this review. We believe the suggested changes will significantly improve our manuscript and we hope the revised version will match your expectations.

B) Specific comments on Federer et al. (2024): Synoptic Perspective on the Conversion and Maintenance of Local Available Potential Energy in Extratropical Cyclones

1. Line 104: Why is omega only computed at 500-hPa. Wouldn't a lower level (e.g., 850-hPa or 700-hPabe be better for sampling and quantifying thermal advection?

We agree that a lower level would be more suitable to investigate thermal advection along the surface baroclinic zone. However, high APE values are mostly located in the middle and upper troposphere, which is why thermal advection along the surface baroclinic front does not incur large APE tendencies. Therefore, we believe that omega at 500 hPa is most suited for our purpose. Nevertheless, thermal advection along the lower tropospheric baroclinic zone could still affect QG omega at 500 hPa via the far-field effect. To test this hypothesis, we show QG omega due to lower tropospheric forcing in Fig. R2.1 for the two time steps also shown in Fig. 5. The QG omega vertical motion at 500 hPa due to lower tropospheric forcing is strongest in the immediate vicinity of cyclone 1a. However, its amplitude is much smaller than the forcing from upper levels (Fig. 5). Moreover, much of the vertical forcing associated with the descent of trajectories along the western flank of the upper-level trough is purely from upper levels. Therefore, we conclude that QG omega forcing from lower levels is less important for APE conversion.

[Figure]

*Fig. R2.1: Low-level quasi-geostrophic omega on 500 hPa (color shading) for two time steps. Also shown are the 2-PVU contour on the 315 K isentrope (green) and wind speed on 300 hPa (black contours; 60, 80 and 100 m/s).*

2. Line 112: Is it possible that your criterion for identifying CAOs could be sensitive to your choice of the 850-hPa level for air-sea temperature differences? Could shallow cold air be a problem in some situations?

We agree that shallow layers of cold air might not be detected as CAOs if they do not extend to the 850 hPa level. Such shallow layers of cold air are particularly common in the Arctic, for example, over sea ice when they are capped by a strong inversion. However, once the air is advected over open ocean, strong surface fluxes lead to a rapid growth of the boundary layer such that the CAO boundary layer will quickly extend beyond the 850 hPa level and the CAO can be detected (see for example composites of boundary layer heights in Fletcher et al. 2016, their Fig. 9). In the case of CAOs over the western boundary currents, the initial thickness of the boundary layer is likely less shallow than in the Arctic. Moreover, due to the strong horizontal SST gradient across the Gulf Stream front, the air-sea temperature difference quickly becomes large as the air progresses across the front (often far in excess of 8 K used as a threshold here), going along with very intense surface heat fluxes of typically several 100 W / m² (e.g., Grossman and Betts, 1990), and accordingly a rapid expansion of the boundary layer beyond 850 hPa (e.g. Vannière et al. 2017). Thus, in rare cases where the boundary layer is initially very shallow, the CAO would remain undetected only during its very early phase. In addition, the erosion by diabatic heating of very shallow cold layers could only have a small impact on the vertically integrated APE budget since they would represent only a small fraction of the mass in the atmospheric column. For these reasons, we think that for our purposes the 850 hPa level is acceptable.

3. Line 134: ….given by the product of omega and the buoyancy……

We will correct the sentence according to your suggestion.

4. General question: What is the vertical resolution of your dataset? How well are you resolving shallow Arctic air outbreaks over continents and oceans?

We use the ERA5 reanalysis with its full 137 vertical model levels. Since the data is on hybrid sigma-pressure levels, the resolution is higher at lower levels. Up to 850 hPa, the vertical grid spacing ranges from 20-200m (more details can be found here: https://confluence.ecmwf.int/display/UDOC/L137+model+level+definitions). As mentioned above, we think that this vertical resolution sufficiently captures CAOs over the Gulf Stream.

5. Lines 142-148: I had trouble visualizing this somewhat convoluted text. Suggestion: Inserting a schematic diagram here that depicts the calculation of the reference state would be most helpful to the reader.

Thank you for pointing out this difficulty. We agree that the computation of the reference state is not straightforward. Therefore, we will include a schematic (Fig. R2.2) in the supplement and clarify parts of the text describing the methodology.

[Figure]

*Fig. R2.2: Schematic illustration of the computation of the reference state. (a) Meridional cross section of an idealized baroclinic zone composed of air parcels, represented by boxes. Color indicates potential temperature of the air parcels, increasing from blue to red. (b) Sorting of air parcels by increasing potential temperature. (c) Construction of the reference state from sorted air parcels illustrated by a section through the solid Earth (grey) and the atmosphere (color). The dashed line represents Earth's rotation axis. The atmosphere is assumed to be shallow such that each circle has the same circumference.*

6. Lines 150–151: Curious to learn why you chose not to compute reference states for DJF, MAM, JJA, and SON?

The reference state shows considerable seasonality as well as a trend due to global warming. We account for the trend by smoothing the reference state with a 9-yr running mean, as opposed to computing an average over all years. We take the same approach to retain seasonality by computing a 31-day running mean of the daily-mean reference state. Compared to a 31-day running mean, a DJF mean would lead to significant biases in the APE distribution and tendency between early winter and late winter time steps.
We realize that the description of how we compute the reference state in the manuscript is not sufficient. We will add the following to clarify:
"To facilitate a comparison of APE between cyclones at different time steps, we apply a process of time smoothing to the instantaneous reference states. First, we compute daily averages. Next, we apply a 31-day running mean to retain the seasonality. Lastly, we apply a

9-year running mean to each calendar time step to take into account the effect of global warming on the reference state."

7. Figure 1: Could, say, 700-hPa vertical motion be added to the six figure panels so that the locations of major ascent and descent regions relative to trough and ridge positions can be better pictured?

Thank you for this suggestion. While we believe additional fields would be helpful here, we also want to avoid overcrowding the figure with a third set of contour lines. Therefore, we decided to not include the vertical motion here.

8. Figure 1: Labels 1a, 1b, 2c, etc. need to be defined in the figure captions. Label latitude and longitude lines.

Thank you for pointing out the missing definitions. We have added them to the figure caption and added labels to the lat/lon lines.

9. Figure 1f: There appears to be a remnant cyclone near Iceland in panel 1f that is a remnant of cyclone 1b in panel 1e.

Good observation. There is a low pressure area between Greenland and Iceland. Even though our cyclone tracking scheme does not continue the track of cyclone 1b to 23 Feb 1200 UTC, we believe that this cyclone might have arisen by a merger of cyclones 1a and 1b. Since we are mainly interested in the growth stage of the cyclones, where kinetic energy is generated, we believe that describing this remnant will distract the reader from the main message of the figure.

10. Figure 1: There appear to be three cold surges based on equatorward-directed trough extensions in Fig. 1. Significance? How do the three cold surges.

Yes, we agree that we observe three cold surges, in close succession and in a very similar synoptic setting, which we think is a particularly interesting aspect of this case study. The occurrence of these cold surges is likely linked to the fact that the case study is also identified as a European blocking weather regime, which implies some quasi-stationarity in the upstream storm track, as observed here.
The cold surges are related to the expansion and maintenance of the polar APE reservoir into the midlatitudes. For instance, cyclone 2a is accompanied by a strengthening of the APE tongue stretching into the central North Atlantic in its wake. This stage is likely also accompanied by cold air advection in the lower troposphere and a CAO.
While this is certainly fascinating, we only very briefly speculate on the role of APE in maintaining this synoptic situation when we discuss cyclones 3a and 3b. We think that further speculation would go beyond the scope of the analysis and would not be supported by the following climatological analysis.

11. Figure 1: Cyclone 2a, and especially cyclone 3b, are not directly associated with an upper-level trough in their formative stages. Significance?

This is indeed an interesting observation. Upon investigating the upper level forcing using QGomega at 500 hPa (Fig. R2.3) we find that cyclone 2a forms in a region of ascent (Fig.

R2.3c,d). As you noted, the formation is not associated with a new trough propagating into the region. But earlier time steps indicate that the upper-level forcing interacting with cyclone 2a originated from the ageostrophic circulation on the poleward flank of an upstream jet streak (Fig R2.3a,b).

[Figure]

Fig. R2.3: Upper-level quasi-geostrophic omega on 500 hPa (color shading) for four time steps with the 2-PVU contour on the 315 K isentrope (blue) and wind speed on 300 hPa (black contours; 60, 80 and 100 m/s). The positions of cyclones 1a, 1b and 2a are also shown.

The same analysis for cyclone 3b (Fig. R2.4) reveals that it quickly couples with the upper-level forcing due to the curvature of the upper-level trough associated with cyclone 3a (Fig. R2.4d). Cyclone 3b only formed very briefly before experiencing the upper-level forcing (Fig. R2.4c), possibly due to frontal instability.

To conclude, cyclones 2a and 3b are not directly associated with an upper-level trough during their initial phases. Nevertheless, they are substantially influenced by upper-level forcing and, concluding from our analysis, not fundamentally different from the other cyclones described in the case study. Therefore, we believe that the trajectory analysis conducted for cyclone 1a is representative of the general pattern of APE conversion within the case study.

[Figure]

*Fig. R2.4: Upper-level quasi-geostrophic omega on 500 hPa (color shading) for four time steps with the 2-PVU contour on the 315 K isentrope (blue) and wind speed on 300 hPa (black contours; 60, 80 and 100 m/s). The positions of cyclones 2a, 2b, 3a and 3b are also shown.*

12. Lines 185-187: Please be more specific about where the APE decline is occurring.

We will rephrase as follows:
"The dissipation of APE in the central North Atlantic and the end of intense North Atlantic cyclogenesis are accompanied by the development of an upstream cyclone, which inhibits further APE supply to the North Atlantic."

13. Figure 2 caption: Please indicate directions at either ends of the cross-section lines (e.g., west or east, etc.) for reference purposes.

Thank you for the suggestion. We will indicate the direction of the cross-sections with A-A', and also add this to Figure 1.

14. Lines 202-203: This sentence needs further elaboration since Fig. 2c is devoid of vertical motion.

We will rephrase as follows:

"The zonally oriented cross section through the trough (Fig. 2c) further illustrates the cold anomaly below the trough. From left to right the isentropes slope upwards, concurrent with an increase of APE. Thus, the isentropic lifting beneath the upper-level trough is accompanied by large APE values. The APE located in this region is bounded by a jet stream toward the west and cyclone 1a toward the east."

[Figure]

*Fig. 2c: Cross section as indicated in Fig. 1. Shown are APE (color shading), the 2-PVU contour representing the dynamical tropopause (green), the horizontal wind speed (black contours; every 20 m/s) and potential temperature (grey contours; every 6 K).*

15. Figure 3: Geography is hard to distinguish from the color shading and black contours.

Thank you for pointing this out. We will change the coloring of the geography.

16. Figure 3: SLP contours are very difficult to follow because they are so faint in Fig. 3 (and in Fig. 1)

Thank you for pointing this out. We will increase the thickness of the SLP contours such that they are more easily recognizable.

17. More on Fig. 3. So, is it fair to say that APE generation equatorward of the jet axis acts to increase APE in lower latitudes (with a minor offset by diabatic processes in the vicinity of midlatitude cyclones)? If so, what can we conclude about these APE changes from a general circulation perspective?).

The APE generation equatorward of the jet axis in Fig. 3 is due to air parcels ascending polewards of the equatorward flank of the jet, as shown and discussed in Fig. 4. Therefore, those trajectories resupply APE to the polar APE reservoir. At the same time, air parcels descending equatorward consume APE from the polar reservoir. This corresponds to a deceleration (KE → APE) and acceleration (APE → KE) of the flow, respectively. In a steady state, the global integral of the APE tendency would thus be zero. From the global circulation perspective, we expect this integral to be negative because KE is dissipated through friction, meaning there needs to be more local APE consumption than generation.

18. Line 247: Check whether AOE has been defined previously.

This should read APE. We will correct it accordingly.

19. Figure 4: SLP contours are too faint to be see properly, especially in the vicinity of the trajectory swaths.

Thank you for pointing this out. We will increase the thickness of the SLP contours such that they are more easily recognizable.

20. Figure 4b: How sensitive do you think the split between subsiding and ascending trajectories shown in Fig. 4b would be to the starting time and location of the trajectory computations? Presumably, the expected trajectory differences could be linked to the governing dynamics?

We are not sure about what you are referring to. All trajectories in Fig. 4b contribute to the depletion of APE. In this case, this corresponds to a negative adiabatic APE tendency and therefore to a descent. However, some trajectories descend a lot further than others. We think the determining factor is the height at which the trajectories are started. Trajectories from the upper troposphere have quite large initial APE values, and with that a large adiabatic efficiency. Thus, little vertical motion is required for them to contribute to APE depletion, whereas trajectories from the middle troposphere have lower initial APE values, requiring larger vertical displacement to result in the same APE depletion. Furthermore, we believe the trajectories depict a typical flow situation on the western flank of upper-level troughs. This might depend on the life cycle of the trough, but we don't expect large differences in trajectories contributing to APE depletions started from different times or locations.

21. Lines 264–272: So, presumably the extent and magnitude of latent heat release in ascending trajectories would be very sensitive to the air mass in which the trajectories originate. For example, Mediterranean and Gulf of Mexico cyclones that have access to tropical and subtropical air masses would exhibit different APE changes in conjunction with ascending trajectory swarms than the higher-latitude example referenced in this paper? How should we interpret such differences?

You bring up an interesting aspect about the latitude-dependence of APE conversion and the role of latent heat release, which relate to two different aspects of the APE framework. First, we should clarify the role of latent heat release, which we also discuss in the manuscript. The local APE framework is dry-hydrostatic. Therefore, the acceleration of vertical motion through latent heat release is not explicitly represented. This means that while latent heat release results in a negative diabatic APE tendency, it also leads to a greater positive adiabatic tendency. We hypothesize that the positive tendency compensates for the negative tendency, and that the net contribution of latent heat release to the net APE tendency is positive. That said, among two cyclones at the same location, but with different moisture availability, the moister cyclone will exhibit more negative diabatic APE tendency and more positive adiabatic APE tendency in the ascending air stream than the drier cyclone. Second, the magnitude of those APE tendencies will depend on the latitude at which the cyclone occurs. Cyclones further south typically form in a warmer environment, which means

that the adiabatic and thermal efficiencies of APE conversion will be different from cyclones in a colder (northern) environment. However, the efficiency affects both the positive and negative *local* tendencies equally. Thus, we would expect the *volume-integrated* tendency in the vicinity of the cyclone to be unchanged by the latitude.

In any case, the question of how cyclones in different locations contribute to the global integral of APE conversion is very interesting and we will include this point in the outlook of our manuscript.

22. Figure 5 (and accompanying text). Have you checked whether a mesoscale trailing trough behind the main trough could be altering the larger-scale QG ascent-descent dipoles? Would it be helpful to add the position of the surface cyclone to Figs. 5a,b to assist the reader in better understanding the relationship between the trajectory evolution and the surface cyclone location?

Indeed the discussion around your point Nr. 11 and Fig. R2.3/2.4 points towards a mesoscale mobile trough / frontal zone trailing the main trough. We think this feature mainly influences the development of cyclone 2a, whereas the discussion around Fig. 5 is focused on cyclone 1a. Thank you also for your suggestion to include the position of the surface cyclones in Fig. 5, which we will implement.

23. Lines 288-292: While I think that your conclusions mentioned here are correct to first order, it might be a good idea to allow for considerable variability the would likely be seen in a much larger sample of cases (e.g., a secondary trough deepening downstream into a pre-existing trough vs. a weakening trough that was lifting out to the northeast (among other possible trough-ridge scenarios).

Thank you for raising this point, which was also a concern of reviewer #1. Indeed the vertical forcing from upper levels is a combination of different processes, which might arise by the presence of different features, such as troughs, secondary troughs or upper-tropospheric frontal zones and jet streaks. We plan to more explicitly make the reader aware of this variability in the text.

24. Lines 294-311: Very interesting perspective on APE conversion that is well supported by the trajectories shown in Fig. 6 and the APE tendency plots shown in Fig. 7. An obvious question….that really cannot be addressed here….is how general this result would would be across a variety of cyclone evolutions at different times of the year and the choice of where and when to initiate the trajectories.

Thank you. We think that the climatological analysis in the second part supports this result at least for DJF Gulf Stream cyclones. While the question of seasonality is certainly interesting, we would expect the strongest effect of surface fluxes on APE to be concentrated in DJF due to the large land-sea temperature contrast in winter.

25. Fig. 7: Would it be helpful to rescale the y-axes in Fig. 7 so as to make it more obvious how much smaller the diabatic APE tendency is compared to the adiabatic APE tendency?

We agree that this would help to put the difference between the adiabatic and diabatic tendencies better into perspective. However, it then becomes harder to recognize the dip in

diabatic tendency around hour 0, which is still significant (Fig. R2.5). As a middle ground, we will adjust the y-scale of the diabatic tendency to be exactly half of the adiabatic tendency. With that, the timeseries of the diabatic tendency is still readable but at the same time it's easier to grasp the difference in scale to the adiabatic tendency.

[Figure]

*Fig. R2.5: Evolution of (a) APE density, (b) pressure, (c) adiabatic APE tendency and (d) diabatic APE tendency of CAO trajectories, centered on their arrival in the CAO region (time zero). The solid black line indicates the mean, and the grey shading denotes the range between the 5th and 95th percentile of the trajectories.*

26. Line 316 (first bullet): How might this finding change (or not change) if you looked at Arctic PV extrusions across all longitudes as a function of season?

Analyzing the seasonality of APE lies outside the scope of our study, but we do believe that this is a very interesting aspect that could be explored in future work, and we will mention it in the outlook section. We hypothesize that, to first order, the character of the results wouldn't change. Meaning that Arctic PV excursions will be associated with large APE values relative to climatology in any season. This is well substantiated by PV dynamics and the isentropic lifting beneath PV troughs. However, absolute APE values would be a lot smaller in the summer than in the winter hemisphere, as the meridional temperature contrast is weaker. This expectation is also in line with less and weaker cyclogenesis in the summer hemisphere.

27. Line 320 (second bullet): While I agree with this conclusion, to play the contrarian a bit and reference "Animal Farm" meteorology…..while all cyclones may be equal, some cyclones may be more equal than other cyclones. How do we find these "more equal" cyclones and how can we understand why there are "more equal cyclones" with regard to where they form, how they evolve, and how they contribute to APE generation? For example, 1) How "unequal" are continental cyclones from oceanic cyclones?, and 2) Are

APE changes sensitive to whether cyclones are moving mostly poleward (e.g., cyclones along the east coasts of Asia and North America) or moving mostly equatorward (e.g., lee cyclones east of major north-south mountain barriers? Just curious……

While we do not object to this fundamental truth, "Animal Farm" meteorology also states that "Man serves the interests of no creature except himself". In a (dystopian) world, this quote refers to a common and inherent characteristic among all people. Likewise, cyclones share a set of features common to all cyclones. All cyclones intensify to some degree, converting APE to KE to achieve their very own intensification. It is this fundamental commonality among cyclones that we aim to describe in our manuscript, and how this process connects the theory of the general circulation with synoptic dynamics.

For our climatological analysis we specifically focus on a relatively small selection of deep Gulf Stream cyclones to reduce the variability in our composite. Although we don't expect differences in the fundamental character of APE conversion between cyclones (as also supported by our climatology), we agree that it would be interesting to compare cyclones developing in different regions, following different life-cycles or with a different amount of precipitation formation and by that diabatic heating. In fact, we believe that local APE could be a useful metric for the analysis of the atmospheric states giving rise to cyclogenesis more generally. For instance, how is APE supplied to cyclones outside the oceanic storm tracks? And does the absence of APE inhibit cyclone growth? More fundamentally, we think that this question links to the comparison of local APE and baroclinicity, as we have discussed in our outlook section.

28. Line 325: Same question as previously with regard to how general your conclusions are.

Given our climatological analysis in the following section, we are confident that our results are sufficiently general. We analyzed the APE tendency in CAO environments over the Gulf Stream region in the wake of intense extratropical cyclones. These conditions are known to be associated with the largest surface fluxes climatologically (e.g. Papritz and Spengler, 2017). In any other region, it is reasonable to assume that the surface fluxes in the wake of cyclones would be smaller, and hence the diabatic tendency due to those surface fluxes would be even smaller as well. Therefore, their significance would be further diminished, which supports our conclusion.

29: Lines 316-327: What about cyclone families? Lots of cyclones are a part of cyclone families. With respect to cyclone-related APE changes, might there be any relationship between cyclone family duration and overall APE changes?

Cyclone families are a very interesting case to consider. In fact, both cyclones 1b and 2b are frontal cyclones, forming cyclone families with cyclones 1a and 2a, respectively. In both cases, we observe and describe how the secondary cyclogenesis maintains the APE within the storm track by (re)intensifying the upper-level trough. Even though this is an interesting aspect, the focus of the manuscript lies on Gulf Stream cyclones to minimize variability within our climatological analysis.

But we can speculate on the impact of cyclone families on APE changes. We would expect a secondary cyclone to have less impact on overall (i.e. globally integrated) APE than a primary cyclone, since the primary cyclone and its baroclinic interaction with upper levels is typically setting up an intensified large-scale circulation. It is this environment that then

promotes secondary cyclogenesis. This argument is supported by the fact that most APE is converted in the upstream storm tracks (Novak & Tailleux, 2018), whereas secondary cyclogenesis mostly happens in the downstream parts of the storm tracks (Priestley et al., 2020).

30. Figure 9: This figure, comprised of 12 panels, is very difficult to read and assimilate. It is also insufficiently labeled (e.g., orange contours on panels e-h). Might four, four-panel figures work better….and not overwhelm readers? Failing that, perhaps include an option to choose magnifying glasses with the journal when reading this article?

On one hand, we agree that the size of the figures might be a problem for the readers. On the other hand, we regard it as important to show the figures together to
    1) show the life cycle of the cyclone
    2) show how APE and its tendency compare to the PV field
    3) show how the total APE tendency is split into its diabatic and adiabatic contributions
Separating the figure into smaller figures would make this comparison considerably harder. To make the figure more accessible, we will additionally provide larger versions of the figure in the supplement with 2x4 panels. Also, we will improve the labeling of the contours.

31. Text supporting Fig. 9 is a big help in sorting through the overly small figures. For example, the conclusions mentioned on lines 355-358 are well supported by the individual panels in Fig. 9. Likewise, for panels (e-h) and (i-l).

Thank you!

32. Do the very small differences between panels 9n and 9o reflect a smaller time difference and/or the rotation of the precipitation shield around the composite cyclone?

We think that the rotation of the precipitation band and with that the diabatic tendency between panels 9n and 9o reflects the establishment of a distinct cold front in panel 9o.

33. How sensitive are the results shown in Fig. 10 to the domain size over which the calculations were made?

Thank you for bringing up the sensitivity of our analysis. As we discuss in the text, the main factor influencing this sensitivity is the fact that the positive APE tendency is mostly found close to the center of the cyclone, whereas the negative APE tendency can be located further away in later stages of the cyclone life cycle. We think that the chosen radius adequately addresses this issue while still reflecting the dynamics of the synoptic system. To clarify this, we will recompute the spatial integral for different radii and include the resulting figures in the supplement.

34. There is little if any downward extrusion of PV in figure panel 11a. Is this an artifact of the cross section location choice and/or does it reflect that at the time the cyclone maximum intensification that the PV is mostly "used up" (so to speak)?

This is indeed an artifact of the choice of the cross section location. Figure R2.6 shows the cross section through the center of the cyclone at the time of maximum intensification (same

as Fig. 11). From this perspective the PV tower is easily recognizable. However, we lose the information on the vertical structure of the APE depletion in the cold sector of the cyclone in Fig. 11b.

[Figure]

Fig. R2.6: Cross sections of cyclone-centered composites at the time of maximum intensification. (a)-(c) are meridional cross sections and (d)-(f) are zonal cross sections through the cyclone center. (a),(d) PV (shading) and horizontal wind speed (blue; every 10 m/s. (b),(e) adiabatic APE tendency (shading) and APE density (orange contours; every 500 J/kg). (c),(f) diabatic APE tendency. Potential temperature is indicated by grey contours (every 6 K).

35. There needs to be additional labeling of the lines in Fig. 11 (e.g., panels e and f are devoid of contour labels). Additional labels would make it easier for a reader to follow the text better on lines 425-431.

Thank you for pointing out the missing contour labels. We will add additional contour labels for both the wind speed and APE tendency, as well as some contour labels for the potential temperature.

36. There is a factor 4 difference in the vertical scale for Figs. 12c and 12d. Consider using the same vertical scale for these two figure panels to facilitate a better understanding of the difference in the adiabatic and diabatic APE tendencies?

As discussed for your point 25, we believe that applying the same scale to both panels would make it very hard to decipher the evolution of the diabatic tendency. But because we

agree with your argument, we will use half the scale to facilitate the interpretation of the differences.

37. Discussion surrounding Fig. 12 is excellent.

Thank you!

38. Nice synthesis in section 5.1. Would be to comment how the results from a sample of 285 intense NATL cyclones might differ from a much larger overall sample of cyclones?

Thank you for pointing this out. We agree that the variability among different classes of cyclones and/or different regions would be very interesting to investigate and we will include this aspect in the outlook section.

39. Lines 479-485: I fully agree with this interpretation from a synoptic-dynamic/weather map perspective.

Thank you.

40. Lines 485-495: How might your results differ if you distinguished between cyclogenesis associated positively tilted versus negatively tilted troughs? Typically, positively (negatively) tilted troughs are associated with the strongest jet in the southwesterly (northwesterly) flow downstream (upstream) of a tilted trough axis.

This is an interesting point. Our case study analysis contains cyclones with a northwesterly jet upstream of the main trough axis (1a and 2a), and the secondary cyclones with a southwesterly jet downstream of the main trough axis. While our composite study focuses on the upstream storm track region, we think it mostly contains cyclones of the former case. However, the character of APE conversion remains unchanged of the orientation of the upper-level jet changes, i.e. the ageostrophic circulation on the poleward flank of the jet stream will still consume APE, as the poleward flank is still facing towards the trough axis and with that towards the APE reservoir.

41. Lines 505-513: I am still pondering this conclusion because there is so much variability between individual cyclones. Might comparing cyclones in NW versus SW flow possibly be helpful?

We think that this question largely falls back onto different categories of cyclones, which opens up many avenues of new research, e.g. how do the different roles of latent heat release in diabatic Rossby waves, frontal cyclones and primary cyclones compare in terms of the APE framework. However, this comparison lies outside the scope of our analysis. Since we selected very strong Gulf Stream cyclones, for which the role of diabatic heating can be assumed to be similar (e.g. Binder et al. 2016), we believe that our conclusions are well supported.
However, we agree that the reader might wonder about the variability of the cyclones we consider for our composite. We don't think this aspect is discussed sufficiently in the choice of the cyclone tracks. Therefore, we will add more clarification on why we think that our choice of cyclone tracks minimizes variability within our composite.

42. Lines 520-522: Good point. I had never thought this concept though previously in the depth needed to better appreciate the differences.

Thank you!

43. Lines 533-541: Interesting point. Would it be helpful to distingush between high versus low dynamical tropopause storms?

This is an interesting idea. For instance, it would be interesting to see whether high dynamical tropopause storms convert similar amounts of APE as low dynamical tropopause storms, and how this affects their intensification. We will include this thought in the outlook section.

44. Lines 542-546: It might be helpful to distinguish between early season cyclones(when the troposphere is still relatively warm and the dynamic tropopause is still relatively high), from midseason cyclones (when the baroclincity is strongest), and late season cyclones (when warm sector CAPE is becoming a factor in cyclone development and intensification.

In general, we think that the decreased differential heating of Earth and the weaker cyclone intensities outside DJF would imply weaker APE conversions for early and late season cyclones. But this then raises the question of the conditions surrounding very intense early and late cyclones and where their APE originates from. As you point out, this might also relate to the previous point and the height of the dynamical tropopause. Lastly, the role of warm sector CAPE would then also point towards a shift in energy generation from adiabatic APE consumption to diabatic sources, which would be interesting to investigate in more detail in future work. We add a discussion of this point to the outlook section.

**References:**

Binder, H., M. Boettcher, H. Joos, and H. Wernli, 2016: The Role of Warm Conveyor Belts for the Intensification of Extratropical Cyclones in Northern Hemisphere Winter. *J. Atmos. Sci.*, **73**, 3997–4020, https://doi.org/10.1175/JAS-D-15-0302.1.

Bowley, K. A., J. R. Gyakum, and E. H. Atallah, 2019: The Role of Dynamic Tropopause Rossby Wave Breaking for Synoptic-Scale Buildups in Northern Hemisphere Zonal Available Potential Energy. *Mon. Wea. Rev.*, **147**, 433–455, https://doi.org/10.1175/MWR-D-18-0143.1.

Grossman, R. L., & Betts, A. K. (1990). Air-sea interaction during an extreme cold air outbreak from the eastern coast of the United States. *Mon. Wea. Rev.*, 118, 324–342, https://doi.org/10.1175/1520-0493(1990)118<0324:AIDAEC>2.0.CO;2

Federer, M., L. Papritz, M. Sprenger, C. M. Grams, and M. Wenta, 2024: On the Local Available Potential Energy Perspective of Baroclinic Wave Development. *J. Atmos. Sci.*, **81**, 871–886, https://doi.org/10.1175/JAS-D-23-0138.1.

Fletcher, J., Mason, S., & Jakob, C. (2016). The climatology, meteorology, and boundary layer structure of marine cold air outbreaks in both hemispheres. *J. Climate*, **29**, 1999–2014. https://doi.org/10.1175/JCLI-D-15-0268.1

Novak, L., and R. Tailleux, 2018: On the Local View of Atmospheric Available Potential Energy. *J. Atmos. Sci.*, **75**, 1891–1907, https://doi.org/10.1175/JAS-D-17-0330.1.

Papritz, L., & Schemm, S. (2013). Development of an idealised downstream cyclone: Eulerian and Lagrangian perspective on the kinetic energy. *Tellus A: Dynamic Meteorology and Oceanography*, **65**(1). https://doi.org/10.3402/tellusa.v65i0.19539

Papritz, L., and T. Spengler, 2017: A Lagrangian Climatology of Wintertime Cold Air Outbreaks in the Irminger and Nordic Seas and Their Role in Shaping Air–Sea Heat Fluxes. *J. Climate*, **30**, 2717–2737, https://doi.org/10.1175/JCLI-D-16-0605.1.

Priestley MDK, Dacre HF, Shaffrey LC, Schemm S, Pinto JG. The role of secondary cyclones and cyclone families for the North Atlantic storm track and clustering over western Europe. *Q J R Meteorol Soc*. 2020; **146**: 1184–1205. https://doi.org/10.1002/qj.3733

Vannière, B., A. Czaja, H. Dacre, and T. Woollings, 2017: A "Cold Path" for the Gulf Stream–Troposphere Connection. *J. Climate*, **30**, 1363–1379, https://doi.org/10.1175/JCLI-D-15-0749.1

---

## Author Response (AR1)

Author reply for

Synoptic perspective on the conversion and maintenance of local available potential energy in extratropical cyclones

by Marc Federer, Lukas Papritz, Michael Sprenger, and Christian M. Grams

November 8, 2024

In the following, the comments of the reviewers are shown in **blue** and our replies in **black**.
Mentioned references are listed at the end of the document. Line and figure numbers refer to the revised manuscript.

**Review by Kevin A. Bowley**

Review of WCD-2024-2112: Synoptic perspective on the conversion and maintenance of local available potential energy in extratropical cyclones

Overview: This manuscript explored a relatively new technique of quantifying local available potential energy (APE) and applying it to cyclone dynamics for strong extratropical winter cyclones in the North Atlantic basin. Their results show a close connection between the extratropical tropospheric baroclinic zone ("polar front") and a strong gradient in local APE. They also assessed the movement of air parcels using trajectory analysis to explore the tendencies in APE, demonstrating strong APE to KE conversion upstream of a trough axis (advecting into the trough base) and strong KE to APE conversion downstream of a trough axis (ascending generally through the warm conveyor belt). I commend the authors on a well written manuscript that was generally easy to follow. I do have some suggestions for the clarity of the presentation in several places, some changes/new figures that I believe would help aid a reader in interpreting the results, and questions about the sensitivity of some of the analysis to choices in methodology. Thus, I believe this is on the border between a major and minor revision.

Thank you for your positive evaluation of our manuscript. We are grateful for your constructive and insightful feedback, and we highly appreciate the time you put into this review. We believe the suggested changes will significantly improve our manuscript and we hope the revised version will match your expectations.

General Comments:
1. Description of local APE and reference state: I appreciate the challenges of word limits for manuscripts and the necessity of referring readers to prior literature for specifics on a methodology. That said, I would suggest that the authors add a bit more detail/clarity in sections 2.2 and 2.3 to help the readers understand the method a bit more clearly given the relatively new methods being applied here. For example, I found the description of how the reference state was computed (lines 143-145) challenging to understand/visualize, thus making interpretation of the APE figures later in the paper challenging in turn. It was unclear to me at times whether we were comparing a specific parcel to its specific reference state, or a column, or otherwise. Lastly – it may be beneficial up-front in this manuscript (rather than at the end) to discuss local APE in light of other metrics used for assessing energy for storms (eg. Eady growth rate/baroclinicity) and resistance to vertical velocity (eg. static stability).

Thank you for pointing this out. Given that local APE is a concept that is not widely used, we agree that discussing the construction of the reference state and what local APE "means" in more detail already in the introduction and methods sections eases the understanding of the paper. In the introduction, we have added a paragraph comparing baroclinicity and APE (L47-50) and another paragraph discussing the reference state (L70-74). Static stability is implicitly included in this discussion because common frameworks of baroclinicity (e.g. Eady growth rate or isentropic slope (Papritz and Spengler, 2015)) also quantify the effect of static stability.

We have also extended the discussion of how we compute the reference state, which should clarify what we mean by the reference state and how a single air parcel is compared to this reference state. In addition, we have added a schematic of how the reference state is constructed to the supplement (Fig. R1.1).

[Figure]

*Fig. R1.1: Schematic illustration of the computation of the reference state. (a) Meridional cross section of an idealized baroclinic zone composed of air parcels, represented by boxes. Color indicates potential temperature of the air parcels, increasing from blue to red. (b) Sorting of air parcels by increasing potential temperature. (c) Construction of the reference state from sorted air parcels illustrated by a section through the solid Earth (hatched) and the atmosphere (color, not to scale). The dashed lines indicate Earth's rotation axis and the equatorial plane (0° latitude). The atmosphere is assumed to be shallow such that each circle has the same circumference.*

2. Discussion of APE changes linked to trajectories : I appreciated the approach taken here by tracking parcels through the troposphere. However, I found the discussion surrounding

figure 3 challenging. I could understand (eventually) how the method discussed in line 216-217 was applied to the analysis, but it made the interpretation really challenging. I can see the idea behind wanting to see how much APE is depleted from the starting location, but it made a) the interpretation of what the actual change in column APE was at a location really difficult, and b) made comparison to figure 9 really challenging. A more clear motivation here may be helpful, as well as clarification here (or at figure 9) about the differences in what we're looking at. One other approach could be to actually show either APE or dAPE/dt for your trajectories (akin to how you have pressure shown in figure 4). Further through this section, it would be helpful to really quantify the contribution due to omega/diabatic heating relative to the respective efficiency terms. Perhaps a box-and-whisker plot of distribution of the efficiency terms for the trajectories at different levels would help? There is good qualitative discussion here (see lines 264-266), but a bit more time spent on the quantitative side would really help. Lastly, your line 266-268 doesn't quite make sense to me r.e. linking the latent heat release to the second max of APE loss over the Canadian Arctic.

Thank you for bringing up the difficulty of interpreting Fig. 4 and your suggestions for improvement. We agree that the methodology behind Fig. 4 is not straightforward. Therefore, we have included Fig. S5 in the supplement, which shows the evolution of APE, APE tendency and the efficiencies along trajectories from different starting levels. This illustration also helps to understand the second maximum of APE loss over Eastern Canada in Fig. 4c. First, we can consider trajectories descending from 650 hPa, which experience diabatic APE generation in the first 20 hours of their descent due to radiative cooling (Fig. S5c). After 20 hours, however, the diabatic APE tendency turns negative, due to a positive heating rate and a negative APE tendency (Fig. S5f). So why are those trajectories heated? Initially we suspected that the trajectories start to ascend again at this point, which would lead to latent heat release. But as Fig. S5a shows, the trajectories likely do not ascend. Instead, they enter the boundary layer. Therefore, the heating rate must originate from surface heat fluxes and mixing in the boundary layer. This means that the second maximum in diabatic APE loss over Eastern Canada is due to air-sea interaction. We discuss this on L300-305.

We also agree that the difference in methodology between Fig. 4 and Fig. 10 needs to be highlighted more. To better motivate Fig. 4 and to also facilitate the comparison of the case study to the climatology we have included Fig. 3 in section 3.2 before Fig. 4. Figure 3 shows the vertically-integrated instantaneous APE tendencies on February 19, 12 UTC. From this figure we can clearly see that the APE change mostly occurs on the edge of the polar APE reservoir. However, in comparison to Fig. 4, instantaneous APE tendencies are noisier than the Lagrangian APE tendencies and exhibit various dipole structures, which, when averaged along trajectories, are largely smoothed out. In fact, the smoothing of Lagrangian APE tendencies facilitates linking the APE tendencies to the large-scale flow structures. Therefore, in our view showing the Lagrangian tendencies in Fig. 4 is preferable and better illustrates how the baroclinic development depletes the APE reservoir.

[Figure]

*Fig. S5: APE evolution along selected 48 h descending forward trajectories from Fig. 5. Shown are averaged time series for trajectories descending from 650 hPa (left column) and 450 hPa (right column). Shown are (a),(b) APE density (black) and pressure (grey); (c),(d) adiabatic (blue) and diabatic (red) APE tendency; (d),(e) vertical pressure velocity (black) and adiabatic efficiency (blue); (f),(g) diabatic heating rate (black) and diabatic efficiency (red, dimensionless).*

[Figure]

*Fig. 3: Vertically integrated APE tendency on 19 Feb UTC 12 (color shading). Shown are the (a) net change, (b) adiabatic contribution and (c) diabatic contribution. Also shown are the vertically integrated APE distribution (black contours; same intervals as Fig. 1) and the mean sea-level pressure (grey contours; every 10 hPa). The location of cyclone 1a is indicated.*

3. Dynamic interpretation/explanation of results: I appreciated that there were flavors of QG forcing, jet ageostrophic circulations, and PV thinking infused throughout the paper. That said, I think there were a few points where the discussion could've been taken into more depth and/or the QG/ageostrophic/PV discussions could've been more unified (eg. it's three perspectives to explain rising/sinking motion). For example, there was a lot of discussion on jet entrance/exit regions (which were at times not clear to me as entrance/exits rather than poleward flanks of the jet), but much less discussion on the role of curvature (where both the trough and upstream ridge can instigate ageostrophic circulations resulting in +/- omega), transverse thermally (in)direct circulations along jet streaks, and whether there was temperature advection occurring across the jet (which shifts where we see +/- omega in response to ageostrophic responses). Much of this could also be discussed from the perspective of PV theory (which emerges late in the paper but could've been interwoven throughout).

Thank you for this observation. We agree that we neglected the role of curvature in instigating ageostrophic vertical circulations in the discussion of our results. Your minor comments already added many improvements in this regard. Additionally, we have added a discussion of the different influences on QGomega, including temperature advection, to L533-537:
"In addition to the curvature associated with an upper-level PV anomaly, other synoptic environments can contribute to ageostrophic vertical circulations and influence APE tendencies. For instance, the thermally direct transverse circulation in the entrance region of a jet streak is expected to convert APE into KE, while the thermally indirect transverse circulation at the jet exit would convert KE back into APE. Furthermore, temperature advection across the jet can influence the location of the ageostrophic vertical circulation (Keyser and Shapiro, 1986), thereby affecting the associated APE tendencies."

While we believe this effect is very interesting and highly relevant in synoptic meteorology, we do not intend to further distinguish between the effects of confluence/diffluence, curvature and temperature advection on QGomega. Our overarching goal is to extend the planetary perspective of APE to synoptic scales, and then to qualitatively relate it to classical

theories of synoptic meteorology. Providing a quantitative account of the different QG omega forcings would lead too far away from this qualitative goal. Thus, we plan to include a discussion of the different possible effects on QGomega, including temperature advection, without analyzing in detail which effect dominates for our case study.

We'd also like to highlight that the discussion of jet entrace/exit regions in the Discussion section does not aim to explain the mechanisms of QGomega, but to link the climatological distribution of KE (i.e. the jet) to the observed distribution of the APE tendency within individual cyclones.

4. Inclusion of CAOs: Though I found the inclusion of the CAO case and composite interesting, I wasn't really sure how much it contributed to the paper. This may benefit from either removing it, or better motivating why you're including it.

Thank you for this comment. We agree that the discussion of CAOs stands a bit isolated in the manuscript. This part is motivated by the observation of Novak & Tailleux (2018) that local APE is diabatically consumed by upward surface sensible heat fluxes and the release of latent heat in convection over the western boundary currents. Strong upward fluxes are generally related to cold air sweeping across the Gulf Stream front towards the warm waters (e.g., Grossmann and Betts, 1990; Fletcher et al. 2016), which is why we focus on CAOs to study the diabatic APE tendency. We have rephrased to put more emphasis on the relationship between surface fluxes and the APE tendency. For example, we changed the title of section 4.2 to "The role of surface fluxes for the diabatic APE tendency". Additionally, we have better motivated our focus on CAOs as the synoptic condition within which we adress this question.

Specific comments:
• Lines 53-54: Please clarify/elaborate on the 'why' of APE being in the polar middle and upper troposphere and how it is advected (eg. is it a material quantity?).

We have refered to the reference state here to explain why APE is located in the polar middle and upper troposphere and rephrase as follows (L58-60):
"In particular, they find that APE is mainly located in the polar middle and upper troposphere, where the air is colder than its reference state. They also showed that APE is advected from this polar reservoir into the storm tracks."

We also agree that it would be helpful to clarify what is meant by advection of APE. However, we think this is more suitable for Sec. 2 (Data and Methodology), where we formally introduce the equation for the APE tendency. We have added the following to Sec. 2.2 (L153-154):
"Terms (I)-(III) present sources and sinks to local APE. Therefore, APE is conserved along adiabatic flow only if $\omega$ is zero."

• Lines 61-62: For local APE, is it only accelerations/decelerations? For example, do all air parcels in the warm conveyor belt that contribute to a positive APE tendency explicitly have a deceleration occurring?

The acceleration due to a conversion of APE to KE by an air parcel is not necessarily experienced by this air parcel itself. This can be seen from the kinetic energy equation (Novak and Tailleux 2018; Federer et al. 2024):

$$\frac{D}{Dt}\frac{\mathbf{V}^2}{2} + \nabla_h \cdot (\Phi'\mathbf{V}) + \frac{\partial(\omega\Phi')}{\partial p}$$
$$= -\{\alpha(\theta, p) - \alpha[\theta_r(p, t), p]\}\omega + \mathbf{F} \cdot \mathbf{V}, \qquad (5)$$

Here, the first term on the right hand side in curly brackets represents the conversion between APE and KE and the second term is dissipation of KE due to friction. While the conversion of APE to KE is the only source term in this equation, there is also a material change of KE due to the second and third terms on the left hand side, which represent geopotential fluxes. When volume-integrated globally, these fluxes become zero. Locally, however, these terms are generally not zero and will redistribute the KE gained from APE conversion (Papritz and Schemm, 2013). Thus, an air parcel that loses APE, may not gain the equivalent amount of KE in case it is located in a region of divergent geopotential fluxes. We think that our wording might be misleading, which is why we have rephrased as follows (L66-67):
"This is consistent with a local notion of APE, where air parcels accelerate the flow by converting APE to KE, but also decelerate it by converting KE back to APE."

• Lines 110-112: Make two sentences – it's a bit hard to follow as written.

Thank you for the suggestion. We have rephrase accordingly.

• Section 2.2: Please include units throughout here. It may be helpful to also provide equations for specific density and your diabatic heating rate.

Thank you for this suggestion. We have included the equations for specific volume, the pressure velocity and the diabatic heating rate. We think that the units will be more clear from those equations.

• Line 170: From your figure 1, it appears less that an upper level trough is developing rather than propagating in from the west. Please be sure that it is developing (rather than propagating) if you use the term 'upper-level trough forms'.

Thank you for this observation. We plotted the PV field on 315 K for an extended time window centered on the North Atlantic to clarify (Fig. R1.3). We agree that there exists an upstream feature over the western United States (Fig. R1.3a). As cyclone 1a intensifies, this PV trough remains stationary and does not propagate into the North Atlantic (Fig. R1.3b). Given the strong elongation of the trough in the North Atlantic (Fig. R1.3c), we think it's fair to say that this trough formed in the North Atlantic concurrent with the intensification of cyclone 1a. However, we agree that the APE located over the Great Lakes in Fig. 1a appears to propagate into the North Atlantic. This is in line with our argument that APE is supplied to the storm track through advection.

[Figure]

*Fig. R1.3: Isentropic PV on 315 K (color shading) for (a) 18 Feb UTC 12, (b) 19 Feb UTC 12 and (c) 20 Feb UTC 12. Also shown is the mean sea-level pressure (grey contours; every 10 hPa). The location of cyclone 1a is indicated.*

• Lines 171-172: I think you need to spell out more how the collocation of high APE with the DT is demonstrating the connection to the large scale circulation (I don't disagree with you, but it needs to be more clearly demonstrated).

We agree and have rephrased as follows (L189-193):
"High APE values are clearly co-located with the trough extending into the North Atlantic. This points to a strong link between the upper-level PV pattern and the distribution of APE. Because the distribution of PV governs the large-scale circulation, APE is likely linked to the large-scale circulation as well, as will be explored in the following sections."

• Lines 179-180: Do we know that this is low APE advection vs. a time tendency due to conversion? In theory, one could compute APE advection to show this explicitly. The same goes for lines 183-184.

On one hand, we agree that at this point in the manuscript the negative APE tendencies in the warm sector of cyclone 3a could also be interpreted as a local consumption of APE. However, local consumption alone would appear to be somewhat counterintuitive, because we typically observe warm-air advection ahead of a developing cyclone, and we show that APE values to the south of cyclone 3a are not particularly high. Therefore, we think that our claim that low-APE air is advected to this location is well supported. On the other hand, the results following this section also show that the ascending air ahead of cyclone 1a generates APE due to the positive adiabatic efficiency, which is consistent with the southerly origin of the air. It is reasonable to assume that this is also the case for cyclone 3a.
Indeed it is possible to compute the advection of APE explicitly. However, we don't show the advection on purpose. In our opinion, the trajectory analysis yields more insight into how APE is transported along with the flow and how the evolution of the APE is linked to the dynamical processes.

• Lines 202-204: This isn't really a stand-alone paragraph – please aim to elaborate or merge with another paragraph.

We agree and have merged this paragraph with the previous one.

• Lines 206-207: The cross-section does show the cold dome and therefore a troposphere-deep cold anomaly. It does not, however, imply cold air advancing into the midlatitudes. Consider providing different evidence or re-writing.

Considering the co-evolution of the trough and the cold-dome suggests that the development of the upper-level trough is typically associated with cold air advancing equatorward underneath the trough as opposed to local diabatic cooling. However, we agree that this cannot be concluded based on the cross section alone. Therefore, we have rephrased as follows (L226-228):
"The vertical distribution of APE shows that APE is concentrated in the middle and upper troposphere north of the midlatitude baroclinic zone. The upward bending of isentropes by the positive PV anomaly implies a dome of cold air beneath the trough, which in turn is associated with large APE since the air is colder than its reference state."

• Figure 1 cross-section lines/Figure 2: Consider given start/end point markers for your cross-sections (eg. A-A', B-B', C-C') to avoid any ambiguity about the direction of your cross-sections.

Thank you for this suggestion. We have added start and end point markers to Figs. 1 and 2.

• Lines 222-225: This may benefit from a schematic (this infuses with general comment 2 above).

Thank you for this suggestion. We think the figure we describe in response to your comment 2 will address this comment here too.

• Line 227, 228 (and elsewhere): Careful on geolocation descriptions. For example – the Canadian Arctic is a pretty expansive area. But it appears your focus in figure 3 may be more on Hudson Bay/Quebec/Labrador? Likewise, I wouldn't consider the Gulf Stream to be adjacent to the Canadian Arctic. A reference geographic map (or outline of the Gulf Stream region) may be helpful.

Thank you for pointing this out. We have added more precise location references, referring to either Northeastern or Eastern Canada.

• Lines 243-248: There's a large region of positive APE tendency south of Iceland that likely is playing a factor the in the cyclone evolution. It would be helpful to either look at trajectories/APE tendencies in that region as well, or to provide justification for not examining it.

Thank you for this suggestion. We indeed did not discuss this region of large positive APE tendencies. The reason for this is that this APE tendency is due to ascent in the warm sector of the preceding cyclone, which is located south of Iceland in Fig. 4. Therefore, this tendency is most relevant for the downstream cyclones, whereas here we are concerned with the Gulf

Stream region. We have rephrase slightly to put more emphasis on the fact that we are interested in the APE conversions associated with cyclone 1a (L273):
"For this purpose, we select two regions of particularly large APE tendencies in the vicinity of cyclone 1a (Fig. 4b)."

• Lines 258-260: I would find it helpful to see some discussion of this also in reference to the effiiency terms.

Thank you for this suggestion. We agree that this part would benefit from a discussion of the role of efficiency, in particular in determining the sign of the APE tendency. We have added a discussion of the efficiency as follows (L286-290):
"The rapid ascent of these trajectories is reminiscent of a warm conveyor belt (WCB), which is associated with strong latent heat release. Indeed, the region of negative diabatic APE tendency observed slightly downstream of cyclone 1a coincides with the ascending air parcels. This also implies that the thermal efficiency of the air parcels must be negative, which means that the air parcels are colder than their reference state. Therefore, latent heat release in the ascending air stream leads to a destruction of APE."

• Line 280: I'm not sure the line 'detaches from the band of downward …' is the best choice of phrasing. Please reword.

We agree and have reworded.

• Line 283: I'm not sure I would classify this as the entrance region of the jet. It may be better discussed as downstream of the upper-tropospheric ridge.

Thank you for raising this point. We agree that the location of the trajectory descent is better described as downstream of the upper-tropospheric ridge. However, the descent does not occur uniformly distributed along the jet. Therefore, we have reformulated as follows (L317-319):
"On 20 Feb 00 UTC (Fig. 6a) the majority of trajectories descend along the downstream flank of the upper-tropospheric ridge, predominantly on the poleward side of the jet axis, while few trajectories descend further south in the vicinity of cyclone 1a."

• Lines 290-291: Though the trough may play a role here, so too does the upstream ridge (contributing to QG forcing for descent).

We agree that the phrasing is misleading since the ageostrophic circulation is a consequence of the baroclinic wave as a whole. Since we want to simply emphasize the connection between APE conversion and the ageostrophic circulation, we have not mentioned the trough here.

• Figure 5: I found it challenging to differentiate the dark green (2-PVU) contour from the black (300 hPa wind speed) contours. Please consider a slightly more contrasting color choice. I would also add a 'L' to represent your surface low position in both panels.

Thank you for your suggestions. We have chosen a more contrasting color for the 2-PVU line and have marked the location of the surface cyclone.

• Lines 322-323: I would be careful about making this statement. The surface cyclone can influence the upper-troposphere and vice versa (eg. cyclone growth positive feedback cycle), and your cross-section shows a troposphere-deep cyclonic cold anomaly and circulation.

Thank you for pointing this out. We agree that the baroclinic interaction between upper levels and the surface cyclone is inherent to baroclinic instability. Therefore, we think a different phrasing is needed to express our conclusion that APE conversion does not happen at lower levels within the cyclone, as for example meridional heat fluxes do, but is governed by the upper levels. We have rephrased this bullet point as follows (L356-360):
"The vertical motion induced by the interaction of the intensifying surface cyclone and upper-level trough lead to both APE consumption and production. APE is primarily consumed by the descent along the western flank of the trough, while APE is produced by the ascent ahead of the trough. This pattern of ascent and descent is consistent with QGomega forcing by upper levels. Therefore, the major part of APE conversion takes place through the circulation in the mid to upper levels, and not within the surface cyclone itself."

• Figure 8: Aim to make figure 8b seems more its own independent panel (maybe a box around it?). I didn't notice it even when referenced in the text.

We have increased the size of the boxplot to make it stand out more.

• Line 338: How many times steps overlap as being both in the delta-Pmax category and the dmax category?

Each life cycle stage is defined as only a single time step along a cyclone track. As the maximum depth implies that the cyclone does not deepen further, delta-Pmax always occurs before dmax. Hence, there is no overlap. Also for Fig. 11 and the corresponding analysis, there is no overlap between the 12-hour windows used.

• Line 341-342: Again, not sure this is the jet entrance region. I suspect curvature dynamics are playing more of a factor.

Thank you for pointing out the role of curvature here. We agree that the dipole pattern in the QGomega forcing aligns well with the forcing expected from an upper-level trough. At this point, however, we prefer not to speculate on the reason of why we observe the QGomega dipole, as our focus lies on the APE tendencies. Therefore, we descided not to mention the jet entrance region here.

• Line 345: I think you can lean into PV thinking for the occlusion stage here (eg. evidence of a stacked cyclonic PV anomaly circulation from surface to dynamic tropopause, along with the surface low receding into the cold PV hook.

Thank you for this suggestion. We have mentioned the vertical stacking of the cyclonic circulation to better incorporate the dynamics of the occlusion stage.

• Line 351-352 (and elsewhere in this section): Consider adding some values to the

Thank you for this suggestion. We agree that this is helpful and have included some of the amplitudes for the APE tendencies observed in Fig. 10.

• Lines 352-353: Is this PV field shown anywhere? It may be beneficial to include.

We refer to the PV field at the time of maximum depth shown in Fig. 10c. Downstream of the ridge, the curvature of the PV field suggests a new trough.

• Lines 372-374; 399-405: Consider bringing in some discussion of the poleward movement of your composite when discussion the radiative cooling of the free atmosphere.

Thank you for this suggestion. We have addd the mean locations of the composites in Fig. 9 and also included the following discussion (L417-419):
"The diabatic APE generation due to radiative cooling increases between the times dp_max and d_max when the poleward displacement of the mean composite location is largest (Fig. 9). This means that the environment of the cyclone becomes colder with a more negative thermal efficiency. Thus, the radiative cooling leads to larger diabatic APE generation."

• Lines 379-380: How sensitive is your analysis to the region chosen here? I can understand 2000 km from a synoptic scale perspective, but does this strongly impact the interpretation of storm-scale contributions?

Thank you for bringing up the sensitivity of our analysis. As we discuss in the text, the main factor influencing this sensitivity is the fact that the positive APE tendency is mostly found close to the center of the cyclone, whereas the negative APE tendency can be located further away at later stages of the cyclone life cycle. We think that the chosen radius adequately addresses this issue while still reflecting the dynamics of the synoptic system. To illustrate the sensitivity, we have recomputed the spatial integral for different radii and included the resulting figures in the supplement. At a smaller radius of 1000 km (Fig. S8), we notice that the negative adiabatic APE tendency is not captured well, which results in a positive adiabatic APE tendency. Similarly, the diabatic APE tendency mainly captures the concentrated negative tendency due to latent heat release, while the weaker positive tendency due to radiative cooling is not well captured. At a bigger radius of 2500 km (Fig. S10), the life cycle of the cyclone is more muted compared to Fig. 11 because more noise is included.

[Figure]

*Fig. S8: (a) Distribution of minimum mean sea-level pressure of the 285 selected cyclones, averaged +/- 6h around the four life cycle stages. The (b) net, (c) adiabatic and (d) diabatic APE tendency is integrated in a 1000 km radius around the cyclone center within a +/- 6 h window around the four life cycle stages. Whiskers show the 10th-90th percentile range.*

[Figure]

*Fig. S10: Same as Fig. S5, but for an integration radius of 2500 km.*

• Lines 415-416: Why not take your cross-section through the center of the surface cyclone? Here your baroclinic zone is south of the surface cyclone simply because the trough is further south to the west of the cyclone (as you noted).

We take the cross sections to best capture the vertical structure of the strongest APE tendencies. Therefore, they cut through maxima in the negative and positive APE tendencies. Figure R1.4 shows the same as Fig. 12 but through the center of the cyclone. Here, the meridional cross section of the adiabatic APE tendency (Fig. R1.4b) only shows the positive tendency at the cyclone center, while we want to focus on the structure of the negative tendency in the cold sector of the cyclone.

[Figure]

*Fig. R1.4: Cross sections of cyclone-centered composites at the time of maximum intensification. (a)-(c) are meridional cross sections and (d)-(f) are zonal cross sections through the cyclone center. (a),(d) PV (shading) and horizontal wind speed (blue; every 10 m/s). (b),(e) adiabatic APE tendency (shading) and APE density (orange contours; every 500 J/kg). (c),(f) diabatic APE tendency. Potential temperature is indicated by grey contours (every 6 K).*

• Lines 425-426: I found this sentence unclear. Which positive adiabatic APE tendency?

Thank you for pointing this out. We agree that the sentence is unclear because we don't refer to this positive adiabatic APE tendency before. We have reformulated as follows (L475-476):
"Both the positive and negative adiabatic APE tendencies extend through most of the column, peaking around 500 hPa."

• Section 4.2: Please provide information on how many CAO cases were involved.

Roughly 10% of the 285 cyclone tracks (27) were not associated with a CAO. We have included this information in the text.

• Lines 455-456: I think you need to reverse the order here (eg. trough leads to surface cyclone, not vice versa).

Our intention here was not to imply causality, but to describe the interaction between the cyclone and the trough as the mechanism that strengthens both the trough and the APE beneath it. We have improved the phrasing to better reflect this as follows (L506-509):

"Intense Gulf Stream cyclones typically originate at the edge of the polar APE reservoir. The baroclinic interaction of those cyclones with upper-level troughs induces a cold anomaly in the mid to upper troposphere beneath the trough."

• Lines 469-470: Your zonal cross section implies a troposphere-deep circulation, so this feels a bit misleading as written to me.

Thank you for pointing this out. Indeed the word 'aloft' is misleading here. We have rephrased as follows (L520-522):
"We observe complex patterns of APE tendency, largely following the trough and ridge patterns aloft. These ensue from the  ageostrophic circulation induced by large-scale flow features such as upper-level troughs and ridges and their baroclinic interactions with the lower troposphere."

• Lines 485-492: You mentioned the Bowley et al. 2019 paper in the introduction – though their results are a bit hard to interpret relative to yours (given the global zonal APE vs. local total APE perspective), you may want to include some of their interpretation here. For example, they found a dominant mechanism for synoptic scale APE increase to be ascent on the poleward flank of the wave guide in the exit region of the North Pacific jet, which fits well to the results of Koch et al. 2006 and your adiabatic generation interpretation here.

Thank you for your suggestion. We agree that the adiabatic generation of APE through ascent fits well to the findings reported in Bowley et al. 2019. However, Bowley et al. identified diabatic heating and cooling as the dominant mechanism of increasing the meridional temperature gradient across the baroclinic zone in the Pacific and with that increasing zonal APE. Our study suggests the possibility of a different pathway to APE build-ups: Instead of diabatic generation, a dominance of positive adiabatic APE tendency could also lead to an APE build-up. This would correspond to a dry-dynamic global deceleration of the flow, converting KE to APE through ascent of cold air.
Although the discussion of APE build-ups and also collapses is very interesting, we think it would lead too far away from our conclusion about the climatological APE tendency patterns. Therefore, we discussed Bowley et al. 2019 just in the context of the jet exit patterns as follows (L546-549):
"Novak and Tailleux (2018) also find that the conversion from mean KE to mean APE is largest in the downstream part of the North Atlantic storm track, where the exit of the climatological jet stream is located (Koch et al., 2006). This is consistent with Bowley et al. (2019), who identified ascent on the poleward flank of the wave guide in the exit region of the North Pacific jet stream as a mechanism for zonal APE build-up."

• Line 517-520: I think your points here would be beneficial to appear earlier in the manuscript when introducing the local APE framework to help unify the global vs. local perspectives.

Thank you for your suggestion. We agree that the structure of the reference state and its implications for the distribution of local APE should be discussed in more depth in the introduction and have added the following (L70-74):

"First of all, the reference state based on an adiabatic rearrangement of the whole real atmosphere differs from the reference state computed from a symmetric baroclinic channel due to the spherical geometry of Earth. In the baroclinic channel, there is approximately the same amount of APE on the equatorward and the poleward side of the baroclinic zone (Federer et al., 2024). In contrast, the mass of air equatorward of the midlatitude baroclinic zone on Earth is much greater than on the poleward side, which implies that the local APE density is larger near the poles than near the Equator (Novak and Tailleux, 2018)."

Technical corrections:
• Line 75: Change 'and to contribute' to 'and can contribute'.
• Line 98: I'm not sure 'relies' is the right word choice for your data here.
• Line 167: Please write out 'potential vorticity unit' the first time you define PVU.
• Line 343: Please add 'cyclonic' between 'upper-level' and 'Rossby wave breaking'
• Line 430: Should this read 'ascent leading to the positive adiabatic APE tendency'?

Thank you for your suggestions. We fully agree and have corrected the text accordingly.

**Review by Lance F. Bosart**

A) Overview: Federer et al. (2024): Synoptic Perspective on the Conversion and Maintenance of Local Available Potential Energy in Extratropical Cyclones

The authors are to be commended for preparing a well-evidenced story and well-written story on the often challenging concept of atmospheric energetics in general and available potential energy in particular. I learned something from reading this paper.

Thank you for your positive evaluation of our manuscript. We are grateful for your constructive and insightful feedback, and we highly appreciate the time you put into this review. We believe the suggested changes will significantly improve our manuscript and we hope the revised version will match your expectations.

B) Specific comments on Federer et al. (2024): Synoptic Perspective on the Conversion and Maintenance of Local Available Potential Energy in Extratropical Cyclones

1. Line 104: Why is omega only computed at 500-hPa. Wouldn't a lower level (e.g., 850-hPa or 700-hPabe be better for sampling and quantifying thermal advection?

We agree that a lower level would be more suitable to investigate thermal advection along the surface baroclinic zone. However, high APE values are mostly located in the middle and upper troposphere, which is why thermal advection along the surface baroclinic front does not incur large APE tendencies. Therefore, we believe that omega at 500 hPa is most suited for our purpose. Nevertheless, thermal advection along the lower tropospheric baroclinic zone could still affect QGomega at 500 hPa via the far-field effect. To test this hypothesis, we show QGomega due to lower-tropospheric forcing in Fig. R2.1 for the two time steps also shown in Fig. 6. The QGomega vertical motion at 500 hPa due to lower-tropospheric forcing is strongest in the immediate vicinity of cyclone 1a. However, its amplitude is much smaller than the forcing from upper levels (Fig. 6). Moreover, much of the vertical forcing associated with the descent of trajectories along the western flank of the upper-level trough is purely from upper levels. Therefore, we conclude that QGomega forcing from lower levels is less important for APE conversion.

[Figure]

*Fig. R2.1: Low-level quasi-geostrophic omega on 500 hPa (color shading) for two time steps. Also shown are the 2-PVU contour on the 315 K isentrope (green) and wind speed on 300 hPa (black contours; 60, 80 and 100 m/s).*

2. Line 112: Is it possible that your criterion for identifying CAOs could be sensitive to your choice of the 850-hPa level for air-sea temperature differences? Could shallow cold air be a problem in some situations?

We agree that shallow layers of cold air might not be detected as CAOs if they do not extend to the 850 hPa level. Such shallow layers of cold air are particularly common in the Arctic, for example, over sea ice when they are capped by a strong inversion. However, once the air is advected over open ocean, strong surface fluxes lead to a rapid growth of the boundary layer such that the CAO boundary layer will quickly extend beyond the 850 hPa level and the CAO can be detected (see for example composites of boundary layer heights in Fletcher et al. 2016, their Fig. 9). In the case of CAOs over the western boundary currents, the initial thickness of the boundary layer is likely less shallow than in the Arctic. Moreover, due to the strong horizontal SST gradient across the Gulf Stream front, the air-sea temperature difference quickly becomes large as the air progresses across the front (often far in excess of 8 K used as a threshold here), going along with very intense surface heat fluxes of typically several 100 W / m² (e.g., Grossman and Betts, 1990), and accordingly a rapid expansion of the boundary layer beyond 850 hPa (e.g. Vannière et al. 2017). Thus, in rare cases where the boundary layer is initially very shallow, the CAO would remain undetected only during its very early phase. In addition, the erosion by diabatic heating of very shallow cold layers could only have a small impact on the vertically integrated APE budget since they would represent only a small fraction of the mass in the atmospheric column. For these reasons, we think that for our purposes the 850 hPa level is acceptable.

3. Line 134: ….given by the product of omega and the buoyancy……

We have corrected the sentence according to your suggestion.

4. General question: What is the vertical resolution of your dataset? How well are you resolving shallow Arctic air outbreaks over continents and oceans?

We use the ERA5 reanalysis with its full 137 vertical model levels. Since the data is on hybrid sigma-pressure levels, the resolution is higher at lower levels. Up to 850 hPa, the vertical grid spacing ranges from 20-200m (more details can be found here: https://confluence.ecmwf.int/display/UDOC/L137+model+level+definitions). As mentioned above, we think that this vertical resolution sufficiently captures CAOs over the Gulf Stream.

5. Lines 142-148: I had trouble visualizing this somewhat convoluted text. Suggestion: Inserting a schematic diagram here that depicts the calculation of the reference state would be most helpful to the reader.

Thank you for pointing out this difficulty. We agree that the computation of the reference state is not straightforward. Therefore, we have included a schematic (Fig. R2.2) in the supplement and clarify parts of the text describing the methodology. Also, we restructured the text to add more clarity to the explantation of how we compute the reference state.

[Figure]

*Fig. R2.2: Schematic illustration of the computation of the reference state. (a) Meridional cross section of an idealized baroclinic zone composed of air parcels, represented by boxes. Color indicates potential temperature of the air parcels, increasing from blue to red. (b) Sorting of air parcels by increasing potential temperature. (c) Construction of the reference state from sorted air parcels illustrated by a section through the solid Earth (hatched) and the atmosphere (color, not to scale). The dashed lines indicate Earth's rotation axis and the equatorial plane (0° latitude). The atmosphere is assumed to be shallow such that each circle has the same circumference.*

6. Lines 150–151: Curious to learn why you chose not to compute reference states for DJF, MAM, JJA, and SON?

The reference state shows considerable seasonality as well as a trend due to global warming. We account for the trend by smoothing the reference state with a 9-yr running mean, as opposed to computing an average over all years. We take the same approach to retain seasonality by computing a 31-day running mean of the daily-mean reference state. Compared to a 31-day running mean, a DJF mean would lead to significant biases in the APE distribution and tendency between early winter and late winter time steps.
We realize that the description of how we compute the reference state in the manuscript is not sufficient. We have added the following to clarify (L166-169):
"To facilitate a comparison of APE between cyclones at different time steps, we apply a temporal smoothing to the instantaneous reference state vertical profiles. First, we compute daily averages, removing the daily cycle. Next, we apply a 31-day running mean, smoothing

out day-to-day variations but retaining the seasonality. Lastly, we apply a 9-year running mean to each calendar time step to take into account the effect of global warming on the reference state."

7. Figure 1: Could, say, 700-hPa vertical motion be added to the six figure panels so that the locations of major ascent and descent regions relative to trough and ridge positions can be better pictured?

Thank you for this suggestion. While we believe additional fields would be helpful here, we also want to avoid overcrowding the figure with a third set of contour lines. Therefore, we decided to not include the vertical motion.

8. Figure 1: Labels 1a, 1b, 2c, etc. need to be defined in the figure captions. Label latitude and longitude lines.

Thank you for pointing out the missing definitions. We have added them to the figure caption and added labels to the lat/lon lines.

9. Figure 1f: There appears to be a remnant cyclone near Iceland in panel 1f that is a remnant of cyclone 1b in panel 1e.

Good observation. There is a low pressure area between Greenland and Iceland. Even though our cyclone tracking scheme does not continue the track of cyclone 1b to 23 Feb 12 UTC, we believe that this cyclone might have arisen by a merger of cyclones 1a and 1b. Since we are mainly interested in the growth stage of the cyclones, where kinetic energy is generated, we believe that describing this remnant will distract the reader from the main message of the figure.

10. Figure 1: There appear to be three cold surges based on equatorward-directed trough extensions in Fig. 1. Significance? How do the three cold surges.

Yes, we agree that we observe three cold surges, in close succession and in a very similar synoptic setting, which we think is a particularly interesting aspect of this case study. The occurrence of these cold surges is likely linked to the fact that the case study is also identified as a European blocking weather regime, which implies some quasi-stationarity in the upstream storm track, as observed here.
The cold surges are related to the expansion and maintenance of the polar APE reservoir into the midlatitudes. For instance, cyclone 2a is accompanied by a strengthening of the APE tongue stretching into the central North Atlantic in its wake. This stage is likely also accompanied by cold-air advection in the lower troposphere and a CAO.
While this is certainly fascinating, we only very briefly speculate on the role of APE in maintaining this synoptic situation when we discuss cyclones 3a and 3b. We think that further speculation would go beyond the scope of the analysis and would not be supported by the following climatological analysis.

11. Figure 1: Cyclone 2a, and especially cyclone 3b, are not directly associated with an upper-level trough in their formative stages. Significance?

This is indeed an interesting observation. Upon investigating the upper-level forcing using QGomega at 500 hPa (Fig. R2.3) we find that cyclone 2a forms in a region of ascent (Fig. R2.3c,d). As you noted, the formation is not associated with a new trough propagating into the region. But earlier time steps indicate that the upper-level forcing interacting with cyclone 2a originated from the ageostrophic circulation on the poleward flank of an upstream jet streak (Fig R2.3a,b).

[Figure]

*Fig. R2.3: Upper-level quasi-geostrophic omega on 500 hPa (color shading) for four time steps, starting on 20 Feb 12 UTC with 12 hours intervals. Shown are the 2-PVU contour on the 315 K isentrope (blue) and wind speed on 300 hPa (black contours; 60, 80 and 100 m/s). The positions of cyclones 1a, 1b and 2a are also shown.*

The same analysis for cyclone 3b (Fig. R2.4) reveals that it quickly couples with the upper-level forcing due to the curvature of the upper-level trough associated with cyclone 3a (Fig. R2.4d). Cyclone 3b only formed very briefly before experiencing the upper-level forcing (Fig. R2.4c), possibly due to frontal instability.

To conclude, cyclones 2a and 3b are not directly associated with an upper-level trough during their initial phases. Nevertheless, they are substantially influenced by upper-level forcing and, concluding from our analysis, not fundamentally different from the other cyclones described in the case study. Therefore, we believe that the trajectory analysis conducted for cyclone 1a is representative of the general pattern of APE conversion within the case study.

[Figure]

*Fig. R2.4: Upper-level quasi-geostrophic omega on 500 hPa (color shading) for four time steps, starting on 23 Feb 12 UTC with 12 hours intervals. Shown are the 2-PVU contour on the 315 K isentrope (blue) and wind speed on 300 hPa (black contours; 60, 80 and 100 m/s). The positions of cyclones 2a, 2b, 3a and 3b are also shown.*

12. Lines 185-187: Please be more specific about where the APE decline is occurring.

We have rephrased as follows (L206-208):
"The dissipation of APE in the central North Atlantic and the end of intense North Atlantic cyclogenesis are accompanied by the development of an upstream cyclone, which inhibits further APE supply to the North Atlantic."

13. Figure 2 caption: Please indicate directions at either ends of the cross-section lines (e.g., west or east, etc.) for reference purposes.

Thank you for the suggestion. We have indicated the direction of the cross-sections with A-A', and also added this to Figure 1.

14. Lines 202-203: This sentence needs further elaboration since Fig. 2c is devoid of vertical motion.

We have rephrased as follows (L220-225):

"The zonally oriented cross section through the trough (Fig. 2c) further illustrates the cold anomaly below the trough. From left to right the isentropes slope upwards, concurrent with an increase of APE. Thus, the isentropic lifting beneath the upper-level trough is accompanied by large APE values. The APE located in this region is bounded by a jet stream toward the west and cyclone 1a toward the east."

[Figure]

*Fig. 2c: Cross section as indicated in Fig. 1. Shown are APE (color shading), the 2-PVU contour representing the dynamical tropopause (green), the horizontal wind speed (black contours; every 20 m/s) and potential temperature (grey contours; every 6 K).*

15. Figure 3: Geography is hard to distinguish from the color shading and black contours.

Thank you for pointing this out. We have changed the coloring of the geography.

16. Figure 3: SLP contours are very difficult to follow because they are so faint in Fig. 3 (and in Fig. 1)

Thank you for pointing this out. We have increased the thickness of the SLP contours such that they are more easily recognizable.

17. More on Fig. 3. So, is it fair to say that APE generation equatorward of the jet axis acts to increase APE in lower latitudes (with a minor offset by diabatic processes in the vicinity of midlatitude cyclones)? If so, what can we conclude about these APE changes from a general circulation perspective?).

The APE generation equatorward of the jet axis in Fig. 3 is due to air parcels ascending polewards of the equatorward flank of the jet, as shown and discussed in Fig. 4. Therefore, those trajectories resupply APE to the polar APE reservoir. At the same time, air parcels descending equatorward consume APE from the polar reservoir. This corresponds to a deceleration (KE → APE) and acceleration (APE → KE) of the flow, respectively. In a steady state, the global integral of the APE tendency would thus be zero. From the global circulation perspective, we expect this integral to be negative because KE is dissipated through friction, meaning there needs to be more local APE consumption than generation.

18. Line 247: Check whether AOE has been defined previously.

This should read APE. We have correced it accordingly.

19. Figure 4: SLP contours are too faint to be see properly, especially in the vicinity of the trajectory swaths.

Thank you for pointing this out. We have increased the thickness of the SLP contours such that they are more easily recognizable.

20. Figure 4b: How sensitive do you think the split between subsiding and ascending trajectories shown in Fig. 4b would be to the starting time and location of the trajectory computations? Presumably, the expected trajectory differences could be linked to the governing dynamics?

We are not sure about what you are referring to. All trajectories in Fig. 5b contribute to the depletion of APE. In this case, this corresponds to a negative adiabatic APE tendency and therefore to a descent. However, some trajectories descend a lot further than others. We think the determining factor is the height at which the trajectories are started. Trajectories from the upper troposphere have quite large initial APE values, and with that a large adiabatic efficiency. Thus, little vertical motion is required for them to contribute to APE depletion, whereas trajectories from the middle troposphere have lower initial APE values, requiring larger vertical displacement to result in the same APE depletion. Furthermore, we believe the trajectories depict a typical flow situation on the western flank of upper-level troughs. This might depend on the life cycle of the trough, but we don't expect large differences in trajectories contributing to APE depletions started from different times or locations.

21. Lines 264–272:  So, presumably the extent and magnitude of latent heat release in ascending trajectories would be very sensitive to the air mass in which the trajectories originate. For example, Mediterranean and Gulf of Mexico cyclones that have access to tropical and subtropical air masses would exhibit different APE changes in conjunction with ascending trajectory swarms than the higher-latitude example referenced in this paper? How should we interpret such differences?

You bring up an interesting aspect about the latitude-dependence of APE conversion and the role of latent heat release, which relate to two different aspects of the APE framework. First, we should clarify the role of latent heat release, which we also discuss in the manuscript. The local APE framework is dry-hydrostatic. Therefore, the acceleration of vertical motion through latent heat release is not explicitly represented. This means that while latent heat release results in a negative diabatic APE tendency, it also leads to a greater positive adiabatic tendency. We hypothesize that the positive tendency compensates for the negative tendency, and that the net contribution of latent heat release to the net APE tendency is positive. That said, among two cyclones at the same location, but with different moisture availability, the moister cyclone will exhibit more negative diabatic APE tendency and more positive adiabatic APE tendency in the ascending air stream than the drier cyclone.

Second, the magnitude of those APE tendencies will depend on the latitude at which the cyclone occurs. Cyclones further south typically form in a warmer environment, which means that the adiabatic and thermal efficiencies of APE conversion will be different from cyclones in a colder (northern) environment. However, the efficiency affects both the positive and negative *local* tendencies equally. Thus, we would expect the *volume-integrated* tendency in the vicinity of the cyclone to be unchanged by the latitude.

In any case, the question of how cyclones in different locations contribute to the global integral of APE conversion is very interesting and we have included this point in the outlook of our manuscript.

22. Figure 5 (and accompanying text). Have you checked whether a mesoscale trailing trough behind the main trough could be altering the larger-scale QG ascent-descent dipoles? Would it be helpful to add the position of the surface cyclone to Figs. 5a,b to assist the reader in better understanding the relationship between the trajectory evolution and the surface cyclone location?

Indeed the discussion around your point Nr. 11 and Fig. R2.3/2.4 points towards a mesoscale mobile trough / frontal zone trailing the main trough. We think this feature mainly influences the development of cyclone 2a, whereas the discussion around Fig. 6 is focused on cyclone 1a. Thank you also for your suggestion to include the position of the surface cyclones in Fig. 6, which we have implemented.

23. Lines 288-292: While I think that your conclusions mentioned here are correct to first order, it might be a good idea to allow for considerable variability the would likely be seen in a much larger sample of cases (e.g., a secondary trough deepening downstream into a pre-existing trough vs. a weakening trough that was lifting out to the northeast (among other possible trough-ridge scenarios).

Thank you for raising this point, which was also a concern of reviewer #1. Indeed the vertical forcing from upper levels is a combination of different processes, which might arise by the presence of different features, such as troughs, secondary troughs or upper-tropospheric frontal zones and jet streaks. We have added a discussion of the different synoptic processes to the Discussion section (L533-537):
"In addition to the curvature associated with an upper-level PV anomaly, other synoptic environments can contribute to ageostrophic vertical circulations and influence APE tendencies. For instance, the thermally direct transverse circulation in the entrance region of a jet streak is expected to convert APE into KE, while the thermally indirect transverse circulation at the jet exit would convert KE back into APE. Furthermore, temperature advection across the jet can influence the location of the ageostrophic vertical circulation (Keyser and Shapiro, 1986), thereby affecting the associated APE tendencies."

24. Lines 294-311: Very interesting perspective on APE conversion that is well supported by the trajectories shown in Fig. 6 and the APE tendency plots shown in Fig. 7. An obvious question….that really cannot be addressed here….is how general this result would would be across a variety of cyclone evolutions at different times of the year and the choice of where and when to initiate the trajectories.

Thank you. We think that the climatological analysis in the second part supports this result at least for DJF Gulf Stream cyclones. While the question of seasonality is certainly interesting, we would expect the strongest effect of surface fluxes on APE to be concentrated in DJF due to the large land-sea temperature contrast in winter.

25. Fig. 7: Would it be helpful to rescale the y-axes in Fig. 7 so as to make it more obvious how much smaller the diabatic APE tendency is compared to the adiabatic APE tendency?

We agree that this would help to put the difference between the adiabatic and diabatic tendencies better into perspective. However, it then becomes harder to recognize the dip in diabatic tendency around hour 0, which is still significant (Fig. R2.5 shows equal scales). As a middle ground, we have adjusted the y-scale of the diabatic tendency to be exactly half of the adiabatic tendency in Fig. 9. With that, the timeseries of the diabatic tendency is still readable but at the same time it's easier to grasp the difference in scale to the adiabatic tendency.

[Figure]

Fig. R2.5: Evolution of (a) APE density, (b) pressure, (c) adiabatic APE tendency and (d) diabatic APE tendency of CAO trajectories, centered on their arrival in the CAO region (time zero). The solid black line indicates the mean, and the grey shading denotes the range between the 5th and 95th percentile of the trajectories.

26. Line 316 (first bullet): How might this finding change (or not change) if you looked at Arctic PV extrusions across all longitudes as a function of season?

Analyzing the seasonality of APE lies outside the scope of our study, but we do believe that this is a very interesting aspect that could be explored in future work, and we have mentioned it in the outlook section. We hypothesize that, to first order, the character of the results wouldn't change. Meaning that Arctic PV excursions will be associated with large APE values relative to climatology in any season. This is well substantiated by PV dynamics

and upward doming of isentropes beneath PV troughs. However, absolute APE values would be a lot smaller in the summer than in the winter hemisphere, as the meridional temperature contrast is weaker. This expectation is also in line with less and weaker cyclogenesis in the summer hemisphere.

27. Line 320 (second bullet): While I agree with this conclusion, to play the contrarian a bit and reference "Animal Farm" meteorology…..while all cyclones may be equal, some cyclones may be more equal than other cyclones. How do we find these "more equal" cyclones and how can we understand why there are "more equal cyclones" with regard to where they form, how they evolve, and how they contribute to APE generation? For example, 1) How "unequal" are continental cyclones from oceanic cyclones?, and 2) Are APE changes sensitive to whether cyclones are moving mostly poleward (e.g., cyclones along the east coasts of Asia and North America) or moving mostly equatorward (e.g., lee cyclones east of major north-south mountain barriers? Just curious……

While we do not object to this fundamental truth, "Animal Farm" meteorology also states that "Man serves the interests of no creature except himself". In a (dystopian) world, this quote refers to a common and inherent characteristic among all people. Likewise, cyclones share a set of features common to all cyclones. All cyclones intensify to some degree, converting APE to KE to achieve their very own intensification. It is this fundamental commonality among cyclones that we aim to describe in our manuscript, and how this process connects the theory of the general circulation with synoptic dynamics.
For our climatological analysis we specifically focus on a relatively small selection of deep Gulf Stream cyclones to reduce the variability in our composite. Although we don't expect differences in the fundamental character of APE conversion between cyclones (as also supported by our climatology), we agree that it would be interesting to compare cyclones developing in different regions, following different life-cycles or with a different amount of precipitation formation and by that diabatic heating. In fact, we believe that local APE could be a useful metric for the analysis of the atmospheric states giving rise to cyclogenesis more generally. For instance, how is APE supplied to cyclones outside the oceanic storm tracks? And does the absence of APE inhibit cyclone growth? More fundamentally, we think that this question links to the comparison of local APE and baroclinicity, as we have discussed in our outlook section.

28. Line 325: Same question as previously with regard to how general your conclusions are.

Given our climatological analysis in the following section, we are confident that our results are sufficiently general. We analyzed the APE tendency in CAO environments over the Gulf Stream region in the wake of intense extratropical cyclones. These conditions are known to be associated with the largest surface fluxes climatologically (e.g. Papritz and Spengler, 2017). In any other region, it is reasonable to assume that the surface fluxes in the wake of cyclones would be smaller, and hence the diabatic tendency due to those surface fluxes would be even smaller as well. Therefore, their significance would be further diminished, which supports our conclusion.

29: Lines 316-327: What about cyclone families? Lots of cyclones are a part of cyclone families. With respect to cyclone-related APE changes, might there be any relationship between cyclone family duration and overall APE changes?

Cyclone families are a very interesting case to consider. In fact, both cyclones 1b and 2b are frontal cyclones, forming cyclone families with cyclones 1a and 2a, respectively. In both cases, we observe and describe how the secondary cyclogenesis maintains the APE within the storm track by (re)intensifying the upper-level trough. Even though this is an interesting aspect, the focus of the manuscript lies on Gulf Stream cyclones to minimize variability within our climatological analysis.

But we can speculate on the impact of cyclone families on APE changes. We would expect a secondary cyclone to have less impact on overall (i.e. globally integrated) APE than a primary cyclone, since the primary cyclone and its baroclinic interaction with upper levels is typically setting up an intensified large-scale circulation. It is this environment that then promotes secondary cyclogenesis. This argument is supported by the fact that most APE is converted in the upstream storm tracks (Novak & Tailleux, 2018), whereas secondary cyclogenesis mostly happens in the downstream parts of the storm tracks (Priestley et al., 2020).

30. Figure 9: This figure, comprised of 12 panels, is very difficult to read and assimilate. It is also insufficiently labeled (e.g., orange contours on panels e-h). Might four, four-panel figures work better….and not overwhelm readers? Failing that, perhaps include an option to choose magnifying glasses with the journal when reading this article?

On one hand, we agree that the size of the figures might be a problem for the readers. On the other hand, we regard it as important to show the figures together to
1) show the life cycle of the cyclone
2) show how APE and its tendency compare to the PV field
3) show how the total APE tendency is split into its diabatic and adiabatic contributions
Separating the figure into smaller figures would make this comparison considerably harder. To make the figure more accessible, we have additionally provided larger versions of the figure in the supplement with 2x4 panels. Also, we have improved the labeling of the contours.

31. Text supporting Fig. 9 is a big help in sorting through the overly small figures. For example, the conclusions mentioned on lines 355-358 are well supported by the individual panels in Fig. 9. Likewise, for panels (e-h) and (i-l).

Thank you!

32. Do the very small differences between panels 9n and 9o reflect a smaller time difference and/or the rotation of the precipitation shield around the composite cyclone?

We think that the rotation of the precipitation band and with that the diabatic tendency between panels 10n and 10o reflects the establishment of a distinct cold front in panel 9o.

33. How sensitive are the results shown in Fig. 10 to the domain size over which the calculations were made?

Thank you for bringing up the sensitivity of our analysis, which was also mentioned by reviewer #1. As we discuss in the text, the main factor influencing this sensitivity is the fact

that the positive APE tendency is mostly found close to the center of the cyclone, whereas the negative APE tendency can be located further away in later stages of the cyclone life cycle. We think that the chosen radius adequately addresses this issue while still reflecting the dynamics of the synoptic system. To clarify this, we have recomputed the spatial integral for different radii and included the resulting figures in the supplement. At a smaller radius of 1000 km (Fig. S8), we notice that the negative adiabatic APE tendency is not captured well, which results in a positive adiabatic APE tendency. Similarly, the diabatic APE tendency mainly captures the concentrated negative tendency due to latent heat release, while the weaker positive tendency due to radiative cooling is not well captured. At a bigger radius of 2500 km (Fig. S10), the life cycle of the cyclone is more muted compared to Fig. 12 because more noise is included.

[Figure]

*Fig. S8: (a) Distribution of minimum mean sea-level pressure of the 285 selected cyclones, averaged +/- 6h around the four life cycle stages. The (b) net, (c) adiabatic and (d) diabatic APE tendency is integrated in a 1000 km radius around the cyclone center within a +/- 6 h window around the four life cycle stages. Whiskers show the 10th-90th percentile range.*

[Figure]

*Fig. S10: Same as Fig. S5, but for an integration radius of 2500 km.*

34. There is little if any downward extrusion of PV in figure panel 11a. Is this an artifact of the cross section location choice and/or does it reflect that at the time the cyclone maximum intensification that the PV is mostly "used up" (so to speak)?

This is indeed an artifact of the choice of the cross section location. Figure R2.6 shows the cross section through the center of the cyclone at the time of maximum intensification (same as Fig. 12). From this perspective the PV tower is easily recognizable. However, we lose the information on the vertical structure of the APE depletion in the cold sector of the cyclone in Fig. 12b.

[Figure]

*Fig. R2.6: Cross sections of cyclone-centered composites at the time of maximum intensification. (a)-(c) are meridional cross sections and (d)-(f) are zonal cross sections through the cyclone center. (a),(d) PV (shading) and horizontal wind speed (blue; every 10 m/s. (b),(e) adiabatic APE tendency (shading) and APE density (orange contours; every 500 J/kg). (c),(f) diabatic APE tendency. Potential temperature is indicated by grey contours (every 6 K).*

35. There needs to be additional labeling of the lines in Fig. 11 (e.g., panels e and f are devoid of contour labels). Additional labels would make it easier for a reader to follow the text better on lines 425-431.

Thank you for pointing out the missing contour labels. We have added additional contour labels for both the wind speed and APE tendency, as well as some contour labels for the potential temperature.

36. There is a factor 4 difference in the vertical scale for Figs. 12c and 12d. Consider using the same vertical scale for these two figure panels to facilitate a better understanding of the difference in the adiabatic and diabatic APE tendencies?

As discussed for your point 25, we believe that applying the same scale to both panels would make it very hard to decipher the evolution of the diabatic tendency. But because we agree with your argument, we have used half the scale to facilitate the interpretation of the differences.

37. Discussion surrounding Fig. 12 is excellent.

Thank you!

38. Nice synthesis in section 5.1. Would be to comment how the results from a sample of 285 intense NATL cyclones might differ from a much larger overall sample of cyclones?

Thank you for pointing this out. We agree that the variability among different classes of cyclones and/or different regions would be very interesting to investigate and we have included this aspect in the outlook section.

39. Lines 479-485: I fully agree with this interpretation from a synoptic-dynamic/weather map perspective.

Thank you.

40. Lines 485-495: How might your results differ if you distinguished between cyclogenesis associated positively tilted versus negatively tilted troughs? Typically, positively (negatively) tilted troughs are associated with the strongest jet in the southwesterly (northwesterly) flow downstream (upstream) of a tilted trough axis.

This is an interesting point. Our case study analysis contains cyclones with a northwesterly jet upstream of the main trough axis (1a and 2a), and the secondary cyclones with a southwesterly jet downstream of the main trough axis. While our composite study focuses on the upstream storm track region, we think it mostly contains cyclones of the former case. However, the character of APE conversion remains unchanged of the orientation of the upper-level jet changes, i.e. the ageostrophic circulation on the poleward flank of the jet stream will still consume APE, as the poleward flank is still facing towards the trough axis and with that towards the APE reservoir.

41. Lines 505-513: I am still pondering this conclusion because there is so much variability between individual cyclones. Might comparing cyclones in NW versus SW flow possibly be helpful?

We think that this question largely falls back onto different categories of cyclones, which opens up many avenues of new research, e.g., how do the different roles of latent heat release in diabatic Rossby waves, frontal cyclones and primary cyclones compare in terms of the APE framework. However, this comparison lies outside the scope of our analysis. Since we selected very strong Gulf Stream cyclones, for which the role of diabatic heating can be assumed to be similar (e.g. Binder et al. 2016), we believe that our conclusions are well supported.
However, we agree that the reader might wonder about the variability of the cyclones we consider for our composite. We don't think this aspect is discussed sufficiently in the choice of the cyclone tracks. Therefore, we have added more clarification on why we think that our choice of cyclone tracks minimizes variability within our composite on L372-375:
"Our selection process minimizes variability among the chosen cyclones. First, the geographical criterion ensures a focus on Shapiro-Keyser cyclones that predominantly propagate poleward along the Gulf Stream sea surface temperature (SST) front

(Fig. 10a). Second, the intensity criterion ensures that only cyclones of substantial strength are selected, facilitating more direct 390 comparisons and minimizing variability in our composite study."

42. Lines 520-522: Good point. I had never thought this concept though previously in the depth needed to better appreciate the differences.

Thank you!

43. Lines 533-541: Interesting point. Would it be helpful to distingush between high versus low dynamical tropopause storms?

This is an interesting idea. For instance, it would be interesting to see whether high dynamical tropopause storms convert similar amounts of APE as low dynamical tropopause storms, and how this affects their intensification. We have included this thought in the outlook section.

44. Lines 542-546: It might be helpful to distinguish between early season cyclones(when the troposphere is still relatively warm and the dynamic tropopause is still relatively high), from midseason cyclones (when the baroclincity is strongest), and late season cyclones (when warm sector CAPE is becoming a factor in cyclone development and intensification.

In general, we think that the decreased differential heating of Earth and the weaker cyclone intensities outside DJF would imply weaker APE conversions for early and late season cyclones. But this then raises the question of the conditions surrounding very intense early and late cyclones and where their APE originates from. As you point out, this might also relate to the previous point and the height of the dynamical tropopause. Lastly, the role of warm sector CAPE would then also point towards a shift in energy generation from adiabatic APE consumption to diabatic sources, which would be interesting to investigate in more detail in future work. We have added a discussion of this point to the outlook section.

**References:**

Binder, H., M. Boettcher, H. Joos, and H. Wernli, 2016: The Role of Warm Conveyor Belts for the Intensification of Extratropical Cyclones in Northern Hemisphere Winter. *J. Atmos. Sci.*, **73**, 3997–4020, https://doi.org/10.1175/JAS-D-15-0302.1.

Bowley, K. A., J. R. Gyakum, and E. H. Atallah, 2019: The Role of Dynamic Tropopause Rossby Wave Breaking for Synoptic-Scale Buildups in Northern Hemisphere Zonal Available Potential Energy. *Mon. Wea. Rev.*, **147**, 433–455, https://doi.org/10.1175/MWR-D-18-0143.1.

Grossman, R. L., & Betts, A. K. (1990). Air-sea interaction during an extreme cold air outbreak from the eastern coast of the United States. *Mon. Wea. Rev.*, 118, 324–342, https://doi.org/10.1175/1520-0493(1990)118<0324:AIDAEC>2.0.CO;2

Federer, M., L. Papritz, M. Sprenger, C. M. Grams, and M. Wenta, 2024: On the Local Available Potential Energy Perspective of Baroclinic Wave Development. *J. Atmos. Sci.*, **81**, 871–886, https://doi.org/10.1175/JAS-D-23-0138.1.

Fletcher, J., Mason, S., & Jakob, C. (2016). The climatology, meteorology, and boundary layer structure of marine cold air outbreaks in both hemispheres. *J. Climate*, **29**, 1999–2014. https://doi.org/10.1175/JCLI-D-15-0268.1

Novak, L., and R. Tailleux, 2018: On the Local View of Atmospheric Available Potential Energy. *J. Atmos. Sci.*, **75**, 1891–1907, https://doi.org/10.1175/JAS-D-17-0330.1.

Papritz, L., & Schemm, S. (2013). Development of an idealised downstream cyclone: Eulerian and Lagrangian perspective on the kinetic energy. *Tellus A: Dynamic Meteorology and Oceanography*, **65**(1). https://doi.org/10.3402/tellusa.v65i0.19539

Papritz, L., and T. Spengler, 2017: A Lagrangian Climatology of Wintertime Cold Air Outbreaks in the Irminger and Nordic Seas and Their Role in Shaping Air–Sea Heat Fluxes. *J. Climate*, **30**, 2717–2737, https://doi.org/10.1175/JCLI-D-16-0605.1.

Priestley MDK, Dacre HF, Shaffrey LC, Schemm S, Pinto JG. The role of secondary cyclones and cyclone families for the North Atlantic storm track and clustering over western Europe. *Q J R Meteorol Soc*. 2020; **146**: 1184–1205. https://doi.org/10.1002/qj.3733

Vannière, B., A. Czaja, H. Dacre, and T. Woollings, 2017: A "Cold Path" for the Gulf Stream–Troposphere Connection. *J. Climate*, **30**, 1363–1379, https://doi.org/10.1175/JCLI-D-15-0749.1